# Catalytically potent and selective clusterzymes for modulation of neuroinflammation through single-atom substitutions

Haile Liu[1,7], Yonghui Li[1,7], Si Sun[1,7], Qi Xin[1], Shuhu Liu[2], Xiaoyu Mu[1], Xun Yuan[3], Ke Chen[1], Hao Wang[1], Kalman Varga[4], Wenbo Mi[1], Jiang Yang[5] & Xiao-Dong Zhang[1,6✉]

Emerging artificial enzymes with reprogramed and augmented catalytic activity and substrate selectivity have long been pursued with sustained efforts. The majority of current candidates have rather poor catalytic activity compared with natural molecules. To tackle this limitation, we design artificial enzymes based on a structurally well-defined $Au_{25}$ cluster, namely clusterzymes, which are endowed with intrinsic high catalytic activity and selectivity driven by single-atom substitutions with modulated bond lengths. $Au_{24}Cu_1$ and $Au_{24}Cd_1$ clusterzymes exhibit 137 and 160 times higher antioxidant capacities than natural trolox, respectively. Meanwhile, the clusterzymes demonstrate preferential enzyme-mimicking catalytic activities, with $Au_{25}$, $Au_{24}Cu_1$ and $Au_{24}Cd_1$ displaying compelling selectivity in glutathione peroxidase-like (GPx-like), catalase-like (CAT-like) and superoxide dismutase-like (SOD-like) activities, respectively. $Au_{24}Cu_1$ decreases peroxide in injured brain via catalytic reactions, while $Au_{24}Cd_1$ preferentially uses superoxide and nitrogenous signal molecules as substrates, and significantly decreases inflammation factors, indicative of an important role in mitigating neuroinflammation.

[1] Department of Physics and Tianjin Key Laboratory of Low Dimensional Materials Physics and Preparing Technology, School of Sciences, Tianjin University, 300350 Tianjin, China. [2] Beijing Synchrotron Radiation Facility (BSRF), Institute of High Energy Physics (IHEP), Chinese Academy of Sciences (CAS), 100049 Beijing, China. [3] School of Materials Science and Engineering, Qingdao University of Science and Technology, 266042 Qingdao, Shandong, China. [4] Department of Physics and Astronomy, Vanderbilt University, Nashville, TN 37235, USA. [5] School of Medicine, Sun Yat-sen University, 510060 Guangzhou, China. [6] Tianjin Key Laboratory of Brain Science and Neural Engineering, Academy of Medical Engineering and Translational Medicine, Tianjin University, 300072 Tianjin, China. [7] These authors contributed equally: Haile Liu, Yonghui Li, Si Sun. ✉email: xiaodongzhang@tju.edu.cn

Due to their exclusive catalytic activity and selectivity, artificial enzymes are exploited as promising tools for wide-reaching biomedical implications[1–8], particularly as advanced diagnostics[9,10] and therapeutics[11–16] of diseases. Earlier studies shed light on the oxidase- and peroxidase-like activities of noble metals[17]. Gold-based materials were unraveled to possess versatile enzyme-like activities, such as nuclease, glucose oxidase, peroxidase (POD), catalase (CAT), and superoxide dismutase (SOD)[17,18]. The Michaelis–Menten constant ($K_m$) to the $H_2O_2$ substrate of gold nanoparticles toward the POD enzymatic reaction is below 1 mM, but the catalytic activity is weak[19]. In contrast, Pt-based materials generally confer a high overall catalytic activity but it can only show a good $H_2O_2$ substrate affinity when $K_m$ is up to 16.7 mM[14,20], and modulation of selective catalysis often needs to be purposely realized through rationally designed combination with other catalysts[21]. Meanwhile, metal oxides have also revealed great potentials as enzyme mimetics[22]. Typically, $Fe_3O_4$ nanoparticles display the POD-like activity[23,24] but are limited by their affinity to the $H_2O_2$ substrate ($K_m$ at ~154 mM) and a maximal reaction rate (5.9 μM/min) that do not meet expectations. $Mn_3O_4$ nanoparticles concurrently exhibit SOD-, CAT-, and glutathione peroxidase (GPx)-like activities via the redox switch between $Mn^{3+}$ and $Mn^{4+}$ with a maximum reaction rate reaching 6–125 mM/min at nanomolar levels, which is unfortunately still inferior to natural enzymes[25]. Thus the development of catalytic artificial enzymes with exceptional activity, adequate selectivity, and satisfactory stability remains a major challenge for any foreseeable practical applications.

As is well known to all, most brain injuries involve enzyme-related catalytic processes and continuous neuroinflammation[26–28]. However, it is largely unclear yet which specific catalytic route(s) can be selectively targeted to inhibit neuroinflammatory responses, primarily because brain injuries simultaneously trigger various kinds of multi-enzymatic reactions between free radicals and numerous bioactive molecules[29]. Therefore, exploration of versatile artificial enzymes with different catalytic routes and desirable selectivity is beneficial to establish the relationship between oxidative stress and inflammation and to reveal the underlying molecular pathways of catalysis[30–32]. Atomic-level catalysts suffice a viable solution for the unmet need of improved catalytic activity and precisely modulated selectivity in a controllable manner, with lots of Fe- and Pt-based single-atom nanozymes developed[33–40]. In particular, Au contains excessive transition metal electronic states and rich electronic energy levels, which provide a solid basis for designing atomic-scale enzyme. Nevertheless, hindered by uncontrollable syntheses and complicated spatial coordination, it is difficult to reveal their electronic structures accurately, which can further influence the catalytic activity and prevent researchers from understanding exact catalytic mechanisms at atomic levels[41].

In this work, an exemplified Au-based clusterzyme is rationally designed at atomic precision with ultrahigh catalytic activity and superiority over natural antioxidants, and favorable enzymatic selectivity can be achieved via exquisite single-atom substitution by modulating single Cu or Cd active site, consequently serving as a promising artificial enzyme with tuned catalytic selectivity for treatment of neuroinflammation in the brain.

## Results

### Structural properties of clusterzymes

Exceedingly different from most previously reported nanozymes[1,4,5], the as-developed 3-mercaptopropionic acid (MPA)-protected $Au_{25}$ clusterzyme is stringently defined by its unambiguous atomic configuration and geometry structure (Fig. 1a). The hydrodynamic size of $Au_{25}$ is determined to be 2.0 nm by dynamic light scattering (DLS), and the zeta potentials of all clusterzymes are around −35 mV,

suggesting the ultrasmall size and good colloid stability (Supplementary Fig. 1). The characteristic absorption at 450 and 670 nm of $Au_{25}$ is attributed to its unique interband transitions[42,43], while a single-atom substitution of Cu or Cd induces a 2–3-nm minor shift, showing insignificant influence on optical properties (Fig. 1b and Supplementary Fig. 2). Electrospray ionization–mass spectra (ESI-MS) reveal a distinct $m/z$ peak at ~2271, assigned to $[Au_{25}MPA_{18}–3H]^{3-}$ (Fig. 1c). After one atom substitution by Cu and Cd, the characteristic $m/z$ peak shifts to ~2226.6 and ~2243, respectively. The inductively coupled plasma-mass spectrometry (ICP-MS) confirms that the ratios of Cd and Cu to the total metal are 5 and 4%, respectively, further validating the successful introduction of single atoms (Supplementary Fig. 3). X-ray photoelectron spectroscopy (XPS) further confirms that Au (0) is the dominant state in all clusterzymes (Supplementary Fig. 4). To identify the precise spatial atomic configuration, extended X-ray absorption fine structure (EXAFS) spectra at the Au, Cu, and Cd edges were recorded (Fig. 1d, e and Supplementary Figs. 5 and 6). The $L_3$ edges of Au in all clusterzymes have higher white-line intensities than the bulk standard Au foil. This is ascribed to larger surface area and alloying effects from partial oxidation with more $d$-band vacancies from nanoscale sizes and surface molecule-like interactions (Au(I)–thiolate). The characteristic absorption edges of Au clusterzymes were found at ~11,920 eV, which is assigned to the $2p \rightarrow 5d$ electronic transition of Au suggesting a reduced population of unoccupied valence $d$-states. The increase of intensity in MPA-protected $Au_{24}Cu_1$ and $Au_{24}Cd_1$ indicates that the density of $5d$ electrons of Au is decreased by the one atom substitutions of Cu and Cd through the transfer of their $4d$ electrons (Supplementary Fig. 5)[44,45]. The $k$-space oscillations of $Au_{25}$ clusters and the Au foil are shown in Supplementary Fig. 5. The $k$-space of the Au foil exists in typical fcc oscillation patterns, which are apparently absent in all Au clusterzymes due to their small core sizes. Besides, we also investigated the X-ray absorption near edge structure (XANES) spectra of Cu and Cd foils as well as the corresponding atomic counterparts within clusterzymes, clearly displaying differences between single atoms and bulk metals (Supplementary Fig. 6). To further pinpoint the doping sites of Cu and Cd atoms, we performed fitting analysis on the EXAFS data of Cu and Cd. Figure 1d shows the $R$ space of the EXAFS data of the Cu K edge in $Au_{24}Cu_1$. It can be seen that there is only one major peak in the range of 1.6–5.0 Å. This peak roughly corresponds to the scattering path of photoelectron waves from the X-ray absorbing Cu atom to the neighboring S atoms of different shells, and IFEFFIT program is used to fit this peak. The EXAFS parameters obtained after fitting are shown in Supplementary Table 1. The Cu-S coordination number (CN) obtained from the fitting is 1.9 ± 0.2 Å. This value is close to 2 Å, which may indicate that the replacement of $Au_{25}$ by a Cu atom occurs at the oligomer site, consistent with previous work[44]. Similarly, the $R$ space of EXAFS data on Cd $L_3$ edge in $Au_{24}Cd_1$ shows a peak in the range of 1.6–5.0 Å, and the fitted Cd-S CN is 2.3 ± 1.7 Å, which is close to the CN of the bond with S at the oligomer site of $Au_{25}$, indicating that Cd atom substitution may occur at the oligomer site (Fig. 1e)[46].

### Antioxidant properties of clusterzymes

We tested the general antioxidative properties of all clusterzymes using the ABTS method (Fig. 2a) with reference to standard antioxidants trolox and anthocyanin. Negligible antioxidant activity was observed for pure MPA-protected $Au_{25}$ clusterzyme. Single-atom substitutions with Cu or Cd in the structure, however, induce a dramatic increase in antioxidant activity with increasing concentrations (Fig. 2b). Among a variety of metals including Ag, Cu, Zn, Er, Pt,

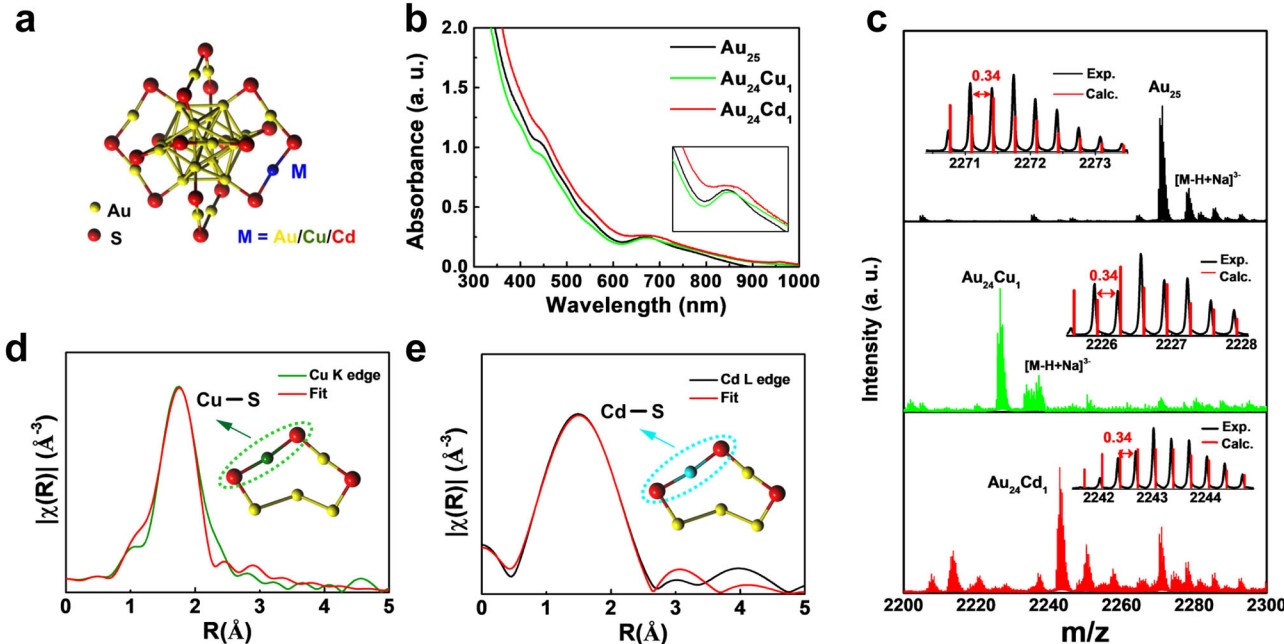

**Fig. 1 Structural characterization of Au25 clusterzymes. a** Structure illustration of $Au_{25}$ and Cu- and Cd-substituted $Au_{24}Cu_1$ and $Au_{24}Cd_1$. **b** UV-vis absorption spectra and **c** electrospray ionization mass spectra (in negative ion mode) of $Au_{25}$ (black curve) before and after Cu (green curve) and Cd (red curve) substitutions. The inset in **b** is a magnification of the absorption spectrum at ~670 nm. It can be seen that the characteristic absorption is slightly redshifted after introduction of Cu and Cd, indicating decreased band gaps. The red line of the insets in **c** represents the simulated isotope distribution of $[Au_{25}MPA_{18}-3H]^{3-}$, $[Au_{24}Cu_1MPA_{18}-3H]^{3-}$, and $[Au_{24}Cd_1MPA_{18}-3H]^{3-}$, respectively. Exp. experiment, Calc. calculated. **d** Cu $K$-edge and **e** Cd $L_3$-edge FT-EXAFS spectra and associated fitting in $R$ space of $Au_{24}Cu_1$ and $Au_{24}Cd_1$, showing the surrounding atoms adjacent to the Cu and Cd atoms.

and Cd, single-atom substituted candidates $Au_{24}Cu_1$ and $Au_{24}Cd_1$ show the highest activity, representing the optimal substituting elements and ratios (Supplementary Fig. 7). Time-course kinetics of $Au_{24}Cu_1$ and $Au_{24}Cd_1$ exhibit rapid responses to the substrate in seconds with high reaction rates (Fig. 2c). The quantitative results show $Au_{24}Cu_1$ and $Au_{24}Cd_1$ are 41 and 48 times higher in antioxidant activity than $Au_{25}$, respectively (Fig. 2c). Compared with standard natural antioxidant controls, $Au_{24}Cu_1$ and $Au_{24}Cd_1$ are 137 and 160 times higher in activities than trolox and 7.5 and 9 times higher than anthocyanin, respectively. The reaction rates of $Au_{24}Cu_1$ and $Au_{24}Cd_1$ at 10 and 14 μM/s are 8–11 times higher than $Au_{25}$ or 38 and 51 fold higher than trolox, respectively. In addition, in a parallel comparison with other elements, substitution with exactly one atom of Cu or Cd presents the foremost activity amidst all substituents (Supplementary Fig. 7d). Preceding studies have evidenced that atomically precise gold clusters, such as $Au_{25}$ and $Au_{38}$, are endowed with the oxidation catalytic activities[47–50], but their antioxidant activities are rarely reported. Herein, we discovered its ultrahigh antioxidant activity with fast kinetics via atom substitution.

**Enzyme-like properties of clusterzymes**. The general catalytic profile of clusterzymes and the schematic diagram showing catalytic processes are displayed as in Fig. 3a, b. To pinpoint catalytic selectivity of these clusterzymes, we first investigated the GPx-like activity of $Au_{25}$, $Au_{24}Cu_1$ and $Au_{24}Cd_1$ at the concentration of 10 ng/μL. Surprisingly, $Au_{25}$ shows the strongest tendency toward GPx-like activity with a maximum reaction rate of 0.47 mM/min, higher than 0.34 mM/min for $Au_{24}Cu_1$ and 0.10 mM/min for $Au_{24}Cd_1$ (Fig. 3c), and also significantly higher than those of previously reported $Mn_3O_4$ nanoflowers (0.056 mM/min)[51] and Co/PMCS (0.013 mM/min)[52]. The turnover frequency (TOF)

value of $Au_{25}$ calculated by the Michaelis–Menten equation is 320 $min^{-1}$, 4.7 times higher than $Au_{24}Cd_1$ (Supplementary Fig. 8). This result is interesting because metals are generally considered to have low GPx-like activity, but the high GPx-like activity of $Au_{25}$ can be exploited to eliminate lipid peroxides and oxidative damages. The CAT-like activity of clusterzymes were studied at the concentration of 20 ng/μL as in Fig. 3d. The maximum reaction rate of $Au_{25}$ is 0.074 mM/min, whereas the introduction of a Cu single atom gives rise to a 4.7-fold increase to 0.35 mM/min, suggesting its CAT-like catalytic preference. The calculated TOF value of $Au_{24}Cu_1$ for CAT-like activity is 116.7 $min^{-1}$ (Supplementary Fig. 9), which is significantly higher than that of Pd octahedrons (1.51 $min^{-1}$)[53]. The SOD-like activity of pure $Au_{25}$ can only inhibit 31% of the substrate, while one Cd atom substitution considerably increases the inhibition rate to 89%, empowering SOD-like selectivity (Fig. 3e). The aforementioned results suggest enzyme-mimicking preferences of each individual clusterzyme: $Au_{25}$ as GPx and $Au_{24}Cu_1$ and $Au_{24}Cd_1$ as CAT and SOD, respectively. The structures of clusterzymes before and after reaction with $H_2O_2$ suggest unchanged structures of the clusterzymes (Supplementary Figs. 10 and 11)[54,55]. Previous work mainly focused on the atomic substitutions of $Au_{25}$ using noble metals for catalytic reactions of hydrogen and $CO/CO_2$[56–59]. Our work herein constructively hypothesized and demonstrated that $Au_{25}$ can possess various unique enzyme-like activity modulated by single-atom substitution with non-precious metals like Cu and Cd, instead of Pt, in the geometric structure.

The corresponding specific scavenging of free radicals by the clusterzymes was further investigated. The scavenging of •OH free radical was investigated using electron spin resonance (ESR) by employing 5-tert-butoxycarbonyl-5-methyl-1-pyrroline N-oxide (BMPO) as the trapping agent. The ESR signal of •OH is strong for the BMPO control, suggesting the presence of excessive •OH, while there is only a minor decrease after adding

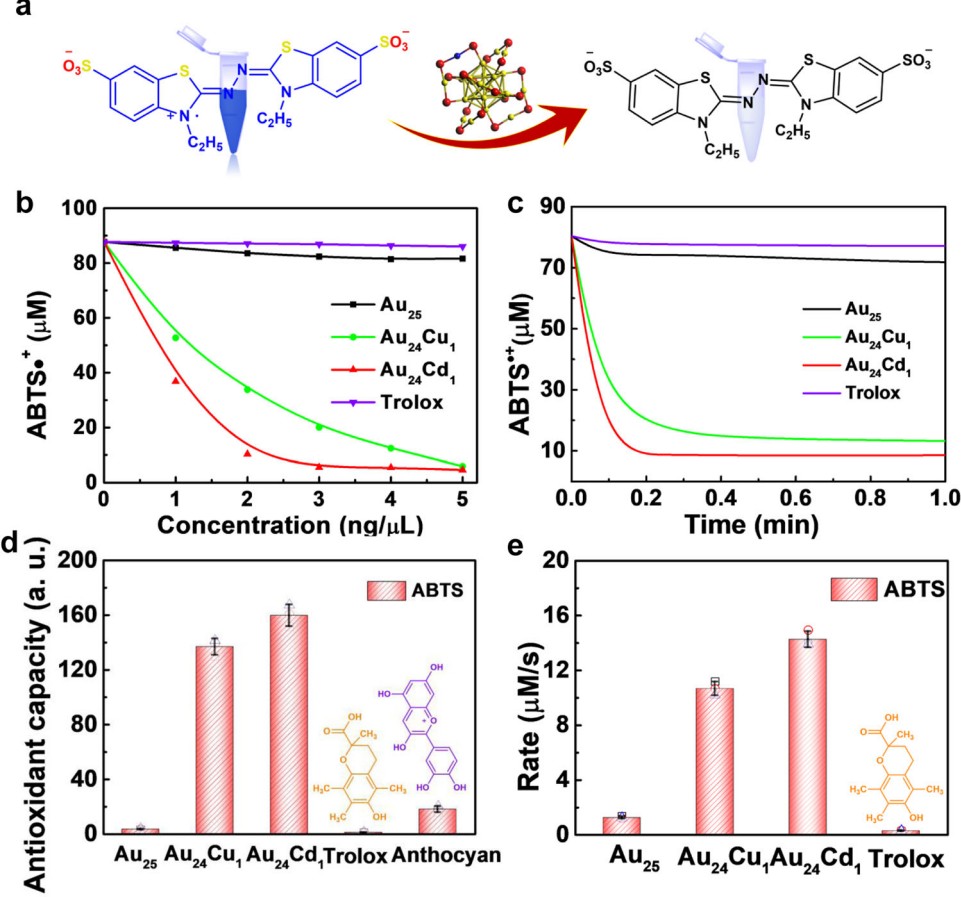

**Fig. 2 The total antioxidant capacity of clusterzymes. a** Schematic illustration of the reaction mechanism of the ABTS assay. **b** Concentration-dependent (0–5 ng/μL) and **c** time-dependent investigation of ABTS$^{\bullet+}$ in the presence of Au$_{25}$, Au$_{24}$Cu$_1$, and Au$_{24}$Cd$_1$. **d** Comparison of the antioxidant capacities and **e** reaction rates of Au$_{25}$, Au$_{24}$Cu$_1$, and Au$_{24}$Cd$_1$ and natural antioxidants show that the antioxidant performance is greatly improved after one atom substitution with Cu or Cd ($n = 3$ independent experiments, data are presented as mean ± SD). Compared with natural antioxidants, Au$_{24}$Cu$_1$ and Au$_{24}$Cd$_1$ show 137 and 160 times higher activity than trolox, and 7.5 and 9 times higher activity than anthocyanin, respectively.

Au$_{25}$, indicative of a weak scavenging efficiency for •OH (Fig. 3f). However, Au$_{24}$Cu$_1$ almost completely diminishes all ESR signals (~100%), consistent with the observed best CAT-like activity as in Fig. 3d. Similarly, the scavenging of O$_2^{\bullet-}$ by clusterzymes was also investigated (Fig. 3g). The ESR signal stays strong for the control and slightly decreases after addition of Au$_{25}$ and Au$_{24}$Cu$_1$, with surplus remaining residues. In contrast, the ESR signal of O$_2^{\bullet-}$ almost disappears in the presence of Au$_{24}$Cd$_1$ further validating its superior specialized SOD-like activity (Fig. 3e). Besides, we also tested the free-radical scavenging capability of the clusterzymes toward reactive nitrogen species (RNS) including •NO, ONOO$^-$, and DPPH•. Au$_{24}$Cd$_1$ shows the most robust overall scavenging efficiency against DPPH• (Supplementary Fig. 12). The ESR reveals that Au$_{24}$Cd$_1$ has the best scavenging capability toward •NO at a low concentration of 2.7 ng/μL, whereas Au$_{25}$ presents ignorable activity (Supplementary Fig. 13). Likewise, both Au$_{24}$Cd$_1$ and Au$_{24}$Cu$_1$ also manifest significantly higher scavenging efficiency toward ONOO$^-$ than Au$_{25}$ (Supplementary Figs. 14 and 15). Au$_{24}$Cd$_1$ is more selective against RNS than Au$_{24}$Cu$_1$, while Au$_{25}$ has insignificant catalytic activity. Thus it is rational to conclude that the high selectivity for enzymes and radicals originates from the single-atom substitutions of Cu and Cd, which induce redistribution of surface electrons and exert influence on electronic structures and states.

**DFT calculations and the mechanism of catalytic selectivity.** To reveal the catalytic mechanism, the density functional theory (DFT) was employed to investigate the catalytic selectivity and quantum properties. By exploring possible structures in the literature, we adopt the gold core of the well-known Au$_{25}$ clusters[60] protected by ligands, which are still connected to the core via S atoms. To evaluate the catalytic behavior and the intermediate states during the chemical reactions, each ligand unit -SCH$_2$CH$_2$COOH is simplified to -SCH$_3$. DFT optimization confirms the stability of the modeled cluster.

Due to the symmetry of the gold cluster, all possible replacements of the guest metallic atoms fall into three categories as follows: oligomer, the surface of core, and core replacements. Figure 4a demonstrates the surface sites of the Au$_{13}$ core and the oligomer site replacement. The oligomer replacement is the common form that is extensively discussed in the literature, but DFT simulations indicate that the surface replacement may be another possibility. However, based on the coordination analysis in the experiment, the oligomer replacement matches the EXAFS results, which yield a significantly lower CN than the surface replacement. With the optimized structure of the clusters, the associated CNs can be theoretically generated even within the clusters involved in the intermediate structures during the catalytic process. The averaged simulated CN values agree with the experimental values that confirm the oligomer replacement

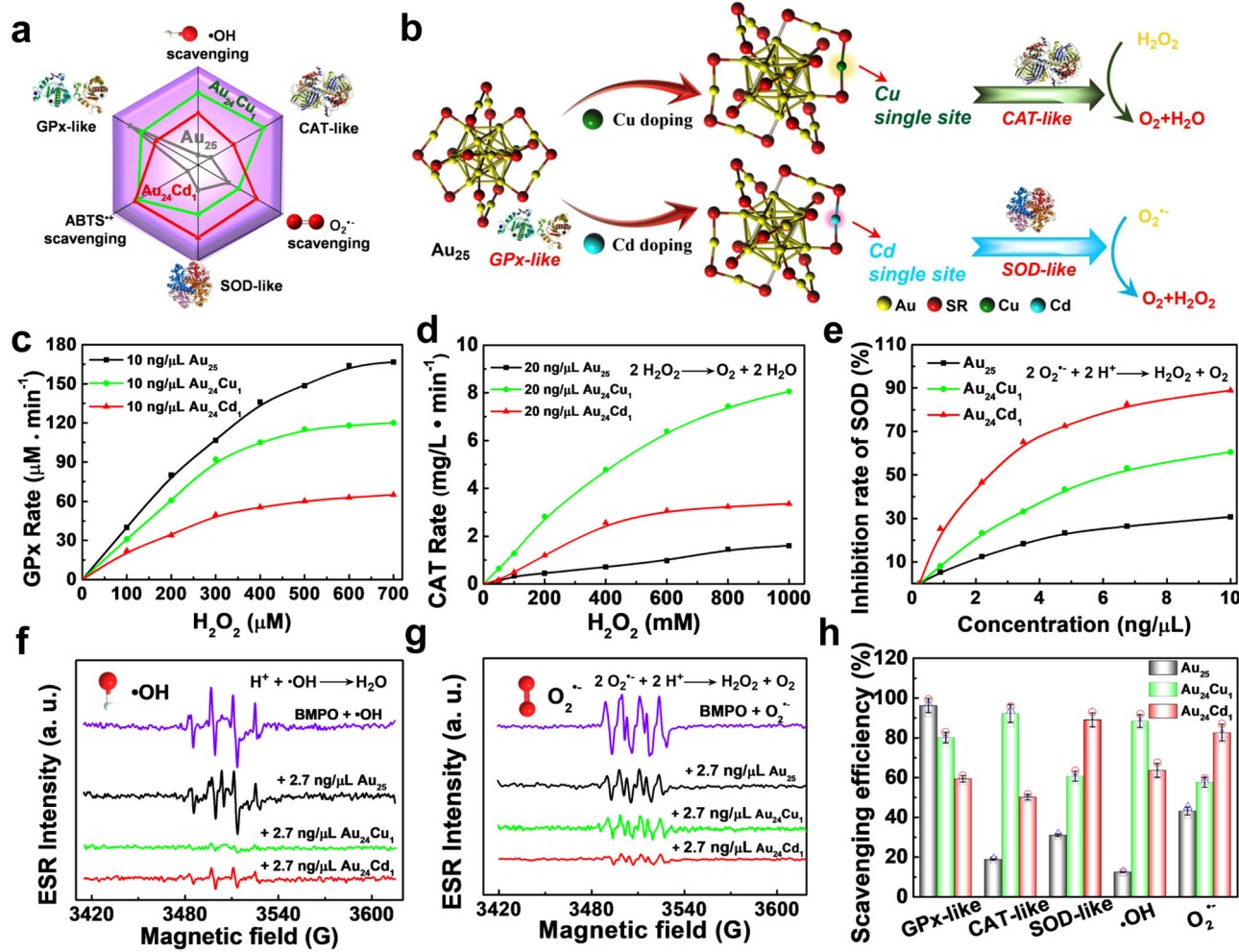

**Fig. 3 Enzyme-mimetic properties and ROS scavenging activity of Au$_{25}$ clusterzymes. a** The radar map of enzymatic activities and free radical scavenging abilities of various clusterzymes. **b** Schematic illustration of catalytic selectivity of the clusterzyme system. Au$_{25}$ exhibits significant superiority in GPx-like activity; Au$_{24}$Cu$_1$ shows advantages in the CAT-like activity through the Cu single active site; Au$_{24}$Cd$_1$ preferably exhibits the SOD-like activity via the Cd single active site, each demonstrating a unique catalytic selectivity. **c** GPx-, **d** CAT-, and **e** SOD-like activities of Au$_{25}$, Au$_{24}$Cu$_1$, and Au$_{24}$Cd$_1$. ROS scavenging activities of Au$_{25}$, Au$_{24}$Cu$_1$, and Au$_{24}$Cd$_1$ clusterzymes for **f** •OH and **g** O$_2$•− studied by the ESR spectroscopy. BMPO is used as the ROS-capturing agent and the sources of •OH and O$_2$•− are H$_2$O$_2$ and KO$_2$, respectively. **h** Corresponding quantifications of the scavenging efficiencies ($n = 3$ independent experiments, data are presented as mean ± SD).

(Supplementary Table 2). Although more theoretical investigations on the surface replacements[61] can be found in the appendix (Supplementary Figs. 16–19 and Supplementary Tables 3 and 4), we focus on the oligomer replacement and the associated catalytic efficiency.

Unlike the surface replacement that may cause the expansion of the core, the oligomer replacement causes the oligomer bending. It is different from the normal S-Au-S chain, which aligns in a (nearly) straight line (Supplementary Fig. 20). The doped Cu shrinks the S-X-S chain while the doped Cd extends it. Compared with the typical bond length of S-Au at 2.3 Å, S-Cu and S-Cd bonds are 2.2 and 2.55 Å, respectively, as shown in Fig. 4b and Supplementary Fig. 21. With the bent chain, the distances between Cu/Cd atoms to the surface of the core are comparable, around 3.1 Å. The similarity between the Cu and Au atoms guarantees that the binding of S-Cu-S is so "firm" that the relative positions of Cu to S atoms can be hardly changed by the dynamics during the catalytic procedures, which are discussed extensively below. In contrast, the relative position of the doped Cd atom may be significantly affected by the local environment

such as the adhesion of small chemical units (Supplementary Figs. 22 and 23).

We observed the excellent performance of the clusterzymes in both CAT and SOD reactions with the reaction pathways summarized in Fig. 4c. The CAT reaction usually refers to the catalytic degradation of hydrogen peroxide, and the decomposition mechanism of H$_2$O$_2$ may involve multiple chemical stages (Fig. 4d, e). For the process of SOD, we assumed that the clusters were involved in similar mechanisms to the general catalytic scheme of SOD reaction. It is worth noting that the release of oxygen completes the CAT process, while the SOD process occurs simultaneously, and the two processes are mutually permeated. The reduced cluster, Cluster(I), may also be involved in both CAT and SOD processes, which depends on the concentration of different components.

Inspired by the Arrhenius equation, we performed the search of transition states and ground states of various types of molecules and ions to estimate the activation energies and evaluated the catalytic efficiencies. The energy profiles in Fig. 4f, g agree with the behaviors of the clusterzymes in our experiments.

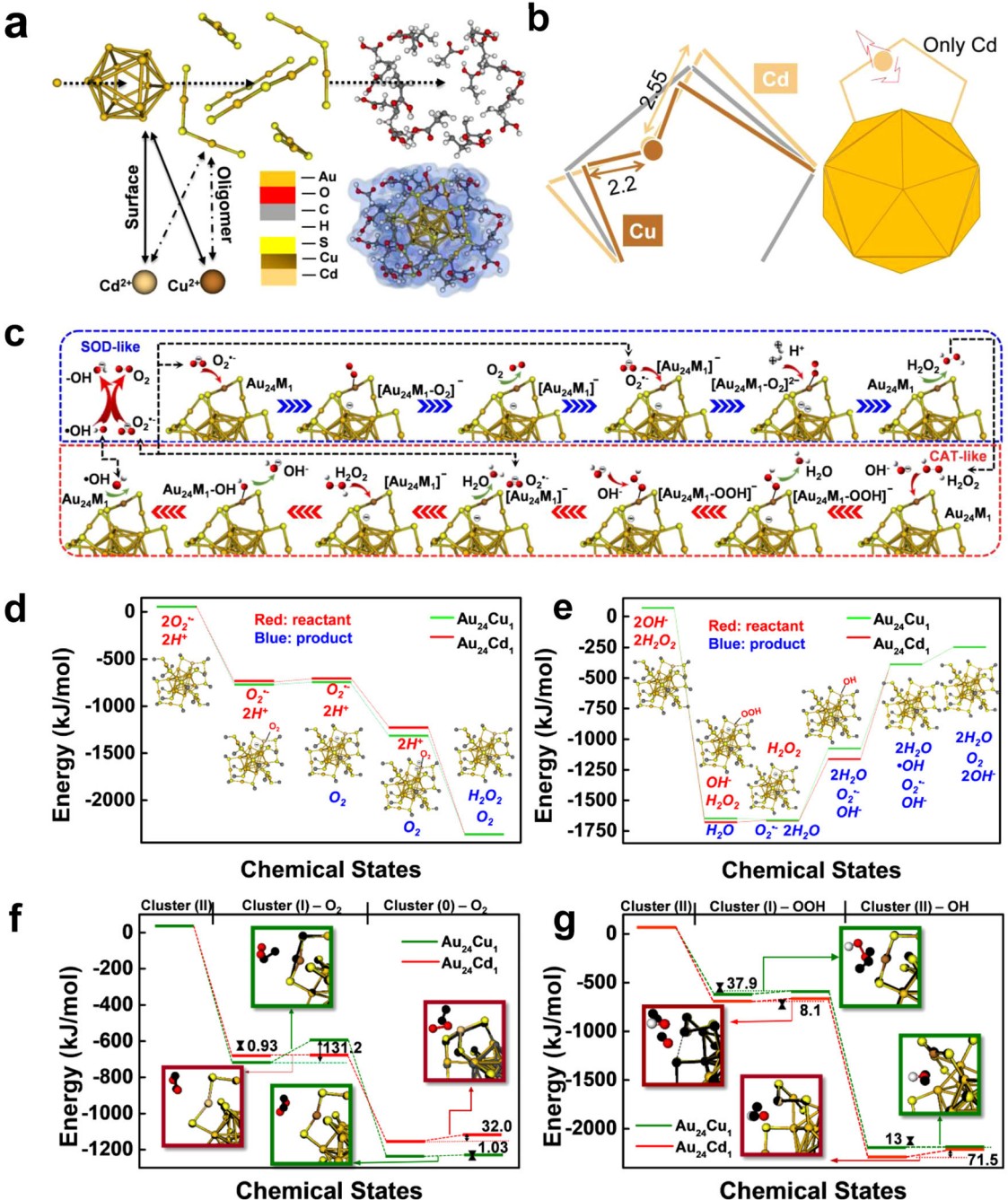

**Fig. 4 DFT calculations and the mechanism of catalytic selectivity. a** Demonstration of atomic doping: surface and oligomer replacement. **b** Doped atom caused change in bonds: the more rigid Cu-S bond and the more flexible Cd-S bond. The flexibility of Cd allows the angular motion. **c** Mechanism and **d**, **e** energies profiles of catalytic process of the SOD and CAT processes. The black dotted line indicates that the catalytic products may be connected to other processes. Energy profiles and geometry structures of the intermediate states of **f** SOD and **g** CAT process in the lower panel. Align optimized intermediate structures (normal color) and transition structures (black).

In a series of reactions with multiple steps, the reaction rate is dominated by the slowest step, i.e., the transition with the largest activation energy. Such a feature can be seen in the first part of the catalytic SOD process by $Au_{24}Cu_1$. The ground state of the electrons in the corresponding intermediate structure is a triplet state suggested by DFT simulations. It indicates that the high activation energy of 131.2 kJ/mol is related to the spin matching issues, which can be selective to the spin of superoxide ions. In the CAT processes, the critical step is related to the decomposition of (cluster…OOH)$^{2+}$, in which the $Au_{24}Cd_1$ exhibits higher activation energy (71.3 kJ/mol) that reduces the efficiency. The

simulations clearly explain the SOD–CAT-selective behaviors of the doped clusterzymes. The details of the reaction pathways are provided in "Methods."

Our results of DFT simulations show some insights of the catalytic mechanisms. Assume in typical clusters, a substituted atom may turn into an active site itself to be involved in the catalytic process, which may be accompanied by changes in the geometry. Herein, we named two mechanisms as SA (simple adhesion) and MA (bond modulated adhesion) correspondingly. The distances between the adsorbed molecule/ion and metal atoms designate the roles of the substituted atom.

The SA mechanism is mainly seen in $Au_{24}Cu_1$. Due to the firmness of the S-Cu-S oligomer, the Cu atom is relatively rigid (Fig. 4f, g). The SA mechanism is also seen in the first step of SOD process catalyzed by $Au_{24}Cd_1$. The catalytic process includes the distance change and orientation change of small units. Significant changes in the distance between the active site (doped atom) and the small units are seen in most of the SOD processes. In contrast, the orientation change is the main character in most of the CAT processes.

The MA mechanism is seen in $Au_{24}Cd_1$ in the second stage (Cluster(I)) of the SOD processes and most of the CAT processes. The bond modulation refers to the position change of the Cd atom, which may deviate from the oligomer plane until a third S atom from another oligomer stops it. Thus the S-Cd bonds are changed significantly (Fig. 4f, g). The characteristics of transition and intermediate states involve the rotation of the superoxide ion (or oxygen molecule) and the position adjustment of the doped Cd atom. To be more precise, the motion of the Cd atom is along the perpendicular direction of the oligomer plane. Once a small unit joins the doped cluster to form an intermediate structure, the Cd atom sometimes leaves the oligomer plane. Therefore, the CN value of the Cd atom is larger when the Cd atom becomes the neighbor of three S atoms. When the Cd atom starts from its original state (S-Cd = 2.55 and 2.55 Å), passes its transition state (S-Cd = 2.63 and 2.85 Å), and arrives at the intermediate state (S-Cd = 2.57 and 2.60 Å), the angular motion is terminated (Supplementary Fig. 24). During such a process, the distance between the attached oxygen atoms is slightly expanded toward the normal distance of oxygen molecules, which indicates the completion of the entire catalytic procedure (Supplementary Tables 5 and 6). A similar procedure for the $Au_{24}Cd_1$ can be observed at the adhesion of OOH- at the first stage of CAT process. Such a unique process allows the doped Cd atom to be an active site that can be self-modulated in a wide spatial range compared to the firm Cu atom. This may explain its good performance in the SOD process.

**Modulation of neuroinflammation**. To reveal the biological activity of clusterzymes, the cell toxicity for different nerve cell lines (HT22, BV2, and MA-c) were measured by the 3-[4,5-dimethylthiazol-2-yl]-2,5 diphenyl tetrazolium bromide (MTT) assay (Fig. 5a and Supplementary Fig. 25), showing that $Au_{25}$, $Au_{24}Cu_1$, and $Au_{24}Cd_1$ present acceptable biocompatibility. Cell survival of $H_2O_2$-stimulated neuron cells was performed with the incubation of $Au_{25}$, $Au_{24}Cd_1$, or $Au_{24}Cu_1$. As shown in Fig. 5b, the clusterzyme treatment could improve the viability of neuron cells. To explore the correlation between the oxidative stress and the neuron viability, reactive oxygen species (ROS), especially •OH and $O_2^{•-}$, were quantified and detected by a FACS flow cytometer and a fluorescence microscope using hydroxyphenyl fluorescein (HPF) and dihydroethidium (DHE) fluorescence probes, respectively. The $H_2O_2$ stimulation significantly elevates the fluorescence signal, indicating the presence of excessive amount of •OH and $O_2^{•-}$ (Fig. 5c–h). All clusterzymes decrease the ROS signals, with $Au_{24}Cu_1$ showing the best clearance efficiency against •OH (Fig. 5c, e, g) and $Au_{24}Cd_1$ displaying the best clearance capability for $O_2^{•-}$, suggesting their individual selectivity (Fig. 5d, f, h). Meanwhile, mouse models of traumatic brain injury (TBI) were used to examine the in vivo effects of clusterzymes. As shown in Fig. 5i–l, the indicators of malondialdehyde (MDA), $H_2O_2$, SOD, and GSH/GSSG in the TBI group are relatively severe at day 1 post injury but are slightly alleviated 3 days post injury and are further improved slightly 7 days post injury. Therefore, the decrease in SOD and GSH/GSSG levels from TBI can be well rescued by clusterzymes with

prominent recoveries 7 days after treatment (Fig. 5i, j). Comparatively, $Au_{24}Cd_1$ induce a better recovery in SOD than $Au_{24}Cu_1$, which correlates well with their in vitro SOD-like activity (Fig. 3). As the by-products of the oxidative stress, lipid peroxides and $H_2O_2$ show higher accumulations in the brain following TBI, resulting in severe oxidative damage (Fig. 5k, l). Both $Au_{24}Cu_1$ and $Au_{24}Cd_1$ significantly inhibit the production of these harmful molecules, while $Au_{25}$ barely alters the TBI-induced increase. These results are conceivable because $O_2^{•-}$ is known to be continuously produced by immediate injuries at the early stage, followed with subsequent production of lipid peroxides and $H_2O_2$. With regard to $Au_{24}Cd_1$, it can recover the diminished SOD in the first place due to its high catalytic selectivity for $O_2^{•-}$ and then sustain the continuous decrease of lipid peroxides and $H_2O_2$ as the secondary catalytic options. In contrast, $Au_{24}Cu_1$ is primarily prone to increase the levels of lipid peroxides and $H_2O_2$ at the early stage due to its preference for CAT-like activity and •OH, but these molecules are intermediate products at relatively low concentrations after TBI onset, and consequently it accounts for the increasing clearance capability in the long term.

Finally, the effects of clusterzymes on neuroinflammation were examined. From the western blots and the relevant quantification analysis (Fig. 6a, d), the expression levels of interleukin (IL)-1β and IL-6 are significantly upregulated following TBI 1 day post injury, indicative of strong local inflammations. $Au_{24}Cd_1$ sharply downregulates IL-1β and IL-6 levels, suggesting the anti-neuroinflammation effect. In comparison, $Au_{25}$ only shows minor downregulation. Similarly, TBI leads to significant upregulation of tumor necrosis factor-α (TNFα) 1 day post injury, but $Au_{24}Cu_1$ can significantly downregulate the expression of TNFα, presenting superior efficacy over $Au_{25}$ and $Au_{24}Cd_1$ (Fig. 6a). Although the expression levels of these inflammation cytokines induced by TBI are gradually suppressed by auto-immunity in the vehicle control group over time, especially 7 days post injury, there are still significant differences in IL-1β and IL-6 levels between the Sham and TBI groups (Fig. 6b–d). However, the clusterzyme treatment results in cytokines close to the normal level, indicating a better suppression effect on neuroinflammation. Enzyme-linked immunosorbent assay (ELISA) further validated the immunoblotting results that $Au_{24}Cd_1$ and $Au_{24}Cu_1$ are capable of decreasing the inflammatory cytokines in brain tissues such as IL-1β, IL-6, and TNFα, while $Au_{25}$ does not significantly alter the inflammatory cytokine patterns (Fig. 6e–g). $Au_{24}Cd_1$ can eliminate IL-1β- and IL-6-associated inflammatory responses, while $Au_{24}Cu_1$ has a better effect on reduction of TNFα, indicating their relevant selectivity toward modulation of neuroinflammation. Finally, immunofluorescence staining of cerebral cortex harvested from mice also shows that $Au_{24}Cd_1$ and $Au_{24}Cu_1$ can remarkably decrease the TBI-elevated expression levels of IL-1β, IL-6, and TNFα (Fig. 6h, i and Supplementary Figs. 26 and 27), therefore alleviating neuroinflammation. Colocalization studies with markers for neurons (NeuN), microglia (Iba-1), or astrocytes (glial fibrillary acidic protein (GFAP)) were performed in injured cortex on day 3 post injury. Figure 6h, i reveal that IL-1β is mainly produced by microglia after TBI, similar with IL-6 and TNFα (Supplementary Figs. 26 and 27). In addition, quantitative analyses of the number of positive cells show that massive microglia and astrocytes are activated, and many neurons are depleted after TBI (Fig. 6j). With the clusterzyme treatment, most of these nerve cells are rescued. Meanwhile, the morphology of TBI-activated astrocytes can be recovered to near normal levels after treatment with clusterzymes (Supplementary Fig. 28), and the neuroinflammatory responses are also prevented likewise by verified histology (Supplementary Figs. 29–32). The clusterzymes can also restore the TBI-induced

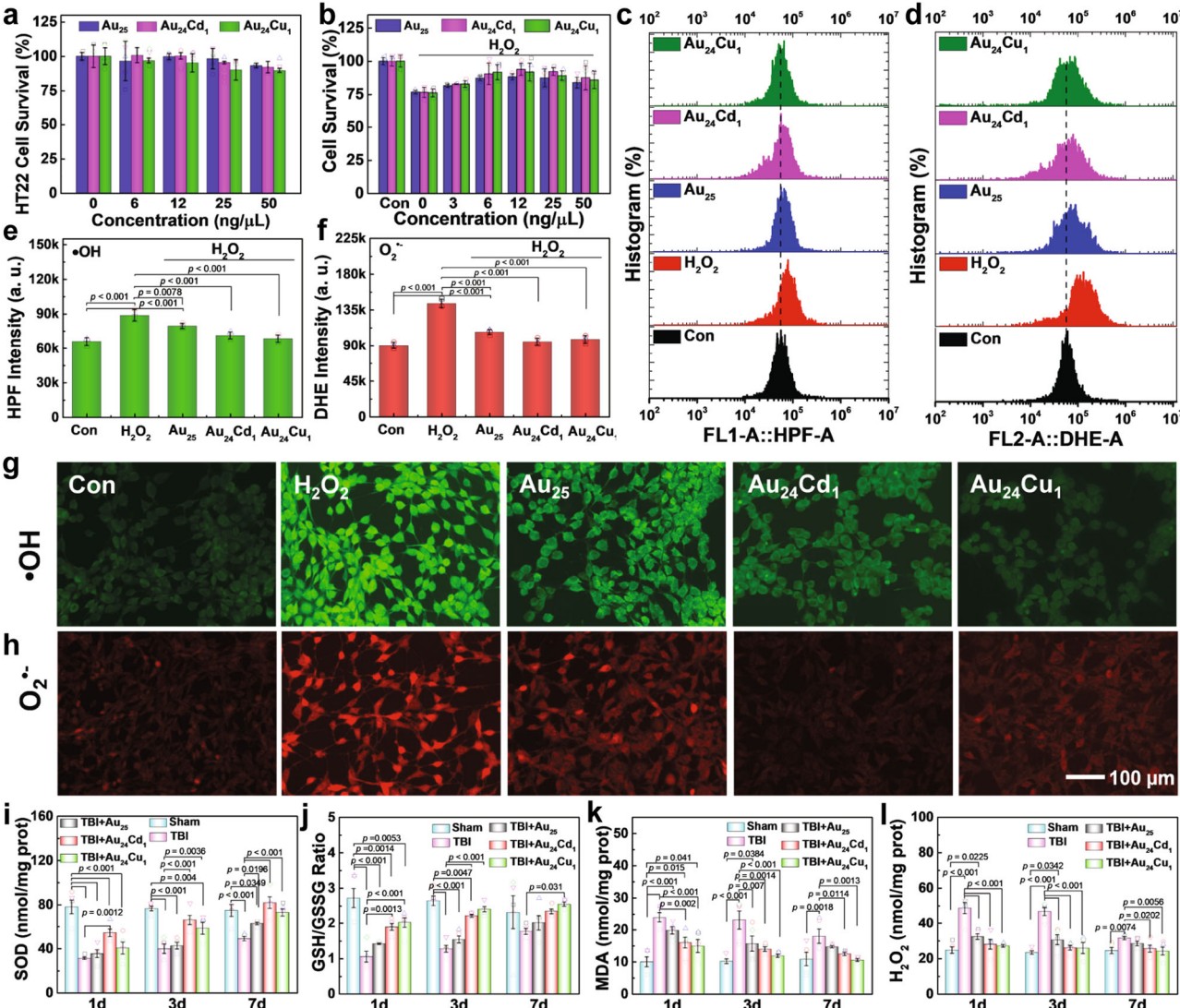

**Fig. 5 Oxidative stress levels in vitro and in vivo before and after treatment of clusterzymes. a** HT22 cell viability of clusterzymes ($n = 5$ per group, data are presented as mean ± SD). **b** HT22 cell viability in the presence of $H_2O_2$ with or without treatment of clusterzymes as determined by MTT assays ($n = 6$ per group, data are presented as mean ± SD). Fluorescence quantification of cell staining for **c**, **e** •OH and **d**, **f** $O_2^{•-}$ by flow cytometry ($n = 3$ per group). Data are presented as mean ± SD and compared with the Con and $H_2O_2$ groups, analyzed by one-way ANOVA with two-sided LSD test (adjusted $p$ values are shown). Fluorescence microscopic images of intracellular **g** •OH (green) and **h** $O_2^{•-}$ (red) levels induced by 100 μM $H_2O_2$ with or without clusterzymes treatment, stained by HPF and DHE probes, respectively. It can be seen that $Au_{24}Cu_1$ has a better scavenging ability for •OH and $Au_{24}Cd_1$ shows better specificity for $O_2^{•-}$, suggesting their individual selectivity for •OH and $O_2^{•-}$ respectively. **i–l** Indicators for oxidative stress, including SOD, GSH/GSSG, MDA, and $H_2O_2$, of TBI mice with or without treatment of clusterzymes 1, 3, and 7 days post injury ($n = 5$ per group). Data are presented as mean ± SEM and compared with the Sham and TBI groups, analyzed by one-way ANOVA with two-sided LSD test (adjusted $p$ values are shown). Experiments were repeated independently **g**, **h** three times with similar results.

body weight loss (Supplementary Fig. 33). Moreover, behavioral tests were studied by the Morris water maze (MWM). As shown in Supplementary Fig. 34a, b, all the mice apparently learned the task during the acquisition phase of days 13–17 and 28–31, while the distance traveled and latency to hidden platform with $Au_{24}Cu_1$ and $Au_{24}Cd_1$ treatment obviously decreased. For the probe trial on days 18 and 32 (Supplementary Fig. 34c, d), the percentage time in the missing platform quadrant and the number of platform crossings were significantly reduced in the TBI group but almost return to the normal level after $Au_{24}Cu_1$ and $Au_{24}Cd_1$ treatment. These results reveal trends in the improvements of learning ability and spatial memory with $Au_{24}Cu_1$ and $Au_{24}Cd_1$ treatment. In addition, we systematically studied the pharmacokinetics and toxicology of clusterzymes. It

can be seen that the clusterzymes accumulated in major organs can be removed by the kidney (urine) and liver (feces). After 48 h, ~80% of the total dose can be excreted, and most of it is excreted through the kidney (>70%) (Supplementary Fig. 35). No significant changes in organs or blood chemistry or hematology are found, suggesting that renal-clearable clusterzymes do not cause significant biological toxicity in vivo (Supplementary Figs. 36–38). Artificial enzymes have persistently been shown to exhibit multiple enzyme-like catalytic activities with a diversified class of materials[15]. Low catalytic activity as compared to natural enzymes, however, is one of the most noticeable disadvantages due to limited electron transfers at atomic levels[15]. The rationally designed clusterzymes with single-atom substitutions overcome such barriers with antioxidant activity nine times higher than that

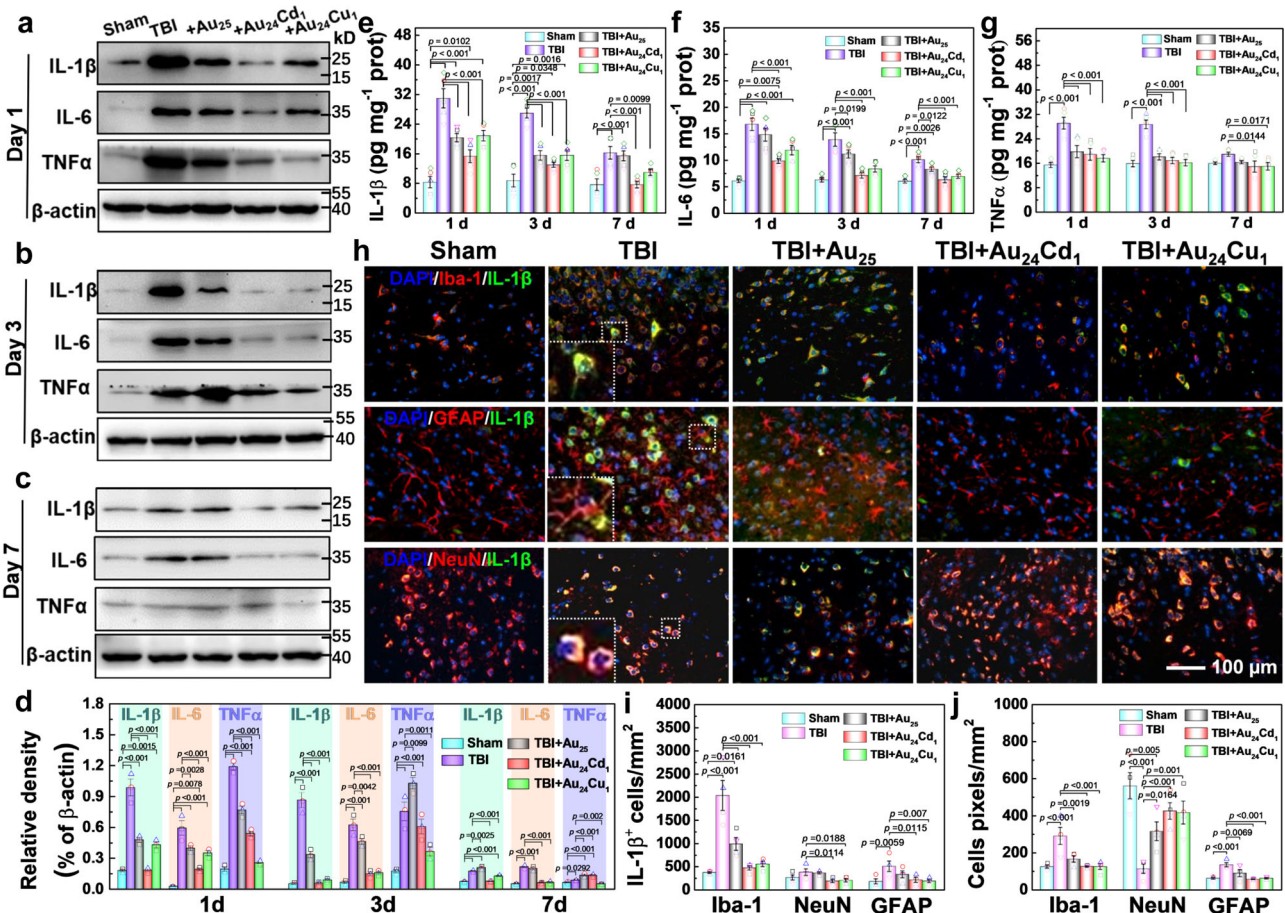

**Fig. 6 Inflammation levels in brain tissues after clusterzyme treatment. a–c** Western blotting for IL-1β, IL-6, and TNFα in the brain tissues 1, 3, and 7 days post TBI after treatment ($n = 3$ per group), respectively. **d** Western blotting quantitative analysis of inflammatory factors at different time points ($n = 3$ per group). All the samples were derived from the same experiment and blots were processed in parallel. Data are presented as mean ± SEM and compared with the Sham and TBI groups, analyzed by one-way ANOVA with two-sided LSD test (adjusted $p$ values are shown). It can be seen that Au$_{24}$Cd$_1$ can rapidly and significantly reduce the upregulated inflammatory cytokines of IL-1β and IL-6 after brain injury, while Au$_{24}$Cu$_1$ has a better ability to reduce the expression of TNFα. **e–g** ELISA quantitative analysis of IL-1β, IL-6, and TNFα levels in brain tissues on days 1, 3, and 7 with or without clusterzymes treatment ($n = 5$ per group), respectively. Data are presented as mean ± SEM and compared with the Sham and TBI groups, analyzed by one-way ANOVA with two-sided LSD test (adjusted $p$ values are shown). **h** Immunofluorescence co-staining of IL-1β and microglia (Iba-1), astrocytes (GFAP), or neurons (NeuN) in injured cortex 3 days post injury with or without clusterzyme treatment. Quantitative analysis of **i** the number of IL-1β$^+$ expression in different positive cells and **j** the pixels density of Iba-1/NeuN/GFAP cells in the injured cortex with or without clusterzyme treatment ($n = 3$ per group). Data are presented as mean ± SEM and compared with the Sham and TBI groups, analyzed by one-way ANOVA with two-sided LSD test (adjusted $p$ values are shown). Experiments were repeated independently **a–c** twice and **h** three times with similar results.

of anthocyanin, which is known to be one of the most reactive antioxidant molecules in nature. Besides, unlike the structurally ambiguous traditional artificial enzymes, the definitive molecular structures of clusterzymes are accurately elucidated, allowing us to distinguish the catalytically active sites and scrutinize the electronic structures and reaction energies[62–65]. As a result, the substituting single atoms can be arranged into a specific spatial location of the clusterzyme freely, thus tuning electronic structures and affecting the catalytic activity[66–70]. Meanwhile, the interactions between host atoms (i.e., Au) and the introduced substituting atoms (i.e., Cu or Cd) can induce coupled electron states and in turn influence the catalytic selectivity[71]. In our work, the GPx-, SOD-, and CAT-like catalytic selectivity were assigned to Au$_{25}$, Au$_{24}$Cd$_1$, and Au$_{24}$Cu$_1$, respectively, via modulated bond lengths to the active center, and thus it is conceived that such a platform of clusterzymes will generate various selectivity against different molecules. By employing the three catalytically selective clusterzymes, we successfully established the relationship between

oxidative stress and neuroinflammation, demonstrating the importance of O$_2^{\bullet-}$ and long-term benefits in TBI. Specifically, Au$_{24}$Cd$_1$ can significantly mitigate the neuroinflammation via inhibiting IL-1β and IL-6[29,72], while Au$_{24}$Cu$_1$ differentially reduces neuroinflammation by inhibiting TNFα, showing selectivity against anti-neuroinflammation. Meanwhile, due to the innate ultrasmall size of clusterzymes, it can penetrate the kidney barriers and be excreted by renal, avoiding long-term hepatotoxicity and multi-organ injuries. Therefore, the clusterzymes are presumably influential as a biomedicine, especially in the field of neuroscience.

## Discussion

In summary, we report a systemic single-atom substitution approach to fabricate artificial enzymes on the basis of MPA-protected Au$_{25}$ clusters, namely clusterzymes. The clusterzymes show the ultrahigh antioxidant activity up to 137–160 times

higher than the natural trolox. Moreover, the catalytic selectivity toward GPx, CAT, SOD, and nitrogen-related signaling molecules can be fine-tuned by single-atom substitutions. DFT calculations conclude that reaction pathways are modulated by the single active site of $Au_{24}Cd_1$ and $Au_{24}Cu_1$ at bond lengths. The biological results show that $Au_{24}Cd_1$ preferentially decreases IL-1β and IL-6, while $Au_{24}Cu_1$ tends to decrease TNFα, indicative of their different selectivity for modulating alleviation of neuroinflammation.

## Methods

**Materials**. All chemicals are commercially available with the highest purity and used without further treatment. Gold chloride ($HAuCl_4·3H_2O$) was purchased from Sigma-Aldrich; sodium hydroxide (NaOH), sodium borohydride ($NaBH_4$), copper nitrate ($Cu(NO_3)_2$), cadmium nitrate ($Cd(NO_3)_2$), and MPA were purchased from Aladdin. Ultrapure water (18.2 MΩ·cm) was used for all the experiments.

**Materials' preparation**. The gold nanoclusters were synthesized according to the previous literature[73]. In detail, aqueous solutions of $HAuCl_4$ (20 mM, 0.25 mL) and MPA (5 mM, 2 mL) were added to water (2.35 mL) and stirred at room temperature for 5 min. Then an aqueous NaOH solution (1 M, 0.3 mL) was added to the reaction solution, followed by the addition of 0.1 mL of an $NaBH_4$ solution (prepared by dissolving 43 mg of $NaBH_4$ powder in 10 mL of 0.2 M NaOH solution). The whole reaction was carried out in the dark, and $Au_{25}MPA_{18}$ was collected after stirring at room temperature for 3 h, and the final reaction solution aged at 4 °C for 12 h. The syntheses of various metal-substituted $Au_xM_{25-x}SG_{18}$ were also based on the same method. The only difference was that the Au atoms in $HAuCl_4$ (20 mM, 0.25 mL) were replaced by various nitrate metal ions ($Cu^{2+}$, $Cd^{2+}$) at a 4% molar ratio (Au:M = 24:1). For further purification of nanoclusters, we used ultrafiltration tubes of 3 and 10 K at 3500 rpm/min for ultrafiltration to remove smaller organic ligands and larger-sized clusters and lyophilized to obtain the purified product for further testing and application.

**Materials' characterization**. Ultraviolet–visible (UV-vis) absorption spectra were recorded on Shimadzu 3600 UV-vis-NIR spectrophotometer. ESI-MS were acquired on Bruker microTOF-Q system. XANES along with EXAFS analyses were tested and provided by Beijing Synchrotron Radiation Facility. The module ARTEMIS of programs of IFEFFIT were used for processing data of XANES and EXAFS[74,75]. Clusterzymes were pressed into pellets, and Au $L_3$-edge, Cu $K$-edge, and Cd $L_3$-edge were measured at room temperature. The Au/Cu/Cd foil was also measured for comparison. XPS of the metal element was performed on a K-Alpha spectrometer with a monochromatic Al Kα X-ray source operating at 300 W (ThermoFisher Scientific). The C 1s level of 284.8 eV was used as an internal standard to correct for peak drift, and the XPS peaks were fitted using the XPSPEAK41 software. Fourier transformed infrared spectra of various clusterzymes before and after the reactions were recorded on AVATR360 Spectrometer (Thermo Nicolet, US). The scan wavenumber ranges from 400 to 3800 $cm^{-1}$, and the samples were determined by powder method. Raman spectrum was performed on INVIA Reflex spectrometer (Renishaw, UK) excited by a 633 nm He-Ne laser. A Malvern Zetasizer nano ZS90 (UK) was employed for measuring DLS to test the hydrodynamic size and zeta potential of clusterzyme. ICP-MS was tested with 7900 ICPMS (Agilent, US) to determine the content of metallic elements in clusterzymes. The scavenging process of •OH, •NO, and $O_2^{•-}$ was determined by ESR spectrometer (Bruker EMX plus, Germany). The kinetic test of CAT-like was performed by Dissolved Oxygen Meter (HACH HQ40d, US) with LDO101 probe.

### Antioxidant and free radical scavenging tests

*Total antioxidant capacity test (ABTS rapid method)*. The total antioxidant capacity (T-AOC) of clusterzyme and contrast (trolox and anthocyanin) was determined by the rapid ABTS method using the T-AOC Assay Kit (S0121, Beyotime). Please refer to the specification for specific sampling methods. The antioxidant capacity was evaluated by measuring the absorption value at 414 nm. In the process of reaction kinetic analysis, we adjusted the concentration of different concentration of $ABTS^{•+}$ (the molar extinction coefficient of $ABTS^{•+}$: $\varepsilon_{414\ nm} = 3.6 \times 10^4$ $mol^{-1}$ $cm^{-1}$) by changing the concentration of $H_2O_2$. The reaction kinetic analysis process was reflected by the change of absorbance at 414 nm monitored by the UV-vis spectrophotometer under kinetic mode. The steady-state kinetic parameters were determined by varying the concentration of $ABTS^{•+}$ in the presence of clusterzymes (5 ng/μL). The maximum reaction velocity ($V_{max}$) and Michaelis–Menten constant ($K_m$) were calculated using the Lineweaver–Burk equation.

*RNS scavenging test*. The RNS scavenging capacity of clusterzyme was performed for 1,1-diphenyl-2-picrylhydrazyl radical (DPPH•). The scavenging capacity of free radicals was evaluated by measuring the absorption wavelength at 510 nm. Briefly, 50 μM DPPH• and 5 ng/μL clusterzyme were dissolved in a mixture of dimethyl

sulfoxide and water (1:40). The changes of absorption spectra with time in the range of 300–1000 nm were determined.

*$ONOO^-$ scavenging test*. The preparation of $ONOO^-$ was reported by methods in the literature[4]. Specifically, the aqueous solution of $NaNO_2$ (5 mL, 50 mM) and $H_2O_2$ (5 mL, 50 mM) was rapidly stirred and mixed in an ice bath. Then HCl (2.5 mL, 1 M) and NaOH (2.5 mL, 1.5 mM) were quickly added and continued to stir for 5 min to get the yellowish $ONOO^-$ reserve solution. The $ONOO^-$ reserve was diluted to about 1.3 mM ($\varepsilon_{302\ nm} = 1670 \pm 50$ $M^{-1}$ $cm^{-1}$), and 5 ng/μL of clusterzyme was added. The scavenging capacity of $ONOO^-$ was evaluated by measuring the change of absorption spectra in the range of 250–400 nm over time.

*•OH scavenging test*. First, a 5 mM $H_2O_2$ solution was prepared with a 10 mM phosphate-buffered saline (PBS) buffer, and then the hydroxyl radical was generated by UV-laser irradiation for 5 min. We used an ESR spectrometer (Bruker EMX plus, Germany) to determine the scavenging process of hydroxyl radicals. BMPO (50 mM) was used as the capturing agent for hydroxyl radical, and the spin adduct (BMPO/•OH) generated with •OH presented four peaks under ESR spectrometer. The removal process of •OH was determined by testing the change of its peak strength before and after the addition of clusterzymes (2.7 ng/μL).

*$O_2^{•-}$ scavenging test*. $KO_2$ of 2.5 mM and 18-crown-6 of 3.5 mM were used as the generation source and stabilizer of $O_2^{•-}$. The 25 mM 5-(diethoxyphosphoryl)−5-methyl-1-pyrroline-*N*-oxide (DEPMPO) was used as a spin capturing agent, and its spin adduct (DEPMPO/$O_2^{•-}$) presented six peaks under ESR spectrometer. Monitoring the signal intensity change of peaks before and after the addition of 2.7 ng/μL clusterzyme could verify the scavenging capacity to $O_2^{•-}$.

*•NO scavenging test*. Carboxy-PTIO of 10 μM with 5 peaks under ESR spectrometer was used as the capturing agent, and *S*-nitroso-*N*-acetylpenicillamine of 250 μM was used as the •NO contributor. The spin adduct (Carboxy-PTI) generated presented seven peaks under ESR spectrometer. The scavenging capacity for •NO was evaluated by monitoring the changes of peaks intensity and types before and after the addition of 2.7 ng/μL clusterzymes.

### Enzyme-like activity test

*CAT-like test*. CAT-like activity of clusterzymes was determined by two methods. First, according to the unique absorption peak of $H_2O_2$ at 240 nm, the optical density decreases with the decomposition of $H_2O_2$, and the extinction coefficient of $H_2O_2$ (43.6 $mM^{-1}$ $cm^{-1}$ at 240 nm) was used to calculate its activity. The reaction solutions contained 53 μM $H_2O_2$ and 10 ng/μL of clusterzyme in 200 μL PBS. $H_2O_2$ 10 M was treated with or without 50 ng/μL clusterzymes for 30 min to obtain the photos of $H_2O_2$ decomposition in the centrifuge tube. Another method was to measure the kinetics of the CAT-like using the Dissolved Oxygen Meter (HACH HQ40d, US) with LDO101 probe. First, the solubility of $O_2$ in solution was reduced to 0.6 mg/L by continuous infusion of Ar into 5 mL PBS solution. Then, different concentrations of $H_2O_2$ (50–1000 μM) and 20 ng/μL of clusterzymes were added to the system to monitor the solubility change of $O_2$ every 10 s. The interference deducted from each set of experiments is the effect of $H_2O_2$ self-decomposition under the same conditions. The maximum reaction velocity ($V_{max}$) and Michaelis–Menten constant ($K_m$) were calculated using the Lineweaver–Burk equation by the Origin 9.0 software.

*GPx-like test*. The GPx-like activity of clusterzymes was determined by the method in the literature. Briefly, 200 μM $H_2O_2$, 2 mM GSH, 200 μM NADPH, 1.7 units/mL GR, and 10 ng/μL clusterzymes were added to 200 μL PBS neutral buffer. The activity of GPx-like was evaluated by monitoring the changes in absorbance at 340 nm. The absorbance at 340 nm represents the concentration of NADPH ($\varepsilon_{340\ nm} = 6.22$ $mM^{-1}$ $cm^{-1}$). GPx-like activity = $(A_{Con.} − A_{Clusterzyme})/A_{Con.} \times 100\%$. The reaction kinetic analysis process was reflected by the change of absorbance at 340 nm monitored by the UV-vis spectrophotometer under kinetic mode. The formation concentration of the substrate was changed by adjusting the concentration of $H_2O_2$ (100–700 μM), and the other conditions remained the same. The maximum reaction velocity ($V_{max}$) and Michaelis–Menten constant ($K_m$) were calculated using the Lineweaver–Burk equation by the Origin 9.0 software.

*SOD-like test*. The SOD-like activity of clusterzymes was tested according to the description in the SOD Activity Assay Kit. After adding clusterzymes (0–10 ng/μL) of different concentrations, the absorbance changes at 560 nm were monitored with UV-vis spectrometer to further evaluate SOD-like activity.

**DFT calculations**. To investigate the catalytic effect, we focus on the energy profile of the clusterzymes in their intermediate structures. The adsorption energy that describes the energy change of an adsorbate when being attracted by a cluster is calculated by the following equation:

$$E_{ads} = E_{cata+mol} - (E_{cata} + E_{mol}). \qquad (1)$$

The $E_{\text{cata}}$ and $E_{\text{mol}}$ denote the energy of the cluster and the adsorbate, respectively, while the $E_{\text{cata+mol}}$ is the energy of the intermediate structure. In addition, we also simulate the activation energies, which are the differences of the energies between the transition states and energetically stable states.

$$E_{\text{act}} = E_{\text{cata+mol}}^{\text{TS}} - E_{\text{cata+mol}}. \tag{2}$$

The transition states, which are used to estimate the activation energy barriers that corresponds to the saddle points on an energy surfaces ($E_{\text{cata+mol}}^{\text{TS}}$), are searched using Berny algorithm.

**The CAT process**. A CAT reaction usually refers to the catalytic degradation of hydrogen peroxide. The total reaction is

$$2H_2O_2 \rightarrow 2H_2O + O_2. \tag{3}$$

In our experiment, we observed outstanding performance of the clusterzymes as the CATs. The mechanism of the decomposition of $H_2O_2$ may involve multiple steps and multiple paths. Considering the catalytical mechanism of copper (II), we propose the clusterzyme initiate the reaction in the following steps:

$$\text{cluster}^{2+} + H_2O_2 \rightarrow (\text{cluster}\cdots\text{OOH})^+ + H^+. \tag{4}$$

Then the intermediate structure, $\text{cluster}\cdots\text{OOH}^+$, decomposes into three pieces: a cation, a superoxide ion, and the original cluster with an extra electron, Cluster (I):

$$(\text{cluster}\cdots\text{OOH})^+ \rightarrow \text{cluster}^+ + O_2^- + H^+, \tag{5}$$

$$\cdot\text{OH} + O_2^- \rightarrow O_2 + \text{OH}^-. \tag{6}$$

The superoxide ion may be involved in a process of hydroxyl scavenging (Eq. 6) or follow another SOD reaction path. The reduced cluster can cause a disproportionation-like process in which the hydrogen peroxide is not equally divided:

$$\text{cluster}^+ + H_2O_2 \rightarrow (\text{cluster}\cdots\text{OH})^{2+} + \text{OH}^-, \tag{7}$$

$$(\text{cluster}\cdots\text{OH})^{2+} \rightarrow \text{cluster}^{2+} + \cdot\text{OH}. \tag{8}$$

The hydroxyl radicals produced (Eq. 8) may be cleaned by the process in Eq. (6). The superoxide ions produced in Eq. (5) are partially involved in the SOD reaction as we shall discuss in the next part.

**The SOD process**. For the process of SOD, we assume that the clusters are involved in similar mechanisms to the general catalytic scheme of SOD reaction [Eq. (6)]:

$$O_2^- + \text{Cluster}^{2+} \rightarrow O_2 + \text{Cluster}^+, \tag{9}$$

$$O_2^- + 2H^+ + \text{Cluster}^+ \rightarrow H_2O_2 + \text{Cluster}^{2+}. \tag{10}$$

Equation (9) shows the release of oxygen, which complete the CAT process, and it is also the initialization of SOD process. The reduced cluster, Cluster (I), may also be involved in both CAT and SOD process, which depends on the concentration of different components. Equations (9) and (10) are combined to the total reaction:

$$2O_2^- + 2H^+ \rightarrow H_2O_2 + O_2. \tag{11}$$

Please note that Eq. (10) is not the end of the reaction, but the end of SOD. The hydrogen peroxide can be decomposed by the CAT process mentioned above. Furthermore, the superoxide ions may react with the hydrogen peroxide as the following reaction:

$$O_2^- + H_2O_2 \rightarrow \cdot\text{OH} + \text{OH}^- + O_2. \tag{12}$$

The hydroxyl radicals are produced in such reaction, but the reaction is slow to be biologically significant. So in this work, we do not consider such reaction and the production of hydroxyl radicals.

**In vitro experiments**. Mouse hippocampal neuronal HT22 cells were obtained from the Institute of Radiation Medicine, Chinese Academy of Medical Sciences, and Peking Union Medical College and employed in all the cellular experiments. Mouse microglia BV2 cells and mouse astrocytes-cerebellar MA-c cells were obtained from Tianjin Huanhu Hospital. Cells were cultured in Dulbecco's Modified Eagle Medium (DMEM) (Gibco), supplemented with 10% fetal bovine serum (FBS, BI) at 37 °C with 5% $CO_2$. In all, 100 U/mL penicillin and 100 mg/mL streptomycin sulfate (Solomo) were applied according to the growth state.

**Cytotoxicity assay**. HT22 cells ($2 \times 10^3$), BV2 cells ($3 \times 10^3$), and MA-c cells ($4 \times 10^3$) were seeded in 96-well plates filled with 0.01 M PBS (Gibco) at the border in 100 μL medium overnight. The culture medium was replaced by different doses of Au$_{25}$, Au$_{24}$Cu$_1$, or Au$_{24}$Cd$_1$ dissolved in the DMEM, and then cells were incubated for another 24 h. Wells were washed with 0.01 M PBS once, and the medium were replaced by fresh culture medium with serum-free DMEM. Cell cytotoxicity was determined by SPSS 19 MTT assay at the MTT concentration of 5 mg/mL for 2.5 h and detected at optical density (OD) 490 nm.

**Cell viability**. HT22 cells ($2 \times 10^3$) were cultured in the 96-well plate in 100 μL culture media. When reaching roughly 60% confluence in each well, cells were stimulated by 100 μM $H_2O_2$ for 6 h. Then the culture media were substituted by fresh media containing Au$_{25}$, Au$_{24}$Cu$_1$, or Au$_{24}$Cd$_1$ at different doses, and cells were incubated overnight. The plates were washed with PBS and incubated with 5 mg/mL MTT for 2.5 h. Cell viability was determined by MTT assay and analyzed at OD 490 nm.

**Measurement of intracellular oxidative stress**. HT22 cells ($2 \times 10^5$) were cultured into 6-well plate in 2 mL culture medium. HT22 cells were grown to 60% confluence and treated for 6 h under 100 μM $H_2O_2$ conditions. The solution was replaced by fresh culture medium with 10% FBS containing 6 ng/μL Au$_{25}$, Au$_{24}$Cu$_1$, or Au$_{24}$Cd$_1$ clusterzymes and cells were incubated for another 18 h. Then cells were incubated with 25 μM DHE (Beyotime, S0063) for 25 min at 37 °C in the dark to determine $O_2^{\bullet-}$ level. After 25 min, the culture medium containing DHE was removed, and the wells were washed with 0.01 M PBS. For •OH levels, the cells were incubated with 50 μM HPF solution (Sigma-Aldrich, H4290) for 25 min at 37 °C in the dark. Intracellular oxidative stress was captured by using a fluorescence microscope (EVOS, AMG), collected data by a FACS flow cytometer (BD Accuri™ C6), and used Flowjo 10.6.2 for quantitative analysis of free radical. Cells were gated based on size and granularity by forward and side scatter (SSC-A versus FCS-A). Then cell gate is analyzed for fluorescence intensity to determine the scavenging ability of clusterzymes. Among samples, control is recognized as the negative group, and $H_2O_2$ is referred as the positive group.

**In vivo treatment**. All animal procedures were approved by the Institute of Radiation Medicine, Chinese Academy of Medical Sciences, and Peking Union Medical College. Procedures were applied to minimize the number of animals used and the pain mice suffered.

**Animal models**. Male C57BL/6J mice were purchased from SPF (Beijing) Biotechnology Co., Ltd. Mice were housed in a constant temperature (21–23 °C) and animal humidity environment (45–60%) with a 12-h light–dark cycle. Food and water were available ad libitum. Surgery was performed after 1 week of transportation to adapt to the environment. Controlled cortical impact (CCI) models were conducted on adult male C57BL/6J mice (7–9 weeks, 21–23 g). C57BL/6 mice were assigned to the Sham ($n = 37$ per group), TBI, TBI+Au$_{25}$, TBI+Au$_{24}$Cu$_1$, and TBI+Au$_{24}$Cd$_1$ groups ($n = 49$ per group) randomly. Mice were anesthetized with 10% chloral hydrate (10 mg/kg) by intraperitoneal injection. The period of anesthesia was judged through skin pinching reaction and toe stimulation reaction. The surgery was conducted when the mice were in the deep anesthesia stage. Mice were fixed in a stereotaxic frame and the scalp was cut to expose the skull. The craniotomy was performed by drilling the skull above the right side of the parietal–temporal cortex in the circle of 2 mm in diameter. The circular lesion (coordinates; between bregma and lambda in the parietal bone centered at 2 mm lateral from the sagittal suture) was produced by exposing the cortex through removing the bone flap. A controlled cortex impact driven by an electromagnetically CCI injury device (eCCI-6.3, Custom Design & Fabrication, Inc.) was made with an impactor of 5 m/s velocity, 0.61 mm depth, 150 ms duration, and 20° angle of dura mater on the vertical axis. The scalp was sutured together carefully. Then the mice were kept on a heated blanket under control until being recovered from anesthesia. The clusterzymes were intravenously injected into the CCI mice at the dose of 50 mg/kg dissolved in 0.01 M PBS. An injection volume of 200 μL was used for the Sham, TBI, and other groups. All mice were marked, classified, and put into the cages (5 mice/cage) under an specific pathogen-free-level environment. Sham-injured groups (control) received the same craniotomy and intravenously injected 0.01 M PBS without CCI injury. CCI-induced TBI groups received the same injury and intravenously injected 0.01 M PBS as the vehicle group. All animals fully recovered from surgical procedures and gradually gained weight after surgery. Mice were treated with cervical dislocation after finishing related animal experiments. Brain tissues and other organs (heart, liver, spleen, lung, kidney, bladder, and testicular) were taken out on days 1, 3, and 7 post injury. MWM tests were conducted on days 13–18 and 28–32 post injury.

**Ex vivo treatment**

*Blood–brain barrier (BBB) penetration*. Brain tissues were harvested from mice with CCI injury by flushing blood from blood vessels through the heart with cold PBS at the time points of 1, 4, 12, and 24 h post injection ($n = 3$ per group per time point). Brain tissues were weighed and detected for the Au element by ICP-MS to evaluate the BBB penetration.

*Oxidative stress and inflammatory levels*. At days 1, 3, and 7 after the TBI or Sham operation ($n = 5$ per group), mice were cleaned with 10 mL PBS perfusion, and brain samples were rapidly harvested. Homogenates were centrifuged at $10,000 \times g$ for 10 min, and the supernatants were saved at −80 °C for preparation. Supernatant protein concentration was measured and used for oxidative stress- and inflammation-related cytokine quantification with enhanced BCA Protein Assay Kit (Beyotime, P0010). Oxidative stress-related factors MDA, SOD, GSH/GSSG, and $H_2O_2$ were detected with lipid peroxidation MDA Assay Kit (Beyotime,

S0131S), total SOD Assay Kit with WST-8 (Beyotime, S0101M), GSH and GSSG Assay Kit (Beyotime, S0053), and Hydrogen Peroxide Assay Kit (Beyotime, S0038). ELISA kits for IL-6 (Abcam, ab100712), IL-1β (Abcam, ab197742), and TNFα (Abcam, ab208348) were used to detect inflammation levels. These assays were carried out according to the instructions provided by the manufacturer. Each sample was detected twice at least and analyzed using the Microsoft Excel 2010 software.

*Immunostaining*. Mice brain samples were fixed in 4% paraformaldehyde (PFA) for 24–48 h, embedded in paraffin, and mounted on slides (4 μm coronal sections). The tissue slices were dewaxed twice in xylene for 10 and 5 min and dehydrated in a gradient ethanol solution (100, 95, 80, and 70%). Slices were rinsed three times and 2 min each with PBS. Antigen retrieval was performed in citrate antigen retrieval solution (C1032, Solarbio, China) at 95 °C for 10 min in a pressure cooker. After cooling naturally in the retrieval solution, slices were rinsed with PBS, followed by blocking with 5% bovine serum albumin at room temperature for 2 h, and the excess liquid was shaken off. The primary antibodies of target cells and cytokines were added at 4 °C overnight. The primary antibody information is as follows: anti-TNFα antibody (1:150, Abcam, ab183218), anti-IL-6 antibody (1:200, Bioss, bs-0782R), anti-IL-1β (1:200, Bioss, bs-0812R), antibody anti-NeuN antibody (1:800, GeneTex, GTX00837), anti-Iba1 antibody (1:300, Abcam, ab48004), and anti-GFAP (1:400, Abcam, ab90601). The primary antibody was removed and slices were rinsed with PBS three times 3 min each. Indicated fluorescence-labeled secondary antibodies were added and incubated at room temperature for 1 h in dark. The secondary antibody information is as follows: CoraLite488-conjugated Affinipure Donkey Anti-Rabbit IgG (H+L) (1:500, Proteintech, SA00013-6), Goat Anti-Chicken IgY H&L (Alexa Fluor® 647) (1:1000, Abcam, ab150171), Donkey Anti-Sheep IgG H&L (Alexa Fluor® 647) (1:1000, Abcam, ab150179), and Donkey Anti-Goat IgG H&L (Alexa Fluor® 647) (1:1000, Abcam, ab150131). Finally, slices were mounted with anti-fade mounting medium with 4,6-diamidino-2-phenylindole (S2110, Solarbio, China) and photographed with fluorescence microscope (EVOS, AMG). For immunohistochemistry staining, the primary antibody information utilized is as follows: anti-TNFα antibody (1:200, Abbkine, ABP0127), anti-IL-6 antibody (1:200, Proteintech, 66146-1-lg), and anti-IL-1β (1:200, Abbkine, ABP52932). After rinsing with PBS, the biotinylated secondary antibody was initially applied for 30 min, after reaction enhancer in Universal Two-step Detection Kit (ZSGB-BIO, pv9000) was used for additional 30 min. 3,3'-Diaminobenzidine tetrahydrochloride hydrate was utilized for detection. Slides were then counterstained with hematoxylin to stain nuclei. Samples were captured by microscopy.

*MWM tests*. MWM performance was assessed on days 13–18 and 28–32 following injury ($n = 7$ per group). The hidden platform test was used to investigate spatial learning and memory. Water in the pool was maintained at 25 °C (±1 °C), made opaque by milk. Before spatial learning, visual discrimination learning was performed to determine whether the vision of mice was normal. In this procedure, each animal performed one trial where the platform was placed above the water level without recording. Spatial learning was assessed across repeated trials for 5 days approximately at the same time each day between 13:00 and 18:00. A circular stainless steel tank 122 cm in diameter and 51 cm in height on both sides with non-reflective interior surfaces was used. The water maze was divided into four quadrants (I–IV), and the platform was set in the center of quadrant I. Mice were given four trials a day with an inter-trial interval (ITI) of 60 min using a starting position randomly within four positions. The procedures of MWM followed a previously published protocol[76]. Each trial was limited within 60 s with an ITI of 15 s. The mouse failing to find the platform hidden behind the water within this time limit was allowed for 15 s to be placed on the platform or guided to learn. The trials were repeated for 5 days, and the probe trials to evaluate long-term spatial memory were administered for 60 s on day 6 at a new starting position with the platform removed. Latency to locate and rest on the hidden platform and the distance traveled (path length) to the hidden platform were recorded for spatial learning trials. During the probe trials, the percentage of time spent in the missing platform quadrant and the number of platform location crossings were recorded to analyze the search strategy of mice in each group.

*Western blotting*. Tissue samples ($n = 3$ per group per day) were lysed, operated on ice, and extracted protein from brain homogenates in radio-immunoprecipitation assay lysis buffer (strong) (CWBIO, CW2333) containing protease inhibitors at 95 °C for 5 min. Tissue extract supernatant protein concentrations were determined by a BCA assay (Beyotime, P0010). Sodium dodecyl sulfate-polyacrylamide gel electrophoresis was performed to resolve protein lysates (50 μg) before protein lysates were transferred onto nitrocellulose membranes (CWBIO, CW0022S). Antibodies specific for TNFα, IL-6, and IL-1β were used: anti-TNFα (1:1000, Abcam, ab34674), anti-IL-6 (1:1000, Abcam, ab7737), anti-IL-1β (1:1000, Abcam, ab234437), and β-actin antibody (1: 2000, Sigma-Aldrich, A5441). Full scans of all the blots used in this work are provided in Source data.

*Pharmacokinetic and toxicological studies*. The pharmacokinetic parameters of clusterzymes were measured on 7–9 weeks (21–23 g) male C57BL/6J mice. Mice were subject to intravenous injection at a dose of 50 mg/kg in 200 μL volume to

evaluate the biodistribution ($n = 3$ per group), blood half-life ($n = 3$ per group), and excretion ($n = 3$ per group) of different clusterzymes. Mouse organs (heart, lung, liver, spleen, kidney, muscle, bladder, testicles, intestine, and brain) were collected, washed with PBS, and weighed 24 h post injection. To determine the half-life of clusterzymes, blood was collected from the retro-orbital sinus at 2 min, 12 min, 1 h, 3 h, 5 h, 8 h, 24 h, and 48 h, and the volume was 50 μL. The elemental Au in organs and blood was quantified with ICP-MS. Standards were prepared and counted along with tissue samples to calculate the percentage-injected dose per gram of tissue (%ID/g). Drug excretions of clusterzymes were determined as well. Stool and urine were collected within 48 h and detected for Au element. Hematology and blood biochemistry panels were detected on the day 7 post injection. Blood samples were obtained from retro-orbital sinus and saved in tubes with K2EDTA for testing. Blood samples for biochemistry analysis were left to stand for 30 min and then centrifuged twice at 3500 rpm for 15 min. All organs were collected and fixed in 4% PFA for 24–48 h, embedded in paraffin, and mounted on slides (4-μm coronal sections). Slides were stained through hematoxylin–eosin staining to observe the toxicity of clusterzymes in major organs, including the heart, liver, spleen, lung, kidney, and brain.

**Quantitative analysis of immunostaining**. Quantitative image analysis of the immunofluorescence for GFAP, Iba-1, and NeuN cells were performed on five cerebral cortex areas of each 4 brain slices taken with the ×40 objective ($n = 3$ mice per group). Immunofluorescence intensity was calculated using the threshold method and defined as the average number of pixels per slice by the ImageJ software, then divided by the area (mm²) in the imaged field with the average background subtracted[77,78]. For quantification of the immunofluorescence double staining, the co-expressed cells in the five regions of the cortex of each 4 brain slices were counted under a microscope (EVOS, AMG) at ×400 magnification ($n = 3$ mice per group). The results are expressed as an average number of positive cells per unit area (mm²) of each slice[79,80]. For the quantitative analysis of inflammatory factors in immunohistochemistry, the investigators who were blinded to the experimental groups randomly collected five high-power field images at ×400 magnification in cerebral cortex areas under a microscope (EVOS, AMG) of each animal ($n = 3$ mice per group)[80]. The cytoplasmic staining areas that showed light yellow or brownish yellow were selected as positive cells, and the expression of inflammatory factors was quantified by the average count of positive staining cells per animal.

**Statistic methods**. Data are presented as mean ± standard deviation (SD) or standard error of the mean (SEM). For multiple comparisons, one-way analysis of variance (ANOVA) was performed using the SPSS 19 software to assess difference in means among groups and compared with the Sham and TBI groups, analyzed by ANOVA.

**Reporting summary**. Further information on research design is available in the Nature Research Reporting Summary linked to this article.

## Data availability
The data that support the findings of this study are available from the corresponding author upon reasonable request. Source data are provided with this paper.

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

## Acknowledgements
This work was financially supported by the National Natural Science Foundation of China (Grant Nos. 91859101, 81971744, U1932107, 814717866, and 11804248), the Independent Innovation Foundation of Tianjin University, the Natural Science Foundation of Tianjin (Grant No. 18JCQNJC03200), and the NSF (Grant No. IRES 1826917).

## Author contributions
X.-D.Z. conceived and designed the experiments. H.L. contributed to materials synthesis, H.L., S.L., and X.Y. contributed to physical and chemical measurement; Y.L. and K.V. contributed to the simulation of the theoretical calculation; and S.S., Q.X., and K.C. contributed to biological experiment. X.-D.Z., J.X., S.S., H.L., X.M., H.W., W.M., and Y.L. analyzed the data; X.-D.Z., S.S., H.L., and Y.L. prepared the manuscript. All authors discussed the results and commented on the manuscript.

## Competing interests
The authors declare no competing interests.
