## [Peer Review File · Nature Communications]

REVIEWER COMMENTS

Reviewer #1 (Remarks to the Author):

This work reports the enzyme-like activity of Au₂₅ and Cu-/Cd-monodoped clusters. As far as I know, this is probably the first study on the enzyme-like properties of M₂₅(SR)₁₈ with well defined atomic structures, though other nanomaterials such as iron oxides and noble metals (without explicit atomic structures) were reported previously.

The observed high activity of M₂₅ is impressive, and the GPx, SOD, and CAT selectivity of Au₂₅, Au₂₄Cd₁, Au₂₄Cu₁, respectively, is also interesting. DFT simulations provide further insights into the active site and mechanism for the SOD-CAT selectivity based upon the atomic structures of M₂₅(SR)₁₈. The relationship between oxidative stress and neuroinflammation is established.

Overall, I think this work meets the expectations of NC.

Some technical aspects may be improved.

Technical questions:

1 The authors mentioned the Cu or Cd atom "resides at the surface sites of the Au₁₃ core or oligomer sites". Did the authors perform coordination # analysis to locate the dopant precisely? This is highly relevant to the active site identification.

2. The active center structure isn't clearly discussed. While Fig 4 shows the DFT simulations, the very small structures are difficult for me to find out the active site structure (whether the staple Au sites or the underlying shell Au sites). The authors mentioned the modulated bond lengths to the active center, but the discussions aren't very clear. These things should be discussed more clearly.

3. In the calculation of TOFs, was the entire Au₂₅ counted as one active site or every surface atom as active sites?

Minor:

p7, "reaction velocity": one usu. says "reaction rate".

p7, "exploited": I guess "exploited", right?

p11, "the explained interfaces": what does this mean? Do you mean the bowl?

p15, artific[i]al enzymes

Reviewer #2 (Remarks to the Author):

This study describes novel clusterzymes that have preferential enzyme-mimicking catalysis which confers GPx-like and CAT-like activities in neurons and brain that may be relevant for mitigating the damaging effects of TBI in preclinical models. TBI induces secondary injury that includes upregulation of oxidative stress/damage and loss of antioxidant capacity, and this is known to play a role in secondary inflammation and long-term neurodegeneration. As such, inhibiting oxidative stress/damage pathways following TBI are important and rational therapeutic targets to improve outcomes in head injury patients. The identification of novel Au₂₄Cd₁ and Au₂₄Cu₁ enzymes is exciting, and the characterization of their activities in cell based assays suggest that they are more potent antioxidants when compared to standards in the field. Unfortunately, the translation of cell based studies to in vitro and in vivo model systems are problematic. There are many errors in preclinical experimental design and an over-interpretation of the effects of therapeutic Au₂₄Cd₁ and Au₂₄Cu₁ on oxidative stress, inflammation and neurobehavioral recovery. The authors should investigate the clusterzymes using clinically-relevant drug administration protocols (IV, IP, ICV), and improve the preclinical design to include early timepoint relevant to oxidative damage/inflammation.

The following points were identified during review of the manuscript and should be addressed by the authors.

1) Description and rigor of CCI model of TBI. The methods for the CCI model lack appropriate details that are essential for preclinical modelling studies. Were the animal studies approved by an animal ethics and use committee (ICAUUC)? What sex of mice were included in the study? What age of mice? What instrument was used to induce CCI? Details of the instrument and mechanism of impact (pneumatic or electromagnetic) are needed. What was the sham control procedure - did these animals receive craniotomy without impact? What post-surgery care was provided to animals? Were analgesics used after surgery (subcutaneous or topical). Overall, there is a lack in scientific rigor in describing the model, and this is a major issue.

2) Route of post-injury Au²⁴Cu¹/Au²⁴Cd¹ administration: "the clusterzymes was added to the wound of TBI mice at the concentration of 10ug/ul". This description is wholly inadequate and the route of administration is not quantifiable or translational for human studies. Typically in preclinical TBI studies IV, IP, or ICV administration routes are used for neuroprotection analysis of therapeutic drugs. A translationally relevant drug administration route is strongly advised to determine the therapeutic potential of Au²⁴Cu¹/Au²⁴Cd¹.

3) Figure 5A. The timeline schematic of oxidative damage and antioxidant responses in injured brain is misleading because many enzymatic/catalytic events co-exist early after TBI and do not occur in the linear fashion depicted in the schematic. Oxidative damage is much more complex and the timeline schematic is problematic. Critical for the in vivo experiments (Fig 5D-G) many of the antioxidant and oxidative stress responses peak early after TBI, usually within 1-3 days post-injury. Therefore, the delayed analysis of these responses at 7 and 14 days post-injury are sub-optimal.

4) OH^{*} and O₂^{*-} should be quantified in the in vitro experiments using HT22 neuronal cell cultures. Does reduced oxidative stress due to Au²⁴Cu¹ and Au²⁴Cd¹ treatment result in reduced neuronal cell death in these experiments? Viability data should be include to correlate reduced oxidative stress with neuronal health in the H₂O₂ assays.

5) Fig 5D-G, as mentioned before earlier time points at peak of oxidative damage should be included in these experiments. What N was used in the in vivo experiments? It is not clear if there are statistical differences between Au²⁴ treated and vehicle control treated TBI groups. Further, a student's t-test is not appropriate for these comparisons. At a minimum one-way ANOVA should be used, or preferentially two-way ANOVA to capture effect of Time post-injury.

6) Western blot quantification for IL-1b, IL-6 and TNF should be included in Fig 6B,C. For IL-1b, expression levels are very high in sham (control) and there is no induction of IL-1b following TBI. Also, is this mature IL-1b in Western or pro-forms? The authors should include the full uncropped blot for this analysis. The IL-6 Western blot has been cut/stitched using photoshop or equivalent software. Full unmodified blots need to be included for scientific rigor.

7) The chronic expression of IL-1b, IL-6 and TNF is surprising following CCI. These proinflammatory mediators are upregulated early after TBI and do not often persist at more chronic timepoints (7 and 14d post-TBI). Analysis at earlier time point would be optimal. Statistical analysis is problematic in fig 6., and ANOVA is required. Does Au²⁴Cu¹ and Au²⁴Cd¹ treatment reduce expression of cytokines in the injured cortex?

8) What time post-injury is IL-1b and IL-6 imaging in Fig 6G? Quantification is needed for this analysis, and colocalization studies with markers for neurons (NeuN), microglia (Tmem119) or astrocytes (GFAP) is recommended to determine what cell type produces these pro-inflammatory mediators after TBI.

9) The wound healing analysis in Fig S24 is not useful. The data are subjective and it is not known how healing at surgery site related to brain injury response and recovery. Please remove this.

10) Astrocyte analysis in Fig S25 needs to be quantified because morphological transformation of

GFAP astrocytes occur after TBI. Number of GFAP+ astrocytes and morphological characteristics should be measured. What region of the injured brain was analysed, and at what time post-injury?

11) Inflammatory infiltrates at 30 days post-injury is also non-specific and does not add value. Peripheral immune cells (monocytes, B and T cells) are unlikely to still be trafficking to the brain at this time point due to closing of the BBB. If immune infiltration is important and relevant in this study then specific analyses using flow cytometry or related assays are needed.

12) Higher magnification images of IL-6 are needed in Fig S27. Is IL-6 expressed in neurons? Negative controls without primary antibody should be included to demonstrate specificity of antibody in IHC studies.

13) The MWM protocol used in these experiments is not standard and it is difficult to follow the analysis. The authors need to clearly state the N number per group used in this experiment. The error bars are large – are they SEM or SD? It appears that day 5 analysis is the probe trial without platform. It should not be presented with latency data in Fig S28A,B. A major concern about this analysis is that the mice do not appear to be learning the task during the acquisition phase (Fig S28B). All mice have latency times at 60-70sec on day 1 and latency to find hidden platform is not reduced on each subsequent day/trial. Moreover, the effect of TBI is absent, and the Sham (control) mice have equal latency over all testing days as TBI mice. This is a major issue and invalidates the MWM test. The frequency analysis in each quadrant (S28C,D,E) is not standard, and although it shows trends in improvements with Au₂₄Cu₁ or Au₂₄Cd₁ treatment it is not convincing. A proper analysis of search strategy and traditional probe trial (% time in missing platform quadrant) is needed. Also, was a visual acuity trial performed to rule out impairments in visual function required for this task?

Reviewer #3 (Remarks to the Author):

In this paper, the authors designed a unique artificial enzyme, clusterzyme, by single-atom substitutions in MPA-protected Au cluster. Interestingly, the clusterzymes not only showed high antioxidant property, but also performed selective enzyme-like activity towards GPX, CAT, SOD. Theoretical calculation revealed that the catalysis is modulated by the single active site of Au₂₄Cd₁ and the Cu-Au dual active sites of Au₂₄Cu₁ at bond lengths. Importantly, such clusterzymes have showed therapeutic effect on neuroinflammation in TBI model. The work not only provide a new type of artificial enzymes by precise design of Au cluster at single atom level, but also provide fundamental understanding on these clusterzymes and confirmed their bio-functionality. In general, the clusterzymes proposed in this paper may be a new type of nanozymes. Therefore, I think it merits consideration for publication on Nature Communications.

Below are some concerns and questions.

1. In page 3, introduction section, for POD-like activity, the description for “the affinity to substrates” is not clear, as there are two substrates in POD-like activity. The authors need to clarify it.
2. In the paper, the authors demonstrated that three clusterzymes showed preferential enzyme-like activities, Au₂₅ for GPx, Au₂₄Cu₁ for CAT and Au₂₄Cd₁ for SOD. Is it possible to measure the specific activity (U/mg) for each clusterzyme and compare it with the corresponding natural enzyme?
3. In Figure S23, the cell viability of HT22 sharply decreased when Au₂₄Cd₁ up to 100µg/mL.
4. The authors observed that the clusterzymes can significantly decreases inflammation factors such as IL-1β, IL-6, and TNF-α. Can the authors provide some explanation for the possible pathways?
5. What is the pharmacokinetics of clusterzymes in vivo? Due to ultrasmall size, I am wondering if

it has a very short half-life in vivo. In addition, it is not clear whether the clusterzymes can pass through BBB for TBI therapy.

6. How is the stability of clusterzymes in vivo? Can it be degraded under physiological conditions?

7. A lot of typo errors in the paper, such as "activity is week", exploited, theBMPO, TheH2O2, severeoxidative, areupregulated, showminor, THFa(in Supplementary Information). The authors need to check and revise the language carefully.

8. The reference style is not consistent and need to be carefully revised.

Author's Response to Reviewer #1

This work reports the enzyme-like activity of Au_{25} and Cu-/Cd-monodoped clusters. As far as I know, this is probably the first study on the enzyme-like properties of $\text{M}_{25}(\text{SR})_{18}$ with well-defined atomic structures, though other nanomaterials such as iron oxides and noble metals (without explicit atomic structures) were reported previously. The observed high activity of M_{25} is impressive, and the GPx, SOD, and CAT selectivity of Au_{25} , $\text{Au}_{24}\text{Cd}_1$, $\text{Au}_{24}\text{Cu}_1$, respectively, is also interesting. DFT simulations provide further insights into the active site and mechanism for the SOD-CAT selectivity based upon the atomic structures of $\text{M}_{25}(\text{SR})_{18}$. The relationship between oxidative stress and neuroinflammation is established. Overall, I think this work meets the expectations of NC. Some technical aspects may be improved. Technical questions:

1. The authors mentioned the Cu or Cd atom "resides at the surface sites of the Au_{13} core or oligomer sites". Did the authors perform coordination # analysis to locate the dopant precisely? This is highly relevant to the active site identification.

Reply: Thank you very much for the constructive suggestion. According to reviewer's suggestion, we carried out fitting analysis for the EXAFS data of Cu or Cd to obtain the coordination number (CN) of the Cu or Cd atom, so as to obtain precise locations of dopants. **Figure 1R** shows the R space of the EXAFS data of the Cu K edge in $\text{Au}_{24}\text{Cu}_1$. It can be seen that there is only one major peak in the range of 1.6-5.0 Å. This peak roughly correspond to the scattering path of photoelectron waves from the X-ray absorbing Cu atom to the neighboring S atoms of different shells, and we used IFEFFIT program to fit this peak. The EXAFS parameters obtained after fitting are shown as in **Table S1**. The CN of Cu-S obtained from the fitting is 1.9 ± 0.2 Å. This value is close to 2 Å, which may indicates that the first replacement of Au_{25} by a Cu atom occurs at the oligomer site. (*J. Phys. Chem. Lett.* 2012, 3, 2209-2214; *J. Phys. Chem. C* 2014, 118, 25284-25290). Similarly, we also processed the fitting for Cd-L₃ edge of $\text{Au}_{24}\text{Cd}_1$. It can be seen that the scattering path of photoelectron waves from the absorbing Cd atom to the S atoms is observed in the range of 1.6-5.0 Å. The parameters of EXAFS obtained after fitting are shown in **Table S1**. The CN of Cd-S obtained by fitting is 2.3 ± 1.7 Å, which is close to the CN of the bond with S at the oligomer site of Au_{25} , indicating that Cd atom substitution may occur at the oligomer site. The corresponding results and discussion have been added in the revised manuscript.

Figure R1. a Cu K-edge and b Cd L₃-edge by EXAFS fitting in the R space of $\text{Au}_{24}\text{Cu}_1$ and $\text{Au}_{24}\text{Cd}_1$.

Changed in the "Revised Manuscript" (Figure 1, Page 24):

Figure 1. Structural characterization of Au_{25} clusterzymes. **a** Structure illustration of Au_{25} , Cu and Cd substituted $Au_{24}Cu_1$ and $Au_{24}Cd_1$. **b** UV-vis absorption spectra and **c** Electro spray ionization mass spectra (in negative ion mode) of Au_{25} (black curve) before and after Cu (green curve) and Cd (red curve) substitutions. The inset in **b** is a magnification of the absorption spectrum at ~ 670 nm. It can be seen that the characteristic absorption is slightly redshifted after introduction of Cu and Cd, indicating decreased band gaps. The red line of the insets in **c** represents the simulated isotope distribution of $[Au_{25}MPA_{18}-3H]^{3-}$, $[Au_{24}Cu_1MPA_{18}-3H]^{3-}$, $[Au_{24}Cd_1MPA_{18}-3H]^{3-}$, respectively. **d** Cu K-edge and **e** Cd L₃-edge FT-EXAFS spectra and associated fitting in R space of $Au_{24}Cu_1$ and $Au_{24}Cd_1$, showing the surrounding atoms adjacent to the Cu and Cd atoms.

Added in the “Supporting Information” (Table S1, Page 54):

$Au_{24}M_1$	CN	R(Å)	$\sigma^2(10^{-3}\text{Å}^2)$	R-factor
Cu-S	1.9 ± 0.2	2.25 ± 0.02	10.8 ± 3.1	0.028
Cd-S	2.3 ± 1.7	2.3 ± 0.42	1.7	0.037

Table S1. Fitting parameters of Cu K-edge and Cd L₃-edge EXAFS for $Au_{24}Cu_1$ and $Au_{24}Cd_1$. CN is the coordination number, R is the bond length, and σ^2 is the Debye-Waller factor.

Changed in the “Supporting Information” (Figure S6, Page 21):

Figure S6. **a** Cd L₃-edge and **b** Cu K-edge XANES spectra of Au₂₄Cu₁ and Au₂₄Cd₁ clusterzymes. It can be seen that the Cd-L₃ and Cu K-edge XANES spectra are different from the spectra of Cd and Cu bulk, showing characteristics of oxidized states of Cd and Cu.

Changed in the “Revised Manuscript” (Line 12, Page 2):

“Cu single active site”

Changed in the “Revised Manuscript” (Line 16, Page 4):

“single Cu”

Changed in the “Revised Manuscript” (Lines 10-23, Page 5 and Lines 1-14, Page 6):

“X-ray photoelectron spectroscopy (XPS) further confirms that Au (0) is the dominant state in all clusterzymes (**Figure S4**). To identify the precise spatial atomic configuration, Extended X-ray Absorption Fine Structure (EXAFS) spectra at the Au, Cu and Cd edges were recorded (**Figure 1d, e** and **S5-6**). The L₃ edges of Au in all clusterzymes have higher white-line intensities than the bulk standard Au foil. This is ascribed to larger surface area and alloying effects from partial oxidation with more *d*-band vacancies from nanoscale sizes and surface molecule-like interactions (Au(I)-thiolate). The characteristic absorption edges of Au clusterzymes were found at ~11920 eV, which is assigned to the 2p→5d electronic transition of Au suggesting a reduced population of unoccupied valence *d*-states. The increase of intensity in MPA-protected Au₂₄Cu₁ and Au₂₄Cd₁ indicates that the density of 5*d* electrons of Au is decreased by the one atom substitutions of Cu and Cd through the transfer of their 4*d* electrons (**Figure S5**).^{44,45} The k-space oscillations of Au₂₅ clusters and the Au foil are shown in **Figure S5**. The k-space of the Au foil exists in typical fcc oscillation patterns which are apparently absent in all Au clusterzymes due to their small core sizes. Besides, we also investigated the XANES spectra of Cu and Cd foils as well as the corresponding atomic counterparts within clusterzymes, clearly displaying differences between single atoms and bulk metals (**Figure S6**). To further pinpoint the doping sites of Cu and Cd atoms, we performed fitting analysis on the EXAFS data of Cu and Cd. **Figure 1d** shows the R space of the EXAFS data of the Cu K edge in Au₂₄Cu₁. It can be seen that there is only one major peak in the range of 1.6-5.0 Å. This peak roughly correspond to the scattering path of photoelectron waves from the X-ray absorbing Cu atom to the neighboring S atoms of different shells, and we used IFEFFIT program to fit this peak. The EXAFS parameters obtained after fitting are shown in **Table S1**. The Cu-S coordination number obtained from the fitting is 1.9 ± 0.2 Å. This value is close to 2 Å, which may indicates that the replacement of Au₂₅ by a Cu atom occurs at the oligomer site, consistent with previous work.⁴⁴ Similarly, the R space of EXAFS data on Cd L₃ edge in Au₂₄Cd₁ shows a peak in the range of 1.6-5.0 Å, and the fitted Cd-S coordination number is 2.3 ± 1.7 Å, which is close to the coordination number of the bond with S at the oligomer site of Au₂₅, indicating that Cd atom substitution may occurs at the oligomer site. (**Figure 1e**).⁴⁶”

Changed in the “Revised Manuscript” (Figure 3, Page 26):

Figure 3. Enzyme-mimetic properties and ROS scavenging activity of Au₂₅ clusterzymes. **a** The radar map of enzymatic activities and free radical scavenging abilities of various clusterzymes. **b** Schematic illustration of catalytic selectivity of the clusterzyme system. Au₂₅ exhibits significant superiority in GPx-like activity; Au₂₄Cu₁ shows advantages in the CAT-like activity through the Cu *single* active site; Au₂₄Cd₁ preferably exhibits the SOD-like activity *via* the Cd single active site, each demonstrating a unique catalytic selectivity. **c** GPx-, **d** CAT-, **e** SOD-like activities of Au₂₅, Au₂₄Cu₁ and Au₂₄Cd₁. ROS scavenging activities of Au₂₅, Au₂₄Cu₁ and Au₂₄Cd₁ clusterzymes for **f** •OH, **g** O₂^{•-} studied by the ESR spectroscopy. BMPO is used as the ROS capturing agent and the sources of •OH and O₂^{•-} are H₂O₂ and KO₂, respectively. **h** Corresponding quantifications of the scavenging efficiencies.

2. The active center structure isn't clearly discussed. While Fig 4 shows the DFT simulations, the very small structures are difficult for me to find out the active site structure (whether the staple Au sites or the underlying shell Au sites). The authors mentioned the modulated bond lengths to the active center, but the discussions aren't very clear. These things should be discussed more clearly.

Reply: Thank you very much for the constructive suggestion. We appreciate the suggestion of CN analysis from the reviewer which allows us to reexamine the structure, the corresponding active center and the dynamics during the catalytic processes. In the updated version of this manuscript, we replaced the discussion of surface replacement by the oligomer replacement which has been extensively simulated by our group recently. The summary of the simulation changes includes the oligomer bend and catalytic mechanisms. The energy diagrams are updated with respect to the optimized geometries of oligomer replacement but they still agree with the experimental results (selectivity) very well. Therefore, the reaction pathways are still kept the same as in the previous version. To address the mechanism and the active center, we focus on the simple adhesion and bond modulated adhesion. The former is trivial and the latter is featured by the angular motion. To demonstrate the non-trivial angular motion, **Figure 4**, **Figure S21**, **Figure S24** and **Table S2** and **S4** are used to describe it from different aspects: the motion of the doped

is accompanied by the Cd-S bond changes. We hope these clarifications help explain the mechanism of the “modulated bond length”. The corresponding results and discussion have been added in the revised manuscript.

Changed in the “Revised Manuscript” (Lines 19-23, Page 9; Lines 1-23, Page 10; Lines 1-23, Page 11; Lines 1-23, Page 12 and Lines 1-7, Page 13):

“To evaluate the catalytic behavior and the intermediate states during the chemical reactions, each ligand unit -SCH₂CH₂COOH is simplified to -SCH₃. DFT optimization confirms the stability of the modeled cluster.

Due to the symmetry of the gold cluster, all possible replacements of the guest metallic atoms fall into 3 categories as follows: oligomer, the surface of core and core replacements. **Figure 4a** demonstrates the surface sites of the Au₁₃ core and the oligomer site replacement. The oligomer replacement is the common form which is extensively discussed in the literature, but DFT simulations indicate that the surface replacement may be another possibility. However, based on the coordination analysis in the experiment, the oligomer replacement matches the EXAFS results which yield a significantly lower coordination number (CN) than the surface replacement. With the optimized structure of the clusters, the associated CNs can be theoretically generated even within the clusters involved in the intermediate structures during the catalytic process. The averaged simulated CN values agree with the experimental values that confirm the oligomer replacement (**Table S2**). Although more theoretical investigation on the surface replacements can be found in the appendix (**Figure S16-19**, and **Table S3**), we focus on the oligomer replacement and the associated catalytic efficiency.

Unlike the surface replacement which may cause the expansion of the core, the oligomer replacement causes the oligomer bending. It is different from the normal S-Au-S chain which aligns in a (nearly) straight line (**Figure S20**). The doped Cu shrinks the S-X-S chain while the doped Cd extends it. Compared with the typical bond length of S-Au at 2.3 Å, S-Cu and S-Cd bonds are 2.2 and 2.55 Å respectively, as shown in **Figure 4b** and **Figure S21**. With the bent chain, the distances between Cu/Cd atoms to the surface of the core are comparable, around 3.1 Å. The similarity between the Cu and Au atoms guarantees the binding of S-Cu-S is “firm” such that the relative positions of Cu to S atoms can be hardly changed by the dynamics during the catalytic procedure which are discussed extensively below. In contrast, the relative position of the doped Cd atom may be significantly affected by the local environment such as the adhesion of small chemical units (**Figure S22-23**).

We observed the outstanding performance of the clusterzymes in both CAT and SOD reactions with the reaction pathways summarized in **Figure 4c**. The CAT reaction usually refers to the catalytic degradation of hydrogen peroxide, and the decomposition mechanism of H₂O₂ may involve multiple chemical stages (**Figure 4d, e**). For the process of SOD, we assumed the clusters are involved in similar mechanisms to the general catalytic scheme of SOD reaction. It is worth noting that the release of oxygen completes the CAT process, while the SOD process also occurs simultaneously, and the two processes are mutually permeated. The reduced cluster, Cluster(I), may also be involved in both CAT and SOD processes which depends on the concentration of different components.

Inspired by the Arrhenius equation, we performed the search of transition states and ground states of various types of molecules and ions to estimate the activation energies and evaluated the catalytic efficiencies. The energy profiles in **Figure 4f, g** agrees with the behaviors of the clusterzymes in our experiments. In a series of reactions with multiple steps, the reaction rate is dominated by the slowest step, i.e. the transition with the largest activation energy. Such a feature can be seen in the first part of the catalytic SOD process by Au₂₄Cu₁. The ground state of the electrons in the corresponding intermediate structure is a triplet state suggested by DFT simulations. It indicates that the high activation energy of 131.2 kJ/mol is related to the spin matching issues which can be selective to the spin of superoxide ions. In the CAT processes, the critical step is related to the decomposition of (cluster...OOH)²⁺, in which the Au₂₄Cd₁ exhibits higher activation energies (71.3 kJ/mol) which reduce the efficiency. The simulations clearly explain the SOD-CAT selective behaviors of the doped clusterzymes. The details of the reaction pathways are provided in the supporting information.

Our results of DFT simulations show some insights of the catalytic mechanisms. Assume in typical clusters, a substituted atom may turn into an active site itself to be involved in the catalytic process which may be accompanied by changes in the geometry. Herein, we named two mechanisms as SA (simple adhesion) and MA (bond modulated

adhesion) correspondingly. The distances between the adsorbed molecule/ion and metal atoms designate the roles of the substituted atom.

The SA mechanism is mainly seen in $\text{Au}_{24}\text{Cu}_1$. Due to the firmness of the S-Cu-S oligomer, the Cu atom is relatively rigid (**Figure 4f, g**). The SA mechanism is also seen in the first step of SOD process catalyzed by $\text{Au}_{24}\text{Cd}_1$. The catalytic process includes the distance change and orientation change of small units. Significant changes in the distance between the active site (doped atoms) and the small units are seen in most of the SOD processes. In contrast, the orientation change is the main character in most of the CAT processes.

The MA mechanism is seen in $\text{Au}_{24}\text{Cd}_1$ on the 2nd stage (Cluster(I)) of the SOD processes and most of the CAT processes. The bond modulation refers to the position change of the Cd atom which may deviate from the oligomer plane until a third S atom from another oligomer stops it. Thus, the S-Cd bonds are changed significantly (**Figure 4f, g**). The characteristics of transition and intermediate states involve the rotation of the superoxide ion (or oxygen molecule) and the position adjustment of the doped Cd atom. To be more precise, the motion of the Cd atom is along the perpendicular direction of the oligomer plane. Once a small unit joins the doped cluster to form an intermediate structure, the Cd atom sometimes leaves the oligomer plane. Therefore, the CN value of the Cd atom is larger for the Cd atom becomes the neighbor of 3 S atoms. When the Cd atom starts from its original state (S-Cd = 2.55 and 2.55 Å), passes its transition state (S-Cd = 2.63 and 2.85 Å) and arrives at the intermediate state (S-Cd = 2.57 and 2.60 Å), the angular motion is terminated (**Figure S24**). During such a process, the distance between the attached oxygen atoms is slightly expanded towards the normal distance of oxygen molecules which indicates the completion of the entire catalytic procedure (**Table S4**). A similar procedure for the $\text{Au}_{24}\text{Cd}_1$ can be observed at the adhesion of OOH- at the first stage of CAT process. Such a unique process allows the doped Cd atom as an active site that can be self-modulated in a wide spatial range compare to the firm Cu atom. This may explain its good performance in the SOD process.”

Changed in the “Revised Manuscript” (Figure 4, Page 27):

Figure 4. DFT calculations and the mechanism of catalytic selectivity. **a** Demonstration of atomic doping: surface and oligomer replacement. **b** Doped atom caused change in bonds: the more rigid Cu-S bond and the more flexible Cd-S bond. The flexibility of Cd allows the angular motion. **c** Mechanism and **d-e** energies profiles of catalytic process of the SOD and CAT processes. The black dotted line indicates that the catalytic products may be connected to other processes. Energies profiles and geometry structure of the intermediate states of **f** SOD and **g** CAT process in the lower panel. Align optimized intermediate structures (normal color) and transition structures (black).

Added in the “Supporting Information” (Figure S21, Page 36):

Figure S21. S-Cd and S-Cu bonds in optimized structures. Lengths of the bonds are marked above.

Added in the “Supporting Information” (Figure S24, Page 39):

Figure S24. An intermediate structure of the angular motion. Blue line: the oligomer plane. Blue arrow: the deviation of Cd atom off the plane.

Added in the “Supporting Information” (Table S2, Page 55):

Table S2. The CN values in the intermediate structures (including AuCd-OH²⁺, AuCd-OH²⁺, AuCd-OH²⁺, AuCd-OH²⁺) are generated without the attached units.

Structure	CN (Theoretical)	CN (Experimental)
AuCd ²⁺	Cd 3.762	
AuCu ²⁺	Cu 2.597	
AuCd-OH ²⁺	Cd 2.501	
AuCu-OH ²⁺	Cu 2.511	
AuCd-OOH ⁺	Cd 2.592	
AuCu-OOH ⁺	Cu 2.081	
Average	Cd 2.952	Cd 2.5
	Cu 2.400	Cu 2.0

Added in the “Supporting Information” (Table S4, Page 57):

Table S4. The bond lengths between the ion/segment and the clusterzymes with oligomer replacement in the optimized intermediate structures in CAT and SOD processes. the O-O bonds are also provided.

SOD	O-X distance	S-X bond length	O-O bond length
Cu (I)	3.69	2.25	1.25
Cu (I) ts	2.08	2.27	1.31
Cu (0)	3.98	2.25	1.26
Cu (0) ts	3.43	2.25	1.25
Cd (I)	3.32	2.56	1.26
Cd (I) ts	3.45	2.56	1.26
Cd (0)	2.26	2.57; 2.60	1.39
Cd (0) ts	2.27	2.63; 2.85	1.35
Superoxide	-	-	1.41

CAT	O-X distance	S-X bond length	O-O bond length
Cu (I)	2.02	2.28	1.46
Cu (I) ts	2.07	2.27	1.45
Cu (II)	2.07	2.27	-
Cu (II) ts	2.04	2.25	-
Cd (I)	2.15	2.60; 2.65	1.58
Cd (I) ts	2.13	2.89; 2.62	1.60
Cd (II)	2.08	2.67; 2.60	-
Cd (II) ts	2.04	2.60; 2.70	-

3. In the calculation of TOFs, was the entire Au₂₅ counted as one active site or every surface atom as active sites?

Reply: Thanks for your constructive comment. The molecule-like bimetallic nanocluster represents a class of catalysts that bridge homogeneous and heterogeneous catalysis. Like molecular catalysts, bimetallic nanoclusters carry a discrete charge for reactions, which sets these clusters apart from other metal nanoparticle-based systems. Indeed, for most Au₂₅ clusters, researchers use the entire molecule as the active site to calculate the turnover frequency (*Nat. Commun.* 2017, 8, 14723; *Nanoscale*, 2018, 10, 6558-6565; *Nano Res.* 2019, 12, 501-507). Therefore, in this study we also used the entire Au₂₅ molecule as one active site when calculating the catalytic activity of clusterzymes.

Changed in the “Supporting Information” (Lines 7-8, Page 23):

“E is the molality of clusterzymes”

Changed in the “Supporting Information” (Line 4, Page 24):

“E is the molality of clusterzymes”

4. Minor:

p7, "reation velocity": one usu. says "reaction rate".

p7, "explointed": I guess "exploited", right?

p11, "the explained interfaces": what does this mean? Do you mean the bowl?

p15, artific[i]al enzymes

Reply: We sincerely thank the reviewer for careful reading, “the explained interfaces” refers to the “bowl” shaped region, that is, the reaction site exposed to the “bowl” plane. As suggested by the reviewer, we have corrected the rest of errors. In addition, we have checked the manuscript many times to prevent such errors from happening again.

Changed in the “Revised Manuscript” (Line 15, Page 3):

“reaction rate”

Changed in the “Revised Manuscript” (Line 17, Page 3):

“reaction rate”

Changed in the “Revised Manuscript” (Line 18, Page 7):

“reaction rate”

Changed in the “Revised Manuscript” (Line 3, Page 8):

“reaction rate”

Changed in the “Revised Manuscript” (Line 1, Page 8):

“exploited”

Changed in the “Revised Manuscript” (Line 22, Page 16):

“artificial enzymes”

Author's Response to Reviewer #2

This study describes novel clusterzymes that have preferential enzyme-mimicking catalysis which confers GPx-like and CAT-like activities in neurons and brain that may be relevant for mitigating the damaging effects of TBI in preclinical models. TBI induces secondary injury that includes upregulation of oxidative stress/damage and loss of antioxidant capacity, and this is known to play a role in secondary inflammation and long-term neurodegeneration. As such, inhibiting oxidative stress/damage pathways following TBI are important and rational therapeutic targets to improve outcomes in head injury patients. The identification of novel Au₂₄Cd₁ and Au₂₄Cd₁ enzymes is exciting, and the characterization of their activities in cell based assays suggest that they are more potent antioxidants when compared to standards in the field. Unfortunately, the translation of cell based studies to in vitro and in vivo model systems are problematic. There are many errors in preclinical experimental design and an over-interpretation of the effects of therapeutic Au₂₄Cd₁ and Au₂₄Cd₁ on oxidative stress, inflammation and neurobehavioral recovery. The authors should investigate the clusterzymes using clinically-relevant drug administration protocols (IV, IP, ICV), and improve the preclinical design to include early timepoint relevant to oxidative damage/inflammation. The following points were identified during review of the manuscript and should be addressed by the authors.

1. Description and rigor of CCI model of TBI. The methods for the CCI model lack appropriate details that are essential for preclinical modelling studies. Were the animal studies approved by an animal ethics and use committee (ICAU)? What sex of mice were included in the study? What age of mice? What instrument was used to induce CCI? Details of the instrument and mechanism of impact (pneumatic or electromagnetic) are needed. What was the sham control procedure - did these animals receive craniotomy without impact? What post-surgery care was provided to animals? Were analgesics used after surgery (subcutaneous or topical). Overall, there is a lack in scientific rigor in describing the model, and this is a major issue.

Reply: Thank you very much for precise suggestion. According to the reviewer's suggestion, we focused on the details of the experiment. We followed the methods reported in the literature to construct the TBI model of CCI in mice (*Nat. Commun.* 2016, 7, 11980; *Nat. Med.* 2016, 22, 1335-1341). The CCI surgery was performed in adult male C57BL/6 mice (7-9 weeks, 21-23 g) after 1 week of environment adaption. Animals were induced with controlled cortex impact driven by an electromagnetically CCI injury device (eCCI-6.3, Custom Design & Fabrication, Inc) with an impactor of 5 m/s velocity, 0.61 mm depth, 150 ms duration and 20° angle of dura mater on the vertical axis. Mice were assigned into Sham (n=37/group), TBI, TBI+Au₂₅, TBI+Au₂₄Cu₁, and TBI+Au₂₄Cd₁ group (n=49/group) randomly. Sham-injured groups received the same craniotomy without CCI injury. CCI-induced TBI groups received the same injury and intravenously injected 0.01 M PBS as the vehicle group. Other experimental groups were injured and intravenously injected through the tail vein with clusterzymes after mice were recovered from anesthesia. For the first 2 hours after injury, mice were closely monitored. We have obtained approval of animal ethics and use committee (IRM-DWLL-2019099 and IRM-DWLL-2020079) at our institution. All animal procedures were followed the guideline approved by the Institute of Radiation Medicine, Chinese Academy of Medical Sciences and Peking Union Medical College. As for the analgesics, we did not use after the animal models were built. Ethical standards have been developed and enforced to protect the well-being of animals used for biomedical research. This includes proper pain management following surgical procedures. However, post-surgical analgesics have been shown to be neuroprotective and may confound the evaluation of post-injury outcome measures (*Proc. West. Pharmacol. Soc.* 1998, 41, 241-246; *Am. J. Physiol.* 1994, 267, R665-R672). Conversely, chronic administration of analgesics following TBI has led to worsened outcomes (*Exp. Neurol.* 2006, 201, 301-307). Comprehensive dosing paradigms of common analgesic drugs have not been determined in all experimental TBI models (*Nat. Commun.* 2016, 7, 11980). It is also not well known the degree of interference from drugs, whether subtle or substantial, on molecular, functional and behavioral outcomes. Therefore, analgesics were not administered under consideration of influencing the effect of clusterzymes. The specific details have been improved in the "Supporting Information".

Added in the "Supporting Information" (Lines 10-25, Page 10 and Lines 1-16, Page 11):

“All animal procedures were approved by the Institute of Radiation Medicine, Chinese Academy of Medical Sciences and Peking Union Medical College. Procedures were applied to minimize the number of animals used and the pain mice suffered.

Animal models: Male C57BL/6J mice were purchased from SPF (Beijing) Biotechnology Co., Ltd. Mice were housed in a constant temperature and animal humidity environment with a 12-hour light-dark cycle. Food and water were available *ad libitum*. Surgery was performed after 1 week of transportation to adapt to the environment. Controlled cortical impact (CCI) models were conducted on adult male C57BL/6J mice (7-9 weeks, 21-23 g). C57BL/6 mice were assigned to Sham (n=37/group), TBI, TBI+Au₂₅, TBI+Au₂₄Cu₁, and TBI+Au₂₄Cd₁ groups (n=49/group) randomly. Mice were anesthetized with 10 % chloral hydrate (10 mg/kg) by intraperitoneal injection (*i.p.*). The period of anesthesia was judged through skin pinching reaction and toe stimulation reaction. The surgery was conducted until mice were in the deep anesthesia stage. Mice were fixed in a stereotaxic frame and the scalp was cut to expose the skull. The craniotomy was performed by drilling the skull above the right side of the parietal-temporal cortex in the circle of 2 mm in diameter. The circular lesion (coordinates; between bregma and lambda in the parietal bone centered at 2 mm lateral from the sagittal suture) was produced by exposing the cortex through removing the bone flap. A controlled cortex impact driven by an electromagnetically CCI injury device (eCCI-6.3, Custom Design & Fabrication, Inc) was made with an impactor of 5 m/s velocity, 0.61 mm depth, 150 ms duration and 20° angle of dura mater on the vertical axis. The scalp was sutured together carefully. Then the mice were kept on a heated blanket under control until being recovered from anesthesia. The clusterzymes were intravenously injected into the CCI mice at the dose of 50 mg/kg dissolved in 0.01 M PBS. An injection volume of 200 μL was used for Sham, TBI and other groups. All mice were marked, classified and put into the cages (5 mice/cage) under an SPF level environment. Sham-injured groups (control) received the same craniotomy and intravenously injected 0.01 M PBS without CCI injury. CCI-induced TBI groups received the same injury and intravenously injected 0.01 M PBS as the vehicle group. All animals fully recovered from surgical procedures and gradually gained weight after surgery. Mice were treated with cervical dislocation after finishing related animal experiments. Brain tissues and other organs (heart, liver, spleen, lung, kidney, bladder, and testicular) were taken out on day 1, 3 and 7 post injury. Morris water maze tests were conducted on days 13-18 and 28-32 post injury.”

2. Route of post-injury Au₂₄Cu₁/Au₂₄Cd₁ administration: “the clusterzymes was added to the wound of TBI mice at the concentration of 10 μg/ul”. This description is wholly inadequate and the route of administration is not quantifiable or translational for human studies. Typically in preclinical TBI studies IV, IP, or ICV administration routes are used for neuroprotection analysis of therapeutic drugs. A translationally relevant drug administration route is strongly advised to determine the therapeutic potential of Au₂₄Cu₁/Au₂₄Cd₁.

Reply: Thanks for your constructive and professional suggestions. We agree with reviewer’s comments, we have studied some literatures and have a more comprehensive understanding of preclinical and clinical management. **Therefore, we have redone all the *in vivo* experiments by *i.v.* administration route** to determine the therapeutic potential of Au₂₄Cu₁/Au₂₄Cd₁.

As reviewer mentioned, intravenous (*i.v.*), intraperitoneal (*i.p.*) and intracerebroventricular (*i.c.v.*) are the three main routine methods of clinical administration. The *i.p.* injections are most often applied to animals and are not quite feasible for humans. The *i.c.v.* injections are significantly more invasive than *i.v.*. In previous TBI treatment studies, researchers mostly used the tail vein for administration (*ACS Nano*, 2012, 6(9), 8007-8014; *Nat. Med.* 2016, 22, 1335-1341). Thus, the intravenous injection, most widely known as a non-invasive procedure, has been employed as our administration route for optimal drug application and neuroprotection effect of clusterzymes. In addition, the description of injected dose has also been revised as 50 mg/kg dissolved in 0.01 M PBS. The corresponding descriptions have been corrected in the revised manuscript.

Added in the “Supporting Information” (Lines 6-7, Page 11):

“The clusterzymes were intravenously injected into the CCI mice at the dose of 50 mg/kg dissolved in 0.01 M PBS.”

3. Figure 5A. The timeline schematic of oxidative damage and antioxidant responses in injured brain is misleading because many enzymatic/catalytic events co-exist early after TBI and do not occur in the linear fashion depicted in the schematic. Oxidative damage is much more complex and the timeline schematic is problematic. Critical for the *in vivo* experiments (Fig 5D-G) many of the antioxidant and oxidative stress responses peak early after TBI, usually within 1-3 days post-injury. Therefore, the delayed analysis of these responses at 7 and 14 days post-injury are sub-optimal.

Reply: Thank you for professional advice. We strongly agree with reviewer's idea that the timeline schematic of oxidative damage and antioxidant responses in the injured brain is not in a simple linear fashion and many of the antioxidant and oxidative stress responses reach peaks early after TBI, usually within 1-3 days post-injury (*Nat. Rev. Neurol.* 2017, 13, 171-191; *J. Cerebr. Blood F. Met.* 2011, 31, 658-670). According to reviewer's comments, based on the fact that oxidative damage is much more complex, **we objectively studied oxidative stress-related factors including SOD, MDA, H₂O₂ and GSH/GSSG at early points such as 1, 3 and 7 days post-injury.**

Brain injury usually leads to local-systemic secondary changes besides the primary (physical) injury, which progressively contribute to the more severe neurological outcome. The secondary cascades, which occur minutes to days following injury, provide a therapeutic window intervene, to prevent, or to reduce the extent of the secondary damage (*Neurosurgery* 2018, 82, N9). The brain is quite sensitive to the free radical damage and the high rate of oxidative metabolisms in the brain and elevated levels of polyunsaturated lipids, which are the targets of lipid peroxidation, and renders it particularly vulnerable to oxidative stresses. The generation of free oxygen radicals, hydrogen peroxide and nitric oxide causes excitotoxicity and the endogenous antioxidant system (i.e., glutathione peroxidase and superoxide dismutase) aims to convert or neutralize the ROS to the less toxic derivatives, preventing damage to DNA, RNA or proteins (*Nature* 2014, 505, 223-228; *J. Neurotraum.* 2000, 17, 871-890). As shown in **Figure 5i-l**, the indicators of MDA, H₂O₂, SOD and GSH/GSSG in the TBI group were relatively severe 1 day post injury, but were slightly alleviated 3 days post injury, and further improved slightly 7 days post injury. After treatment of Au₂₄Cu₁ and Au₂₄Cd₁, these oxidative stress-related indicators basically recovered to the Sham levels at day 7, indicating that the clusterzymes have a potential therapeutic effect on TBI mice. The corresponding experiment details, results and discussion have all been added in the revised manuscript.

Changed in the “Supporting Information” (Lines 23-25, Page 11):

“Oxidative stress and inflammatory levels: At day 1, 3 and 7 after the TBI or Sham operation (n=5 per group), mice were cleaned with 10 mL PBS perfusion and brain samples were rapidly harvested.”

Changed in the “Revised Manuscript” (Lines 21-23, Page 13 and Lines 1-7, Page 14):

“Meanwhile, mouse models of traumatic brain injury (TBI) were used to examine the *in vivo* effects of clusterzymes. As shown in **Figure 5i-l**, the indicators of MDA, H₂O₂, SOD and GSH/GSSG in the TBI group are relatively severe at day 1 post injury, but were slightly alleviated 3 days post injury, and further improved slightly 7 days post injury. Therefore, the decrease in SOD and GSH/GSSG levels from TBI can be well rescued by clusterzymes with prominent recoveries 7 days after treatment (**Figure 5i** and **j**). Comparatively, Au₂₄Cd₁ induced a better recovery in SOD and GSH than Au₂₄Cu₁, which correlates well with their *in vitro* SOD-like activity (**Figure 3**). As the byproducts of the oxidative stress, lipid peroxides and H₂O₂ showed higher accumulations in the brain following TBI, resulting in severe oxidative damage (**Figure 5k and l**).”

Changed in the “Revised Manuscript” (Figure 5, Page 28):

Figure 5. Oxidative stress levels *in vitro* and *in vivo* before and after treatment of clusterzymes. **a** HT22 cell viability of clusterzymes. **b** HT22 cell viability in the presence of H_2O_2 with or without treatment of clusterzymes as determined by MTT assays. Fluorescence quantification of cell staining for **c** and **e** $\cdot OH$ and **d** and **f** $O_2^{\cdot -}$ by flow cytometry. Fluorescence microscopic images of intracellular **g** $\cdot OH$ (green) and **h** $O_2^{\cdot -}$ (red) levels induced by 100 μM H_2O_2 with or without clusterzymes by HPF and DHE probes, respectively. It can be seen that $Au_{24}Cu_1$ has a better scavenging ability for $\cdot OH$, but $Au_{24}Cd_1$ shows better specificity for $O_2^{\cdot -}$, suggesting their individual selectivity for $\cdot OH$ and $O_2^{\cdot -}$ respectively. **i-l** Indicators for oxidative stress including SOD, GSH/GSSG, MDA and H_2O_2 of TBI mice with or without treatment of clusterzymes 1, 3, and 7 days post injury ($n=5$ per group). Data are presented as mean \pm SEM; * $P < 0.05$, ** $P < 0.01$, and *** $P < 0.001$ compared with the Sham group, analyzed by ANOVA.

4. OH^{\cdot} and $O_2^{\cdot -}$ should be quantified in the *in vitro* experiments using HT22 neuronal cell cultures. Does reduced oxidative stress due to $Au_{24}Cu_1$ and $Au_{24}Cd_1$ treatment result in reduced neuronal cell death in these experiments? Viability data should be include to correlate reduced oxidative stress with neuronal health in the H_2O_2 assays.

Reply: Thanks for your suggestions. With reference to reviewer's point, relative quantitative analyses were conducted by a flow cytometer (BD AccuriTM C6). The results have been added in the updated **Figure 5c-f**, showing that excessive amount of $\cdot OH$ and $O_2^{\cdot -}$ produced by H_2O_2 stimulation can be significantly reduced with the clusterzyme treatment. Meanwhile, the $Au_{24}Cd_1$ clusterzyme has a preference for scavenging $O_2^{\cdot -}$ while the $Au_{24}Cu_1$ clusterzyme prefers to scavenge $\cdot OH$, which corresponds well to the results of fluorescence microscopic images

(**Figure 5g** and **h**). Moreover, per reviewer's suggestion, the cell viability of H₂O₂-stimulated neuron cells has been performed with the incubation of Au₂₅, Au₂₄Cd₁ or Au₂₄Cu₁ to determine the neuroprotective effect of clusterzymes. Under stimulation of 100 μM H₂O₂, the cell viability decreases approximately to 75% while the clusterzyme treatment could improve the viability of neuron cells (**Figure 5b**). These results indicate that the reduced oxidative stress shows a close correlation with the neuronal health. The corresponding experimental details, results and discussion have been added in the revised manuscript.

Added in the “Supporting Information” (Lines 16-21, Page 9):

“Cell viability: HT22 cells (2×10³) were cultured in the 96-well plate in 100 μL culture media. When reaching roughly 60 % confluence in each well, cells were stimulated by 100 μM H₂O₂ for 6 hours. Then the culture media were substituted by fresh media containing Au₂₅, Au₂₄Cu₁ or Au₂₄Cd₁ at different doses and cells were incubated overnight. The plates were washed with PBS and incubated with 5 mg/mL MTT for 2.5 hours. Cell viability was determined by MTT assay and analyzed at OD 490 nm.”

Changed in the “Supporting Information” (Lines 5-8, Page 10):

“Intracellular oxidative stress was captured by using a fluorescence microscope (EVOS, AMG) and quantitative analysis of free radical was conducted by a FACS flow cytometer (BD Accuri™ C6).”

Changed in the “Revised Manuscript” (Lines 9-21, Page 13):

“To reveal the biological activity of clusterzymes, the cell toxicity for different nerve cell lines (HT22, BV2 and MA-c) was measured by the MTT assay (**Figure 5a** and **S25**), showing that Au₂₅, Au₂₄Cu₁, and Au₂₄Cd₁ present acceptable biocompatibility. Cell survival of H₂O₂-stimulated neuron cells was performed with the incubation of Au₂₅, Au₂₄Cd₁ or Au₂₄Cu₁. As shown in **Figure 5b**, the clusterzyme treatment could improve the viability of neuron cells. To explore the correlation between the oxidative stress and the neuron viability, ROS, especially •OH and O₂^{•-}, were quantified and detected by a FACS flow cytometer and a fluorescence microscope using hydroxyphenyl fluorescein solution (HPF) and dihydroethidium (DHE) fluorescence probes, respectively. The H₂O₂ stimulation significantly elevates the fluorescence signal, indicating the presence of excessive amount of •OH and O₂^{•-} (**Figure 5c-h**). All clusterzymes decrease the ROS signals, with Au₂₄Cu₁ showing the best clearance efficiency against •OH (**Figure 5c, e** and **g**) and Au₂₄Cd₁ displaying the beset clearance capability for O₂^{•-}, suggesting their individual selectivity (**Figure 5d, f** and **h**).”

Changed in the “Revised Manuscript” (Figure 5, Page 28):

Figure 5. Oxidative stress levels *in vitro* and *in vivo* before and after treatment of clusterzymes. **a** HT22 cell viability of clusterzymes. **b** HT22 cell viability in the presence of H₂O₂ with or without treatment of clusterzymes as determined by MTT assays. Fluorescence quantification of cell staining for **c** and **e** •OH and **d** and **f** O₂^{•-} by flow cytometry. Fluorescence microscopic images of intracellular **g** •OH (green) and **h** O₂^{•-} (red) levels induced by 100 μM H₂O₂ with or without clusterzymes by HPF and DHE probes, respectively. It can be seen that Au₂₄Cu₁ has a better scavenging ability for •OH, but Au₂₄Cd₁ shows better specificity for O₂^{•-}, suggesting their individual selectivity for •OH and O₂^{•-} respectively. **i-l** Indicators for oxidative stress including SOD, GSH/GSSG, MDA and H₂O₂ of TBI mice with or without treatment of clusterzymes 1, 3, and 7 days post injury (n=5 per group). Data are presented as mean ± SEM; *P < 0.05, **P < 0.01, and ***P < 0.001 compared with the Sham group, analyzed by ANOVA.

5. Fig 5D-G, as mentioned before earlier time points at peak of oxidative damage should be included in these experiments. What N was used in the *in vivo* experiments? It is not clear if there are statistical differences between Au₂₄M₁ treated and vehicle control treated TBI groups. Further, a student's t-test is not appropriate for these comparisons. At a minimum one-way ANOVA should be used, or preferentially two-way ANOVA to capture effect of Time post-injury.

Reply: Thank you very much for the constructive suggestion. As mentioned above in Reply 3, we have added earlier time points (1, 3, and 7 days post injury) at the peak of oxidative damage according to reviewer's suggestions. Groups were assigned into Sham, TBI, Au₂₅, Au₂₄Cd₁ and Au₂₄Cu₁. CCI models were conducted on adult male C57BL/6J mice (7-9 weeks, 21-23 g) and the total number of CCI model mice was 233. Injured mice were utilized in the *in vivo* experiments including oxidative stress and inflammatory levels using brain homogenate supernatant and serum

(n=5 per group), western blotting analysis using brain homogenate supernatant (n=3 per group), immunostaining including immunofluorescence and immunohistochemistry staining (n=3 per group). BBB penetration was applied to detect the Au element in injured mice brain at 1, 4, 12, and 24 h post injection (n=3 per group per point). Morris water maze (MWM) performance was assessed on days 13-18 and 28-32 following injury (n=7 per group). In addition, normal C57BL/6J mice were intravenously injected with clusterzymes to study the pharmacokinetics (n=3 per group) and toxicology (n=3 per group). We have added the N used in the *in vivo* experiments in the revised manuscript. Per reviewer's suggestion, the updated data in the revised manuscript are presented as mean \pm standard error of the mean (SEM), *P < 0.05, **P < 0.01, and ***P < 0.001 versus the Sham group, analyzed by one-way analysis of variance (ANOVA). **Figure 5i-l** clearly show that there are statistical differences between Au₂₄Cd₁- and Au₂₄Cu₁-treated groups and the vehicle control group.

Changed in the “Supporting Information” (Lines 13-17, Page 15):

“Statistic methods. Data are presented as mean \pm standard deviation (SD) or standard error of the mean (SEM). Comparison of means between two groups was accomplished by the Student's t-test. For multiple comparison, one-way analysis of variance (ANOVA) were used to assess difference in means among groups. *P < 0.05, **P < 0.01, and ***P < 0.001 versus with the Sham group, analyzed by ANOVA.”

6. *Western blot quantification for IL-1b, IL-6 and TNF should be included in Fig 6B,C. For IL-1b, expression levels are very high in sham (control) and there is no induction of IL-1b following TBI. Also, is this mature IL-1b in Western or pro-forms? The authors should include the full uncropped blot for this analysis. The IL-6 Western blot has been cut/stitched using photoshop or equivalent software. Full unmodified blots need to be included for scientific rigor.*

Reply: Thank you very much for the constructive suggestion. With regard to reviewer's advice above, Western blot has been performed on day 1, 3, and 7 and the related quantifications for IL-1 β , IL-6, and TNF α have also been included in the updated **Figure 6a-d**. The IL-1 β antibody used for Western blot is purchased from Abcam, ab234437, so the IL-1 β in Western blot is pro-forms. Here, we have also provided the full uncropped Western blots in the revised manuscript. As shown in **Figure 6a-d**, the expression levels of IL-1 β and IL-6 are significantly upregulated following TBI compared with the sham 1 day post injury, indicating strong inflammation in the brain. Au₂₄Cd₁ sharply downregulates IL-1 β and IL-6 levels, suggesting the anti-neuroinflammation effect. In comparison, Au₂₅ only shows minor downregulation. Similarly, TBI leads to significant upregulation of TNF α 1 day post injury, but Au₂₄Cu₁ can significantly downregulate the expression of TNF α , presenting superior efficacy over Au₂₅ and Au₂₄Cd₁ (**Figure 6a**). Although the expressions of these inflammation cytokines induced by TBI are gradually suppressed by autoimmunity in the vehicle control group over time, especially at day 7 post injury, there are still significant differences in IL-1 β and IL-6 levels between the Sham and TBI groups (**Figure 6b-d**). However, the clusterzyme treatment results in cytokine levels closed to the normal control, indicating a better suppression effect on neuroinflammation. The corresponding experiment details, results and discussion have been added and changed in the revised manuscript.

Changed in the “Revised Manuscript” (Lines 20-23, Page 14; Lines 1-8, Page 15):

“From the Western blots and the relevant quantification analysis (**Figure 6a and d**), the expression levels of IL-1 β and IL-6 are significantly upregulated 1 day following TBI, indicative of strong local inflammations. Au₂₄Cd₁ sharply downregulates IL-1 β and IL-6 levels, suggesting the anti-neuroinflammation effect. In comparison, Au₂₅ only shows minor downregulation. Similarly, TBI leads to significant upregulation of TNF α 1 day post injury, but Au₂₄Cu₁ can significantly downregulate the expression of TNF α , presenting superior efficacy over Au₂₅ and Au₂₄Cd₁ (**Figure 6a**). Although the expressions of these inflammation cytokines induced by TBI are gradually suppressed by autoimmunity in the vehicle control group over time, especially at day 7 post injury, there are still significant differences in IL-1 β and IL-6 levels between the Sham and TBI groups (**Figure 6b-d**). However, the clusterzyme

treatment results in cytokine levels closed to the normal control, indicating a better suppression effect on neuroinflammation.”

Changed in the “Revised Manuscript” (Figure 6, Page 29):

Figure 6. Inflammation levels in brain tissues after clusterzyme treatment. a-c Western blotting for IL-1β, IL-6, and TNFα in the brain tissues 1, 3, and 7 days post TBI after treatment (n=3 per group), respectively. d Western blotting quantitative analysis of inflammatory factors at different time points. It can be seen that Au₂₄Cd₁ can rapidly and significantly reduce the upregulated inflammatory cytokines of IL-1β and IL-6 after brain injury, while Au₂₄Cu₁ has a better ability to reduce the expression of TNFα. e-g ELISA quantitative analysis of IL-1β, IL-6, and TNFα levels in brain tissues on day 1, 3, and 7 with or without clusterzyme treatment (n=5 per group), respectively. h Immunofluorescence co-staining of IL-1β and microglia (Iba-1), astrocytes (GFAP) or neurons (NeuN) in injured cortex 3 days post injury with or without clusterzyme treatment. Quantitative analysis of i the positive cell numbers and j IL-1β expression with positive cells in the injured cortex with or without clusterzyme treatment (n=3 per group). Data are mean ± SEM; *P < 0.05, **P < 0.01, and ***P < 0.001 compared with the Sham group, as analyzed by ANOVA.

7. The chronic expression of IL-1b, IL-6 and TNF is surprising following CCI. These proinflammatory mediators are upregulated early after TBI and do not often persist at more chronic timepoints (7 and 14d post-TBI). Analysis at earlier time point would be optimal. Statistical analysis is problematic in fig 6., and ANOVA is required. Does Au₂₄Cu₁ and Au₂₄Cd₁ treatment reduce expression of cytokines in the injured cortex?

Reply: Thanks for constructive advice. TBI has been linked to the post-traumatic stress disorder, memory deficits, chronic traumatic encephalopathy, and chronic neuroinflammation. As a result, tumor necrosis factors (TNF), IL-6, IL-1β are rapidly upregulated in local glial cells and represent early effectors that drive post-traumatic neuroinflammation (*Nat. Rev. Neurol.* 2017, 13, 171-191). Usually, cytokines and chemokines reach peaks at day 1-

2, and declines at day 2-3 in cerebrospinal fluid (CSF) or extracellular (ECF) (*J. Cerebr. Blood F. Met.* 2011, 31, 658-670). We totally agree with reviewer's point that these proinflammatory mediators are upregulated promptly right after TBI and do not often persist at later time points (7 and 14 days post TBI). With regard to reviewer's advice above, Western blot and ELISA have been carried out on day 1, 3, and 7 with the corresponding quantifications of IL-1 β , IL-6 and TNF α to be included in the updated **Figure 6a-g**. These inflammatory cytokines are upregulated rapidly 1 day after brain injury. The quantification results of Western blot and ELISA show that the expressions of inflammatory cytokines are noticeably decreased with Au₂₄Cu₁ and Au₂₄Cd₁ treatment compared with the TBI group on day 1, further reduced on day 3 and almost recovered to normal on day 7. In addition, statistical analysis has also been updated in the revised manuscript. Data are presented as mean \pm standard error of the mean (SEM), one-way analysis of variance (ANOVA) were employed in our study, *P < 0.05, **P < 0.01, and ***P < 0.001 versus the Sham group.

Moreover, in order to observe the expression of cytokines in the injured cortex, immunohistochemistry (IHC) staining has also been conducted on day 1, 3 and 7 post injury as shown in **Figure S29-31**. The quantitative analysis of IHC staining demonstrates that the expression of inflammatory cytokines gradually decreased after treatment of Au₂₄Cu₁ and Au₂₄Cd₁ (**Figure S32**), indicating Au₂₄Cu₁ and Au₂₄Cd₁ clusterzymes contribute to alleviation of neuroinflammation. The corresponding experimental details, results and discussion have been changed in the revised manuscript.

Changed in the “Supporting Information” (Lines 23-25, Page 11):

“Oxidative stress and inflammatory levels: On day 1, 3, and 7 after TBI or Sham operation (n=5 per group), mice were cleaned with 10 mL PBS perfusion and brain samples were rapidly harvested....”

Changed in the “Supporting Information” (Lines 13-17, Page 15):

“Statistic methods. Data are presented as mean \pm standard deviation (SD) or standard error of the mean (SEM). Comparison of means between two groups was accomplished by the Student's t-test. For multiple comparison, one-way analysis of variance (ANOVA) were used to assess difference in means among groups. *P < 0.05, **P < 0.01, and ***P < 0.001 versus with the Sham group, analyzed by ANOVA.”

Changed in the “Revised Manuscript” (Lines 20-23, Page 14; Lines 1-23, Page 15; Line 1, Page 16):

“From the Westen blots and the relevant quantification analysis (**Figure 6a and d**), the expression levels of IL-1 β and IL-6 are significantly upregulated following TBI 1 day post injury, indicative of strong local inflammations. Au₂₄Cd₁ sharply downregulates IL-1 β and IL-6 levels, suggesting the anti-neuroinflammation effect. In comparison, Au₂₅ only shows minor downregulation. Similarly, TBI leads to significant upregulation of TNF α 1 day post injury, but Au₂₄Cu₁ can significantly downregulate the expression of TNF α , presenting superior efficacy over Au₂₅ and Au₂₄Cd₁ (**Figure 6a**). Although the expressions of these inflammation cytokines induced by TBI are gradually suppressed by autoimmunity in the vehicle control group over time, especially 7 days post injury, there are still significant differences in IL-1 β and IL-6 levels between the Sham and TBI groups (**Figure 6b-d**). However, the clusterzyme treatment results in cytokines close to the normal level, indicating a better suppression effect on neuroinflammation. Further, the ELISA further validated the immunoblotting results that Au₂₄Cd₁ and Au₂₄Cu₁ are capable of decreasing the inflammatory cytokines in brain tissues such as IL-1 β , IL-6, and TNF α , while Au₂₅ does not significantly alter the inflammatory cytokine patterns (**Figure 6e-g**). Au₂₄Cd₁ can eliminate IL-1 β and IL-6 associated inflammatory responses, while Au₂₄Cu₁ has a better effect on reduction of TNF α , indicating their relevant selectivity towards modulation of neuroinflammation.... Meanwhile, the morphology of TBI-activated astrocytes can be recovered to near-normal shapes after treatment with clusterzymes (**Figure S28**), and the neuroinflammatory responses are also prevented likewise by verified histology (**Figure S29-32**).”

Changed in the “Revised Manuscript” (Figure 6, Page 29):

Figure 6. Inflammation levels in brain tissues after clusterzyme treatment. **a-c** Western blotting for IL-1 β , IL-6, and TNF α in the brain tissues 1, 3, and 7 days post-TBI after treatment (n=3 per group), respectively. **d** Western blotting quantitative analysis of inflammatory factors at different time points. It can be seen that Au₂₄Cd₁ can rapidly and significantly reduce the upregulated inflammatory cytokines of IL-1 β and IL-6 after brain injury, while Au₂₄Cu₁ has a better ability to reduce the expression of TNF α . **e-g** ELISA quantitative analysis of IL-1 β , IL-6, and TNF α levels in brain tissues on day 1, 3, and 7 with or without clusterzyme treatment (n=5 per group), respectively. **h** Immunofluorescence co-staining of IL-1 β and microglia (Iba-1), astrocytes (GFAP) or neurons (NeuN) in injured cortex 3 days post injury with or without clusterzyme treatment. Quantitative analysis of **i** the positive cell numbers and **j** IL-1 β expression with positive cells in the injured cortex with or without clusterzyme treatment (n=3 per group). Data are mean \pm SEM; * P < 0.05, ** P < 0.01, and *** P < 0.001 compared with the Sham group, as analyzed by ANOVA.

Changed in the “Supporting Information” (Figure S29-32, Pages 44-47):

Figure S29. Immunohistochemistry of IL-6, IL-1 β and TNF α in brain tissues on day 1 post injury. Red arrows refer to glial cells that express inflammatory cytokines. Scale bar: 50 μ m.

Figure S30. Immunohistochemistry of IL-6, IL-1 β and TNF α in brain tissues on day 3 post injury. Red arrows refer to glial cells that express inflammatory cytokines. Scale bar: 50 μ m.

Figure S31. Immunohistochemistry of IL-6, IL-1 β and TNF α in brain tissues on day 7 post injury. Red arrows refer to glial cells that express inflammatory cytokines. Scale bar: 50 μ m.

Figure S32. Quantitative analysis of immuno-histochemical results including IL-6, IL-1 β and TNF α on day 1, 3, and 7 post injury (n=3 per group)

8. What time post-injury is IL-1 β and IL-6 imaging in Fig 6G? Quantification is needed for this analysis, and colocalization studies with markers for neurons (NeuN), microglia (Tmem119) or astrocytes (GFAP) is recommended to determine what cell type produces these pro-inflammatory mediators after TBI.

Reply: Thanks for your careful review and constructive suggestion. Considering the advice given above, we have applied immunofluorescence staining for co-localization of markers for neurons (NeuN), microglia (Iba1) and astrocytes (GFAP) to determine the production of these pro-inflammatory mediators on day 3 post injury. Quantification analyses of co-expression cell types and inflammatory cytokines have been performed as well. We should note that due to COVID-19, Tmem119 mentioned above by reviewer has not been purchased and Iba1 probe was adopted to label microglia in the revised manuscript (*Nat. Commun.* 2019, 10, 5156). As shown in revised **Figure 6h, i** and **S26-27**, IL-1 β , IL-6 and TNF α prefer to be secreted by microglia, superior to astrocytes and neurons. This result is similar to those that have been well documented (*Glia*, 2017, 65, 1423-1438; *J. Neuroimmunol.* 1991, 33, 227-236; *J. Neuroimmunol.* 2017, 14, 143; *Exp. Neurol.* 1998, 152, 74-87). In addition, quantitative analyses on the number of positive cells show that massive microglia and astrocytes are activated and many neurons are depleted after TBI (**Figure 6j**). With the clusterzyme treatment, the number of all these nerve cells is mostly rescued. Meanwhile, the morphology of TBI-activated astrocytes can be recovered to almost regular shapes after treatment with clusterzymes (**Figure S28**). The corresponding experimental details, results and discussion have been added and changed in the revised manuscript.

Changed in the “Supporting Information” (Lines 11-25, Page 12; Lines 1-6, Page 13):

“Immunostaining: Mice brain samples were fixed in 4% paraformaldehyde (PFA) for 24-48 hours, embedded in paraffin and mounted on slides (4 μ m coronal sections). The tissue slices were dewaxed twice in xylene for 10 min and 5 min, and dehydrated in a gradient ethanol solution (100 %, 95 %, 80 % and 70 %). Slices were rinsed three times and 2 min each with PBS. Antigen retrieval was performed in citrate antigen retrieval solution (C1032, Solarbio, China) at 95 °C for 10 min in a pressure cooker. After cooling naturally in the retrieval solution, slices were rinsed with PBS, followed by blocking with 5% BSA at room temperature for 2 hours, and the excess liquid was shaken off. The primary antibodies of target cells and cytokines were added at 4 °C overnight. The primary antibody information is as followed: Anti-TNF α antibody (1:150, Abcam, ab183218), anti-IL6 antibody (1:200, Bioss, bs-0782R), anti-IL 1 β (1:200, Bioss, bs-0812R), antibody anti-NeuN antibody (1:800, GeneTex, GTX00837), anti-Iba1 antibody (1:300, Abcam, ab48004), and anti-GFAP (1:400, Abcam, ab90601). The primary antibody was removed and slices were rinsed with PBS three times 3 min each. Indicated fluorescence labeled secondary antibodies were added and incubated at room temperature for 1h in dark. The secondary antibody information is as follows: CoraLite488-conjugated Affinipure Donkey Anti-Rabbit IgG (H+L) (1:500, Proteintech, SA00013-6), Goat Anti-Chicken IgY H&L (Alexa Fluor® 647) (1:1000, Abcam, ab150171), Donkey Anti-Sheep IgG H&L (Alexa Fluor® 647) (1:1000, Abcam, ab150179) and Donkey Anti-Goat IgG H&L (Alexa Fluor® 647) (1:1000, Abcam, ab150131). Finally, slices were mounted with anti-fade mounting medium with DAPI (S2110, Solarbio, China) and photographed with fluorescence microscope (EVOS, AMG).”

Changed in the “Revised Manuscript” (Lines 13-23, Page 15):

“Finally, immunofluorescence staining of cerebral cortex harvested from mice also shows that Au₂₄Cd₁ and Au₂₄Cu₁ can remarkably decrease the TBI-elevated expressions of IL-1 β , IL-6, and TNF α (**Figure 6h, i and S26-27**), therefore alleviating neuroinflammation. Colocalization studies with markers for neurons (NeuN), microglia (Iba-1) or astrocytes (GFAP) were performed in injured cortex on day 3 post injury. **Figure 6h** and **i** reveal that IL-1 β is mainly produced by microglia after TBI, similar with IL-6 and TNF α (**Figure S26-27**). In addition, quantitative analyses of the number of positive cells show that massive microglia and astrocytes are activated and many neurons are depleted after TBI (**Figure 6j**). With the clusterzyme treatment, the number of these nerve cells is mostly rescued. Meanwhile, the morphology of TBI-activated astrocytes can be recovered to near normal levels after treatment with clusterzymes (**Figure S28**)”

Changed in the “Revised Manuscript” (Figure 6, Page 29):

Figure 6. Inflammation levels in brain tissues after clusterzyme treatment. **a-c** Western blotting for IL-1β, IL-6, and TNFα in the brain tissues 1, 3, and 7 days post TBI after treatment (n=3 per group), respectively. **d** Western blotting quantitative analysis of inflammatory factors at different time points. It can be seen that Au₂₄Cd₁ can rapidly and significantly reduce the upregulated inflammatory cytokines of IL-1β and IL-6 after brain injury, while Au₂₄Cu₁ has a better ability to reduce the expression of TNFα. **e-g** ELISA quantitative analysis of IL-1β, IL-6, and TNFα levels in brain tissues on day 1, 3, and 7 with or without clusterzyme treatment (n=5 per group), respectively. **h** Immunofluorescence co-staining of IL-1β and microglia (Iba-1), astrocytes (GFAP) or neurons (NeuN) in injured cortex 3 days post injury with or without clusterzyme treatment. Quantitative analysis of **i** the positive cell numbers and **j** IL-1β expression with positive cells in the injured cortex with or without clusterzyme treatment (n=3 per group). Data are mean ± SEM; **P* < 0.05, ***P* < 0.01, and ****P* < 0.001 compared with the Sham group, as analyzed by ANOVA.

Changed in the “Supporting Information” (Figures S26-S27, Pages 41-42):

Figure S26. a Immunofluorescence co-staining of IL-6 and microglia (Iba-1), astrocytes (GFAP) or neurons (NeuN) in injured cortex 3 days post injury with or without clusterzyme treatment. Quantitative analysis of b the positive cell number and c IL-6 expression with positive cells in injured cortex with or without clusterzymes treatment (n=3 per group). Data are mean \pm SEM; *P < 0.05, **P < 0.01, and ***P < 0.001 versus the Sham group, as analyzed by ANOVA.

Figure S27. a Immunofluorescence co-staining of TNF α and microglia (Iba-1), astrocytes (GFAP) or neurons (NeuN) in injured cortex at 3 days post-injury with or without clusterzyme treatment. Quantitative analysis of b the positive cell number and c TNF α expression with positive cells in injured cortex with or without clusterzymes treatment (n=3 per group). Data are mean \pm SEM; *P < 0.05, **P < 0.01, and ***P < 0.001 versus the Sham group, as analyzed by ANOVA.

9. The wound healing analysis in Fig S24 is not useful. The data are subjective and it is not known how healing at surgery site related to brain injury response and recovery. Please remove this.

Reply: Thank you very much for the kindly suggestion. We agree with reviewer's that the connection between the recovery process of brain injury and the healing at the surgery site is vague, so the degree of brain injury repair could not be evaluated by the wound healing. Therefore, in accordance with reviewer's suggestion, we have deleted the relevant data of the wound healing analysis in the revised manuscript.

10. Astrocyte analysis in Fig S25 needs to be quantified because morphological transformations of GFAP astrocytes occur after TBI. Number of GFAP+ astrocytes and morphological characteristics should be measured. What region of the injured brain was analysed, and at what time post-injury?

Reply: Thank you very much for the professional suggestion. Astrocytes are among the most abundant cell types in the central nervous system, which can respond to the brain injury through hypertrophy and proliferation occurring correlated with the severity of the injury (*Brain*, 2006, 129, 2761-2772; *Nat. Rev. Neurol.* 2017, 13, 171-191). Therefore, studying the morphology and the number of astrocytes is important for detecting the therapeutic effect of TBI. We selected mice before and after treatment 3 days post injury, and localized astrocytes by immunofluorescence staining. As can be seen from the magnified astrocyte cell morphology of **Figure S28a**, the astrocytes in the damaged area and the surrounding area of the cerebral cortex of TBI mice occurs hypertrophy, and the expression is significantly up-regulated compared with the Sham group. After the clusterzyme treatment, both the number and the morphology of astrocytes have been restored, close to the Sham group. The quantitative analysis of astrocytes further supports this conclusion (**Figure S28b**). The corresponding changes in the text and figures have been updated in the revised manuscript.

Changed in the "Revised Manuscript" (Lines 10-15, Page 16):

“In addition, quantitative analyses of the number of positive cells show that massive microglia and astrocytes are activated and many neurons are depleted after TBI (Figure 6j). With the clusterzyme treatment, the number of these nerve cells mostly is retrieved. Meanwhile, the morphology of TBI-activated astrocytes can be recovered to near normal levels after treatment with clusterzymes (Figure S28)”

Changed in the “Supporting Information” (Figure S28, Page 43):

Figure S28. a Astrocyte activation levels in the cerebral cortex by GFAP immunofluorescence staining in TBI mice with or without clusterzyme treatment. Insets show a magnified morphology of astrocyte cells (n=3 per group). **b** Quantitative analysis of the number of astrocytes.

11. Inflammatory infiltrates at 30 days post-injury is also non-specific and does not add value. Peripheral immune cells (monocytes, B and T cells) are unlikely to still be trafficking to the brain at this time point due to closing of the BBB. If immune infiltration is important and relevant in this study then specific analyses using flow cytometry or related assays are needed.

Reply: Thank you very much for the professional suggestions. It is known that inflammation after TBI is now recognized to involve a robust and complex interaction between central and peripheral cells, which occurs by way of the blood-brain barrier (BBB) that becomes disrupted due to injury in the early stage (*Neuron* 2014, 22, 229–48). Acute neuroinflammation starts by early resident microglial activation. Chemoattractants released from pro-inflammatory microglia promote peripheral immune cell infiltration into the injury site (*Neural. Plast.* 2017, 2017, 5405104; *JAMA Neurol.* 2015, 72, 355-362; *Nat. Rev. Neurol.* 2017, 13, 171-191). On day 30 post injury, the BBB has been closed (*Neurosci. Lett.* 2013, 551, 23-27; *J. Neuroinflamm.* 2015, 12, 223-223; *PLoS ONE* 2014, 9, 98143) and thus peripheral immune cells (monocytes, B and T cells) are unlikely to still be trafficking to the brain. The graph shown in **Figure S26 in the previous manuscript** using H&E staining was intended to demonstrate the low toxicity of our clusterzymes in the mouse brain. Per reviewer’s suggestion, we have updated the **Figure S36** in the revised manuscript. H&E staining shows that the clusterzymes are barley toxic to all major organs of mice including brain, heart, liver, lung, spleen, kidneys, bladder, and testicles.

Added in the “Supporting Information” (Lines 19-25, Page 14; Lines 1-12, Page 15):

“Pharmacokinetic and toxicological studies: ... Hematological and biochemical panels were analyzed on day 7 post injection. Blood samples were taken from the retro-orbital sinus in K2EDTA tubes. Biochemical blood samples were

left to stand for 30 min and then centrifuged at 3500 rpm for 15 min twice. All organs were collected and fixed in 4 % PFA for 24-48 hours, embedded in paraffin and mounted on slides (4- μ m coronal sections). Slides were stained through hematoxylin-eosin (H&E) staining to observe the toxicity of clusterzymes in each organ including heart, liver, spleen, lung, kidney and brain.”

Changed in the “Supporting Information” (Figure S36, Page 51):

Figure S36. Histology of major organs in mice (heart, liver, spleen, lung, kidney, bladder and testis) treated with 200 μ L clusterzymes at the concentration of 5 mg/mL after 7 days. Scale bar is 50 μ m, n=3 per group. No significant toxic responses were found in all organs.

12. Higher magnification images of IL-6 are needed in Fig S27. Is IL-6 expressed in neurons? Negative controls without primary antibody should be included to demonstrate specificity of antibody in IHC studies.

Reply: Thank you very much for the constructive suggestions. IHC staining was utilized to observe the expression of inflammatory cytokines. Since we altered the administration route following other reviewers’ advice, new IHC staining was performed. Per reviewer’s suggestion, negative controls without primary antibody were also included to demonstrate specificity of the antibody in IHC studies (**Figure R2**). Due to the specificity of secondary antibodies, negative controls without primary antibodies did not show the expression of inflammatory cytokines (**Figure R2**). IHC high magnification images have been improved and presented in the revised manuscript (**Figure S29-31**). Quantitative analysis of IHC results in **Figure S32** shows that the expression of TNF α , IL-6 and IL-1 β rapidly increases in TBI group. With Au₂₄Cu₁ and Au₂₄Cd₁ treatment, the expression of TNF α , IL-6 and IL-1 β is noticeably decreased on day 1, further downregulated on day 3 and almost recovered to normal levels on day 7, which indicates that Au₂₄Cu₁ and Au₂₄Cd₁ clusterzymes have potentials to relieve inflammation induced by brain injury. From dual immunofluorescence staining analyses (**Figure 6h**), it is found that TNF α , IL-6 and IL-1 β are mostly expressed in the microglia instead of neurons, which is consistent with literatures (*Glia* 2017, 65, 1423-1438; *J. Neuroimmunol.* 1991, 33, 227-236; *J. Neuroimmunol.* 2017, 14, 143; *Exp. Neurol.* 1998, 152, 74-87). The corresponding experimental details, results and discussion have been updated in the revised manuscript.

Figure R2. Immunohistochemical photographs of negative controls without primary antibodies.

Changed in the “Supporting Information” (Figure S29-32, Pages 44-47):

Figure S29. Immunohistochemistry of IL-6, IL-1 β and TNF α in brain tissues on day 1 post injury. Red arrows refer to glial cells that express inflammatory cytokines. Scale bar: 50 μ m.

Figure S30. Immunohistochemistry of IL-6, IL-1 β and TNF α in brain tissues on day 3 post injury. Red arrows refer to glial cells that express inflammatory cytokines. Scale bar: 50 μ m.

Figure S31. Immunohistochemistry of IL-6, IL-1 β and TNF α in brain tissues on day 7 post injury. Red arrows refer to glial cells that express inflammatory cytokines. Scale bar: 50 μ m.

Figure S32. Quantitative analysis of immuno-histochemical results including IL-6, IL-1 β and TNF α on day 1, 3, and 7 post injury (n=3 per group).

13. The MWM protocol used in these experiments is not standard and it is difficult to follow the analysis. The authors need to clearly state the N number per group used in this experiment. The error bars are large – are they SEM or SD? It appears that day 5 analysis is the probe trial without platform. It should not be presented with latency data in Fig S28A,B. A major concern about this analysis is that the mice do not appear to be learning the task during the acquisition phase (Fig S28B). All mice have latency times at 60-70sec on day 1 and latency to find hidden platform is not reduced on each subsequent day/trial. Moreover, the effect of TBI is absent, and the Sham (control) mice have equal latency over all testing days as TBI mice. This is a major issue and invalidates the MWM test. The frequency analysis in each quadrant (S28C, D, E) is not standard, and although it shows trends in improvements with Au₂₄Cu₁ or Au₂₄Cd₁ treatment it is not convincing. A proper analysis of search strategy and traditional probe trial (% time in missing platform quadrant) is needed. Also, was a visual acuity trial performed to rule out impairments in visual function required for this task?

Reply: Thank you very much for the constructive suggestion. Details are not included in our previous work. We have improved and redone the Morris water maze (MWM) experiments according to the standard MWM protocol (Nat. Protoc. 2006, 1, 848-858). MWM performance was assessed on days 13-18 and 28-32 following injury (n=7 per group). Before spatial learning, visual discrimination learning was performed to determine whether the vision of mice was normal and mice with visual problems were abandoned. In this procedure, each animal performed one trial where the platform was placed above the water level without recording. Spatial learning was assessed across repeated trials for 5 days approximately at the same time every day between 13:00 and 18:00. The water maze was divided into four quadrants (I, II, III and IV), and the platform was set in the center of quadrant I. Mice were given four trials

a day using a starting position randomly within four positions. Each trial is limited within 60 s with an inter-trial interval (ITI) of 15 s. The mouse failing to find the platform hidden behind the water within this time limit was allowed 15 s to be placed on the platform or guided to learn. The trials were repeated for 5 days on days 13-17 and 28-31, and the probe trial that intended to assess reference memory was administered for 60 s on day 18 and day 32. The error bars were SD in our previous work, but in the revised manuscript are presented as mean \pm standard error of the mean (SEM), *P < 0.05, **P < 0.01, and ***P < 0.001 versus the Sham group, as analyzed by one-way analysis of variance (ANOVA). Following reviewer's advice, latency to locate and rest on the hidden platform, and the distance traveled (path length) to hidden platform were recorded for spatial learning trials. During the probe trial, the percentage of time spent in the missing platform quadrant and the number of platform location crossings were recorded to analyze the search strategy of mice in each group (*Neurosci. Biobehav. R.* 2018, 88, 187-200; *PNAS*, 1999, 96, 13427-13431). As shown in **Figure S34a** and **b**, all the mice apparently learned the task during the acquisition phase on days 13-17 and 28-31, while the distance traveled and latency to hidden platform with Au₂₄Cu₁ and Au₂₄Cd₁ treatment obviously decrease. For the probe trial on day 18 and day 32 (**Figure S34c** and **d**), the percentage time in the missing platform quadrant and the number of platform crossings significantly was reduced in the TBI group, but almost returned to the normal level after Au₂₄Cu₁ and Au₂₄Cd₁ treatment. These results reveal trends in the improvements of learning ability and spatial memory with Au₂₄Cu₁ and Au₂₄Cd₁ treatment. The corresponding experimental details, results and discussion have been added in the revised manuscript.

Changed in the “Revised Manuscript” (Lines 2-10, Page 16):

“Moreover, behavioral tests were studied by the Morris water maze. As shown in **Figure S34a** and **b**, all the mice apparently learned the task during the acquisition phase of days 13-17 and 28-31, while the distance traveled and latency to hidden platform with Au₂₄Cu₁ and Au₂₄Cd₁ treatment obviously decreased. For the probe trial on day 18 and day 32 (**Figure S34c** and **d**), the percentage time in the missing platform quadrant and the number of platform crossings was significantly reduced in the TBI group, but almost returned to the normal level after Au₂₄Cu₁ and Au₂₄Cd₁ treatment. These results reveal trends in the improvements of learning ability and spatial memory with Au₂₄Cu₁ and Au₂₄Cd₁ treatment.”

Changed in the “Supporting Information” (Lines 14-25, Page 13 and Lines 1-9, Page 14):

“Morris water maze tests: Morris water maze (MWM) performance was assessed on days 13-18 and 28-32 following injury (n=7 per group). The hidden platform test was used to investigate spatial learning and memory. Water in the pool was maintained at 25 °C (\pm 1 °C), made opaque by milk. Before spatial learning, visual discrimination learning was performed to determine whether the vision of mice was normal. In this procedure, each animal performed one trial where the platform was placed above the water level without recording. Spatial learning was assessed across repeated trials for 5 days approximately at the same time each day between 13:00 and 18:00. A circular stainless steel tank 122 cm in diameter and 51 cm in height on both sides with non-reflective interior surfaces was used. The water maze was divided into four quadrants (I, II, III and IV), and the platform was set in the center of quadrant I. Mice were given four trials a day with an inter-trial interval (ITI) of 60 min using a starting position randomly within four positions. The procedures of MWM followed a previously published protocol.⁵ Each trial was limited within 60 s with an ITI of 15 s. The mouse failing to find the platform hidden behind the water within this time limit was allowed for 15 s to be placed on the platform or guided to learn. The trials were repeated for 5 days, and the probe trials to evaluate long-term spatial memory were administered for 60 s on day 6 at a new starting position with the platform removed. Latency to locate and rest on the hidden platform and the distance traveled (path length) to the hidden platform were recorded for spatial learning trials. During the probe trials, the percentage of time spent in the missing platform quadrant and the number of platform location crossings were recorded to analyze the search strategy of mice in each group.”

Added in the “Supporting Information” (Lines 10-11, Page 58):

“5. Vorhees, C. V. *et al.* Morris water maze: procedures for assessing spatial and related forms of learning and memory. *Nature Protocols*. **1**, 848-858 (2006).”

Changed in the “Supporting Information” (Figure S34, Page 49)

Figure S34. Morris water maze tests on days 13-18 and 27-32 in all groups (n=7 per group). **a** Distance traveled (path length) to the hidden platform and **b** latency to locate and rest on the hidden platform recorded for spatial learning trials on days 13-17 and 27-31. **c** Percentage of time during the probe trial spent in the quadrant of tank that previously housed the hidden platform and **d** the number of platform location crossings recorded for the probe trial on day 18 and 32.

Author's Response to Reviewer #3

In this paper, the authors designed a unique artificial enzyme, clusterzyme, by single-atom substitutions in MPA protected Au cluster. Interestingly, the clusterzymes not only showed high antioxidant property, but also performed selective enzyme-like activity towards GPX, CAT, SOD. Theoretical calculation revealed that the catalysis is modulated by the single active site of Au₂₄Cd₁ and the Cu-Au dual active sites of Au₂₄Cu₁ at bond lengths. Importantly, such clusterzymes have showed therapeutic effect on neuroinflammation in TBI model. The work not only provide a new type of artificial enzymes by precise design of Au cluster at single atom level, but also provide fundamental understanding on these clusterzymes and confirmed their bio-functionality. In general, the clusterzymes proposed in this paper may be a new type of nanozymes. Therefore, I think it merits consideration for publication on Nature Communications. Below are some concerns and questions.

1. In page 3, introduction section, for POD-like activity, the description for “the affinity to substrates” is not clear, as there are two substrates in POD-like activity. The authors need to clarify it.

Reply: Thank you very much for the careful review. The substrates we describe for POD activity in “the affinity to substrates” is H₂O₂. We have supplemented the detailed description in the revised manuscript. Thank you again for reminding us.

Changed in the “Revised Manuscript” (Lines 7-14, Page 3):

“However, the Michaelis-Menten constant (K_m) to the H₂O₂ substrate of gold nanoparticles towards the POD enzymatic reaction is below 1 mM, but the catalytic activity is weak.¹⁹ In contrast, Pt-based materials generally confer a high overall catalytic activity but it can only show a good H₂O₂ substrate affinity when K_m up to 16.7 mM^{14,20} and modulation of selective catalysis often needs to be purposely realized through rationally designed combination with other catalysts.²¹ Meanwhile, metal oxides have also revealed great potentials as enzyme mimetics.²² Typically, Fe₃O₄ nanoparticles display the POD-like activity,^{23,24} but are limited by their affinity to the H₂O₂ substrate (K_m at ~ 154 mM)”

2. In the paper, the authors demonstrated that three clusterzymes showed preferential enzyme-like activities, Au₂₅ for GPx, Au₂₄Cu₁ for CAT and Au₂₄Cd₁ for SOD. Is it possible to measure the specific activity (U/mg) for each clusterzyme and compare it with the corresponding natural enzyme?

Reply: Thanks for the constructive suggestion. Indeed, the comparison of activity with natural enzymes is necessary in the development of a new artificial enzyme, and we also agree on its importance. Per reviewer's suggestion, we calculated the specific activities of various clusterzymes and compared them with natural enzymes. According to the formula of GPx activity in the detection system = (A₃₄₀/min)/(ε^{μM} × L(cm) × protein concentration in sample) and the specific activity of clusterzymes were obtained. As we can see from **Figure R3a**, the Au₂₅ shows the strongest tendency towards GPx-like activity with a specific activity of 5.4 U/mg, higher than 3.8 U/mg for Au₂₄Cu₁ and 2.3 U/mg for Au₂₄Cd₁. Au₂₅ has the highest GPx-like activity, which is comparable to the result of 25 U/mL natural GPx, indicating the excellent GPx-like activity of Au₂₅ (**Figure R3b**). Similarly, the CAT specific activity of the clusterzymes were calculated and compared with that of the natural catalase (**Figure R4**). According to the definition of catalase activity: 1 enzyme activity unit (1 unit) can catalyze the decomposition of 1 mmol H₂O₂ in 1 minute at 25°C, pH 7.0. Therefore, the specific activity of CAT in clusterzymes can be calculated. Through calculation, we found that Au₂₄Cu₁ has the highest specific activity of CAT, which is 1.76 U/mg equivalent to 9.2 U/mL of natural catalase. According to the definition of SOD activity, 50 % of NBT photochemical reduction can be inhibited as an enzyme activity unit (1 unit). We calculated the SOD specific activity of clusterzymes and compared it with that of natural SOD (**Figure R5**). The SOD-like activity of pure Au₂₅ can only inhibit 31 % of the substrate with a specific activity of 56.3 U/mg, while one Cd atom substitution considerably increases the inhibition rate to 89 % with a specific activity of 400 U/mg, empowering SOD-like selectivity. This result shows that Au₂₄Cd₁ at 10 ng/μL is comparable with that using 94 U/mL natural SOD, indicating the excellent SOD-like activity of Au₂₄Cd₁.

Figure R3. **a** The GPx-like activity of clusterzymes. **b** GPx-like activities with different clusterzymes and different concentrations of natural GPx.

Figure R4. **a** The CAT-like activity of clusterzymes. **b** CAT-like activities with different clusterzymes (20 ng/ μ L) and different concentrations of natural catalase.

Figure R5. **a** The SOD-like activity of clusterzymes. **b** SOD-like activities with different concentrations of clusterzymes and natural catalase.

3. In Figure S23, the cell viability of HT22 sharply decreased when $Au_{24}Cd_1$ up to 100 $\mu\text{g/mL}$.

Reply: Thank you for constructive comment. We conducted three repeated toxicity tests on HT22 cells with different concentrations of $Au_{24}Cd_1$ and found that the survival rate did decrease sharply at 100 $\mu\text{g/mL}$. Taking into account the effect of clusterzymes on different cell types, we selected two other mouse neuronal cell lines microglia BV2 and cerebellar astrocyte MA-c cells for MTT assays to study the effect of clusterzymes on the survival rate of other neuronal cells. As shown in **Figure R6**, the survivals were still higher than 80% in both types of cells at the concentration of 100 $\mu\text{g/mL}$. Based on the above results, we hypothesized that $Au_{24}Cd_1$ has certain selective toxicity to different nerve cells. This phenomenon has been proved in the toxicological studies of gold nanomaterials (*ACS Nano*, 2012, 6, 5767-5783). In addition, the dose of all cell staining experiments we used was much lower than 100 $\mu\text{g/mL}$, with the highest concentration being only 20 $\mu\text{g/mL}$. At this concentration, the cell survival rate can reach more than 90 % and thus does not significantly affect the cell viability. Even so, considering the issue of biological safety, we conducted toxicological studies in animals. We used C57/BL6 mice (male, 7-9 weeks) and injected 200 μL of 5 mg/mL clusterzymes through the tail vein. After 7 days, the mice were sacrificed and histology (**Figure S36**), blood biochemistry (**Figure S37**) and hematology (**Figure S38**) were analyzed to evaluate the *in vivo* toxicity of the clusterzymes. The results showed that no significant changes were found in major organs as well as in blood biochemistry and hematology. Doses of up to 50 mg/kg fully met subsequent requirements of treatment of TBI, suggesting that renally-clearable clusterzymes do not cause significant biosafety hazards at higher doses.

Figure R6. Cell viability of different cell lines in the presence of clusterzymes. **a** HT22, **b** BV2 and **c** MA-c cells.

Changed in the “Supporting Information” (Lines 9-11 Page 9):

“Cytotoxicity assay: HT22 cells (2×10^3), BV2 cells (3×10^3) and MA-c cells (4×10^3) were seeded in 96-well plates filled with 0.01 M phosphate buffer solution (PBS) (Gibco) at the border in 100 μL medium overnight.”

Added in the “Supporting Information” (Lines 19-25, Page 14; Lines 1-12, Page 15):

“Pharmacokinetic and toxicological studies: ... Hematological and biochemical panels were analyzed on day 7 post injection. Blood samples were taken from the retro-orbital sinus in K2EDTA tubes. Biochemical blood samples were left to stand for 30 min and then centrifuged at 3500 rpm for 15 min twice. All organs were collected and fixed in 4 % PFA for 24-48 hours, embedded in paraffin and mounted on slides (4- μm coronal sections). Slides were stained through hematoxylin-eosin (H&E) staining to observe the toxicity of clusterzymes in each organ including heart, liver, spleen, lung, kidney and brain.”

Changed in the “Revised Manuscript” (Figure 5, Page 28):

Figure 5. Oxidative stress levels *in vitro* and *in vivo* before and after treatment of clusterzymes. **a** HT22 cell viability of clusterzymes. **b** HT22 cell viability in the presence of H₂O₂ with or without treatment of clusterzymes as determined by MTT assays. Fluorescence quantification of cell staining for **c** and **e** \bullet OH and **d** and **f** O₂⁻ by flow cytometry. Fluorescence microscopic images of intracellular **g** \bullet OH (green) and **h** O₂⁻ (red) levels induced by 100 μ M H₂O₂ with or without clusterzymes by HPF and DHE probes, respectively. It can be seen that Au₂₄Cu₁ has a better scavenging ability for \bullet OH, but Au₂₄Cd₁ shows better specificity for O₂⁻, suggesting their individual selectivity for \bullet OH and O₂⁻ respectively. **i-l** Indicators for oxidative stress including SOD, GSH/GSSG, MDA and H₂O₂ of TBI mice with or without treatment of clusterzymes 1, 3, and 7 days post injury (n=5 per group). Data are presented as mean \pm SEM; **P* < 0.05, ***P* < 0.01, and ****P* < 0.001 compared with the Sham group, analyzed by ANOVA.

Changed in the “Supporting Information” (Figure S25, Page 40):

Figure S25. Cell viability of **a** BV2 and **b** MA-c cells in the presence of various concentrations of clusterzymes determined by MTT assays.

Added in the “Supporting Information” (Figure S36-38, Pages 51-53):

Figure S36. Histology of major organs in mice (heart, liver, spleen, lung, kidney, bladder and testis) treated with 200 μ L clusterzymes at the concentration of 5 mg/mL after 7 days. Scale bar is 50 μ m, n=3 per group. No significant toxic responses were found in all organs.

Figure S37. Hematology of mice 7 days after being treated with 200 μ L clusterzymes at the concentration of 5 mg/mL. The results show the mean and standard error of the mean of white blood cells (WBC), red blood cell (RBC), hematocrit (HCT), mean corpuscular volume (MCV), hemoglobin (HGB), platelets (PLT), mean corpuscular hemoglobin (MCH), and mean corpuscular hemoglobin concentration (MCHC), $n=3$ per group. Data are presented as mean \pm SEM; * $P < 0.05$, ** $P < 0.01$, and *** $P < 0.001$ versus the Sham group, analyzed by ANOVA.

Figure S38. Blood biochemistry analysis of mice 7 days after being treated with 200 μ L clusterzymes at the concentration of 5 mg/mL. The results show mean and standard error of the mean of ureal, alanine aminotransferase (ALT), albumin (ALB), aspartate aminotransferase (AST), total protein (TP), creatinine (CREA), and total bilirubin (TB), $n=3$ per group. Data are presented mean \pm SEM; * $P < 0.05$, ** $P < 0.01$, and *** $P < 0.001$ versus with the control group, analyzed by ANOVA.

Added in the “Revised Manuscript” (Lines 10-15, Page 16):

“In addition, we systematically studied the pharmacokinetics and toxicology of clusterzymes. It can be seen that the clusterzymes accumulated in major organs can be removed by the kidney (urine) and liver (feces). After 48 hours, ~

80% of the total dose can be excreted, and most of it is excreted through the kidney (more than 70%) (**Figure S35**). No significant changes in organs or blood chemistry or hematology were found, suggesting that renal clearable clusterzymes do not cause significant biological toxicity *in vivo* (**Figure S36-38**).”

4. The authors observed that the clusterzymes can significantly decrease inflammation factors such as IL-1 β , IL-6, and TNF- α . Can the authors provide some explanation for the possible pathways?

Reply: Thanks for your constructive comment. NF- κ B is a transcription factor of eukaryotes, involved in the expression of a variety of cytokines and adhesion molecules, and is a key factor in regulating inflammation, cell proliferation, differentiation and apoptosis. The activation of the NF- κ B pathway requires the activation of I κ B, which causes the phosphorylation, ubiquitination and degradation of the inhibitor protein I κ B, releases the p50-p65 heterodimer to be transported to the nucleus, and regulates the corresponding gene transcribe and acts as a transcription factor to promote the production of inflammatory factors such as TNF- α , IL-1 β , and IL-6, as well as the production of NO, reactive oxygen species (ROS), and peroxynitrite, triggering an inflammatory response (*Mol. Cancer* 2013, 12, 86; *PNAS*, 2019, 116, 9453). Therefore, the degradation and phosphorylation of I κ B proteasome are the main signs of NF- κ B activation and the subsequent production of pro-inflammatory factors. Therefore, we monitored the expression of p-I κ B α in the brain of TBI mice 3 days before and after treatment by Western blot to explore the pathways for nanozymes to reduce inflammatory factors. We used anti-I κ B α (phospho S36, EPR6235(2)) antibody to detect I κ B α only when it is phosphorylated at serine 36. As shown in **Figure R7a**, the high expression of p-I κ B α in the brain of TBI mice indicates that the NF- κ B pathway is activated and a large number of inflammatory factors are produced, consistent with our previous ELISA and Western blot results on inflammatory cytokines (**Figure 6e-g**). In the brain of TBI mice treated with Au₂₄Cd₁ and Au₂₄Cu₁, the phosphorylation level is low, which is close to the level of the sham group. Through quantitative analysis, we can quantify the relative expression of p-I κ B α protein (**Figure R7b**). It can be seen that the expression of p-I κ B α in the brain of TBI mice after Au₂₄Cd₁ and Au₂₄Cu₁ treatment is significantly inhibited, which is close to that in the sham group. Therefore, we could infer that clusterzymes can reduce the production of inflammatory factors by inhibiting the NF- κ B pathway.

Figure R7. a Western blot of p-I κ B α in brain tissues after treatment 3 days post TBI. **b** Relative protein expressions of p-I κ B α .

5. What is the pharmacokinetics of clusterzymes *in vivo*? Due to ultrasmall size, I am wondering if it has a very short half-life *in vivo*. In addition, it is not clear whether the clusterzymes can pass through BBB for TBI therapy.

Reply: Thanks for your constructive comment. According to reviewer's opinions and considering the safety of the material, we performed pharmacokinetic and toxicological studies of the clusterzymes. We used male C57/BL6 mice (7-9 weeks), injected 200 μ L 5 mg/mL clusterzymes for animal experiments. By measuring Au concentrations in the blood over time, it can be seen that the blood half-life of all clusterzymes are basically the same, 45 min for Au₂₅, 44 min for Au₂₄Cu₁, and 40 min for Au₂₄Cd₁ (**Figure S35a**). From the excretion curves in **Figure S35b**, it can be seen that the clusterzymes can be cleared by the kidney (urine) and liver (feces), and after 48 hours, \sim 80% of the total dose can be excreted, most of which was through the kidney (more than 70%). Biodistribution of clusterzymes in kidney, liver, spleen and intestines was analyzed 1 day after injection and provided additional insights on the clearance pathways primarily through the urinary tract, with high contents in the bladder (**Figure S35c**).

Whether the therapeutic drugs can pass through the blood-brain barrier is one of the key factors affecting the therapeutic effect. In order to verify whether the cluster enzyme can penetrate the blood-brain barrier of TBI mice and display a potential therapeutic effect, we studied the brain uptake of TBI mice at different time points. In the view of the non-clinical applicability of scalp administration, we changed the way of administration and adopted traditional intravenous injection of clusterzymes through the tail vein. We injected 200 μ L of 5 mg/mL Au₂₅, Au₂₄Cd₁, and Au₂₄Cu₁ and 0.01 M PBS into TBI mice, n=12 in each group. The mice were sacrificed at 1 h, 4 h, 12 h, and 24 h, and blood was washed with 0.01 M PBS until the fluid from the right ventricle became clear. The mouse brains were then dissected, weighed, and the content of Au was determined by ICP-MS to determine the amount of clusterzymes passing through the BBB. It can be seen from **Figure S35d** that the brains of mice in the vehicle group hardly contained any Au element, while the content of clusterzymes in the brains of mice in the administration group is significantly increased. The Au content in mouse brains reaches the peak within 1-4 hours and then decreases. After 24 hours, the concentrations of all clusterzymes in brains were still high, indicating that clusterzymes could exert a persistent therapeutic effect through BBB of TBI mice.

In addition, considering the biosafety concerns, we have conducted pharmacokinetic and toxicological studies in animals. We used male C57/BL6 mice of 7-9 weeks, and injected 200 μ L 5 mg/mL, of clusterzymes, through the tail vein, up to 50 mg/kg. After 7 days, we assessed the biosafety of clusterzymes by examining histology (**Figure S36**), blood biochemistry (**Figure S37**) and hematology (**Figure S38**) in mice. The results showed there were no significant changes in organs or blood chemistry or hematology, suggesting that renally clearable clusterzymes did not cause significant biological safety concerns.

Added in the “Supporting Information” (Lines 19-22, Page 11):

“BBB penetration: Brain tissues were harvested from mice with CCI injury by flushing blood from blood vessels through the heart with cold PBS at the time points of 1, 4, 12 and 24 h post injection (n=3 per group per time point). Brain tissues were weighed and detected for the Au element by ICP-MS to evaluate the BBB penetration.”

Added in the “Supporting Information” (Lines 19-25, Page 14; Lines 1-12, Page 15):

“Pharmacokinetic and toxicological studies: The pharmacokinetic parameters of clusterzymes were measured on 7-9 weeks (21-23 g) male C57BL/6J mice. Mice were subject to intravenous injection at a dose of 50 mg/kg in 200 μ L volume to evaluate the biodistribution (n=3 per group), blood half-life (n=3 per group) and excretion (n=3 per group) of different clusterzymes. Mouse organs (heart, lung, liver, spleen, kidney, muscle, bladder, testicles, intestine and brain) were collected, washed with PBS and weighed 24 h post injection. To determine the half-life of clusterzymes, blood was collected from the retro-orbital sinus at 2 min, 9 min, 24 min, 54 min, 114 min, 3 h, 8 h, 24 h, and 48 h and the volume was 50 μ L. The elemental Au in organs and blood was quantified with ICP-MS. Standards were prepared and counted along with tissue samples to calculate the percentage-injected dose per gram of tissue (%ID/g). Drug excretions of clusterzymes were determined as well. Stool and urine were collected within 48 hours and detected for Au element. Hematology and blood biochemistry panels were detected on the day 7 post injection. Blood samples were obtained from retro-orbital sinus and saved in tubes with K2EDTA for testing. Blood samples for biochemistry analysis were left to stand for 30 min and then centrifuged twice at 3500 rpm for 15 min. All organs were collected and fixed in 4% PFA for 24-48 hours, embedded in paraffin and mounted on slides (4- μ m coronal sections). Slides were stained through hematoxylin-eosin (H&E) staining to observe the toxicity of clusterzymes in major organs including heart, liver, spleen, lung, kidney and brain.”

Added in the “Supporting Information” (Figure S35, Page 50):

Figure S35. **a** The time activity curves of clusterzymes in blood. **b** Cumulative total excretion including urine and feces of clusterzymes with collected time up to 48 h, showing excellent excretion. **c** Biodistribution of clusterzymes 1 day p.i. (heart, lung, liver, spleen, kidney, muscle, bladder, testicles, intestine and brain, % ID/g = percentage of the injected dose per gram of tissue). **d** Brain uptake and BBB permeability. Injected dose: 5 mg/mL, 200 μL, n=3 per group.

Added in the “Supporting Information” (Figure S36-38, Pages 51-53):

Figure S36. Histology of major organs in mice (heart, liver, spleen, lung, kidney, bladder and testis) treated with 200 μL clusterzymes at the concentration of 5 mg/mL after 7 days. Scale bar is 50 μm, n=3 per group. No significant toxic responses were found in all organs.

Figure S37. Hematology of mice 7 days after being treated with 200 µL clusterzymes at the concentration of 5 mg/mL. The results show the mean and standard error of the mean of white blood cells (WBC), red blood cell (RBC), hematocrit (HCT), mean corpuscular volume (MCV), hemoglobin (HGB), platelets (PLT), mean corpuscular hemoglobin (MCH), and mean corpuscular hemoglobin concentration (MCHC), n=3 per group. Data are presented as mean ± SEM; **P* < 0.05, ***P* < 0.01, and ****P* < 0.001 versus the Sham group, analyzed by ANOVA.

Figure S38. Blood biochemistry analysis of mice 7 days after being treated with 200 µL clusterzymes at the concentration of 5 mg/mL. The results show mean and standard error of the mean of ureal, alanine aminotransferase (ALT), albumin (ALB), aspartate aminotransferase (AST), total protein (TP), creatinine (CREA), and total bilirubin (TB), n=3 per group. Data are presented mean ± SEM; **P* < 0.05, ***P* < 0.01, and ****P* < 0.001 versus with the control group, analyzed by ANOVA.

“In addition, we systematically studied the pharmacokinetics and toxicology of clusterzymes. It can be seen that the clusterzymes accumulated in major organs can be removed by the kidney (urine) and liver (feces). After 48 hours, ~80% of the total dose can be excreted, and most of it is excreted through the kidney (more than 70%) (Figure S35). No significant changes in organs or blood chemistry or hematology were found, suggesting that renal clearable clusterzymes do not cause significant biological toxicity *in vivo* (Figure S36-38).”

6. How is the stability of clusterzymes *in vivo*? Can it be degraded under physiological conditions?

Reply: Thanks for your constructive comment. According to reviewer’s suggestion, we studied the stability of clusterzymes *in vivo*. C57 mice were injected with 200 μL 5 mg/mL clusterzymes through tail vein, and urine was collected. We used 3 K and 10 K ultrafiltration tubes at 3500 rpm/min to remove the smaller and larger parts, and tested for the antioxidation and catalytic abilities. As shown in the upper inset of Figure R8, it can be seen that we have obtained urine containing the same concentration of clusterzymes. Total antioxidant capacity tests show that urine containing clusterzymes still had higher activity compared with fresh samples (Figure R8a and following inset). Figure R8b show a photo of a centrifuge tube incubated for 2 h containing 1 M H_2O_2 with or without 20 μL clusterzyme-containing urine. It can be seen that clusterzymes still have good CAT enzyme activity, and $\text{Au}_{24}\text{Cu}_1$ had the optimal CAT activity, which was consistent with the trend in *in vitro* experiments. In summary, it can be seen that clusterzymes retained good stability *in vivo*, and still had high activity after a period of time under physiological conditions, and will not be degraded basically.

Figure R8. Study on the stability of clusterzymes *in vivo*. C57 mice were injected with 5 mg/mL 200 μL clusterzymes *via* tail vein, and urine was collected. **a** Antioxidant capacity were tested (the upper inset shows the collected urine of clusterzymes, and the following inset shows the $\text{ABTS}^{+\cdot}$ color reaction). **b** Photo of 1 M H_2O_2 decomposition capacity of urine containing clusterzymes.

7. A lot of typo errors in the paper, such as “activity is week”, exploited, theBMPO, The H_2O_2 , severeoxidative, areupregulated, showminor, THFa(in Supplementary Information). The authors need to check and revise the language carefully.

Reply: We sincerely thank the reviewer for careful reading. We feel sorry for our omission. As suggested by the reviewer, we have corrected the errors. In addition, we have checked the manuscript many times to prevent such errors from happening again.

Changed in the “Revised Manuscript” (Line 9, Page 3):

“activity is **weak**.”

Changed in the “Revised Manuscript” (Lines 19-20, Page 8):

“**the BMPO** control”

Changed in the “Revised Manuscript” (Line 7, Page 14):

“severe oxidative damage”

Changed in the “Revised Manuscript” (Line 1, Page 15):

“show minor”

Changed in the “Supporting Information” (Lines 7-9, Page 12):

“Enzyme-linked immunosorbent assay (ELISA) kits for IL6 (Abcam, ab100712), IL1 β (Abcam, ab197742) and TNF α (Abcam, ab208348) were used to detect inflammation levels.”

8. *The reference style is not consistent and need to be carefully revised.*

Reply: Thank you very much for the careful review. We have carefully rechecked and corrected the references in the whole revised manuscript.

Changed in the “Revised Manuscript” (References 11, 21, 25, 33 and 62):

“Li, S. *et al.* A Nanozyme with Photo-Enhanced Dual Enzyme-Like Activities for Deep Pancreatic Cancer Therapy. *Angew. Chem. Int. Ed.* **58**, 12624-12631 (2019).”

“Huang, Y. *et al.* Self-Assembly of Multi-nanozymes to Mimic an Intracellular Antioxidant Defense System. *Angew. Chem. Int. Ed.* **55**, 6646-6650 (2016).”

“Singh, N. *et al.* A redox modulatory Mn₃O₄ nanozyme with multi-enzyme activity provides efficient cytoprotection to human cells in a Parkinson's disease model. *Angew. Chem. Int. Ed.* **56**, 14267-14271 (2017).”

“Xu, B. *et al.* A Single-Atom Nanozyme for Wound Disinfection Applications. *Angew. Chem. Int. Ed.* **58**, 4911-4916 (2019).”

“Karayilan, M. *et al.* Catalytic Metallopolymers from [2Fe-2S] Clusters: Artificial Metalloenzymes for Hydrogen Production. *Angew. Chem. Int. Ed.* **58**, 7537-7550 (2019).”

REVIEWER COMMENTS

Reviewer #1 (Remarks to the Author):

In the revised manuscript, the authors have adequately addressed my questions and improved the manuscript, esp. the discussions on the atomic-level catalytic mechanism.

I suggest the revised manuscript be accepted.

Reviewer #2 (Remarks to the Author):

NCOMMS-20-18263A revision comments

The authors have addressed my prior concerns by repeating major drug interventions studies using a clinically relevant IV administration route, adding acute time points for neuroinflammation assessments, and by repeating the Morris water maze studies using established behaviour protocols. The addition of this new preclinical data significantly improves the quality and rigor of the in vivo neuroprotection studies and support the concept that their novel and highly potent clusterzymes inhibits secondary oxidative stress and neuroinflammation after TBI to improve cognitive function recovery.

There are some minor details in the supplementary methods that need to be revised/clarified prior to publication.

1) The authors need to provide details of methods of quantification of immunofluorescence and immunohistochemical studies in the supplementary materials. For each protein marker it is not clear what "cells pixel"/mm² refers to? Please carefully define the quantitative measures and provide a detailed method (related to Fig S26-S27). Similar clarity is needed for Fig S32 immunohistochemistry (arbitrary units, a.u., are not appropriate for this type of analysis).

2) Careful copyediting for typos/grammar should be performed prior to publication.

Reviewer #3 (Remarks to the Author):

The authors have carefully addressed all the reviewers' concerns and improved the manuscript. Therefore, I recommend it to be considered for publication on Nature Communications.

Author's Response to Reviewer 2

The authors have addressed my prior concerns by repeating major drug interventions studies using a clinically relevant IV administration route, adding acute time points for neuroinflammation assessments, and by repeating the Morris water maze studies using established behaviour protocols. The addition of this new preclinical data significantly improves the quality and rigor of the in vivo neuroprotection studies and supports the concept that their novel and highly potent clusterzymes inhibits secondary oxidative stress and neuroinflammation after TBI to improve cognitive function recovery. There are some minor details in the supplementary methods that need to be revised/clarified prior to publication.

1. The authors need to provide details of methods of quantification of immunofluorescence and immunohistochemical studies in the supplementary materials. For each protein marker it is not clear what “cells pixel”/mm² refers to? Please carefully define the quantitative measures and provide a detailed method (related to Fig S26-S27). Similar clarity is needed for Fig S32 immunohistochemistry (arbitrary units, a.u., are not appropriate for this type of analysis).

Reply: Thank you very much for precise suggestion. Based on the reviewer comments, we have added the details of methods of quantification for immunostaining in supporting information. In order to observe the changes in the morphology and number of immune cells in brain tissues before and after clusterzymes treatment, we performed immunofluorescence staining. Quantitative image analysis of the immunofluorescence for GFAP, Iba-1, or NeuN cells were performed on five cerebral cortex areas taken with the $\times 40$ objective and using the same optical density analysis method as previously reference (*Nat. Immunol.*, 2018, 19, 442-452; *Brain*, 2019, 142, 3440-3455; *ACS Nano*, 2019, 13, 5591-5601). Immunofluorescence intensity was calculated using the threshold method and defined as the number of pixels by ImageJ software, then divide by the total area (mm²) in the imaged field with the average background subtracted. The result “cells pixel/mm²” represents the pixel density of different immune cells in the cerebral cortex.

To trace the secretion source of inflammatory factors, we performed immunofluorescence double staining on several immune cells and inflammatory factors. For quantification of the immunofluorescence double staining, the co-expressed cells in the five regions of the cortex were counted under a microscope (EVOS, AMG) at $\times 400$ magnification. The results are expressed as an average number of positive cells per unit area (mm²) (*Nature*, 2014, 505, 223-228; *Nat. Neurosci.*, 2020, 23, 194-208; *Nat. Commun.*, 2020, 11, 4524) (**Figure S26-S27**).

In addition, in order to further determine the changes of inflammatory factors before and after clusterzymes treatment, we performed immunohistochemical staining on brain slices. For the quantitative analysis of inflammatory factors in immunohistochemistry, the investigators who were blinded to the experimental groups randomly collected five high-power field images at $\times 400$ magnification in cerebral cortex area per animal under a microscope (EVOS, AMG) (*Nat. Neurosci.*, 2020, 23, 194–208). The cytoplasmic staining areas that showed light yellow or brownish yellow were selected as positive cells, and the expression of inflammatory factors was quantified by the average count of positive staining cells under a random field of view (**Figure S32**). The specific details have been supplemented in the "Supporting information". Thanks again for the valuable comments of the reviewer.

Changed in the “Supporting Information” (Lines 13-25, Page 15; Lines 1-2, Page16):

“Quantitative analysis of immunostaining

Quantitative image analysis of the immunofluorescence for GFAP, Iba-1, and NeuN cells were performed on five cerebral cortex areas taken with the $\times 40$ objective. Immunofluorescence intensity was calculated using the threshold method and defined as the number of pixels by ImageJ software, then divide by the total area (mm²) in the imaged field with the average background subtracted.^{6,7} For quantification of the immunofluorescence double staining, the co-expressed cells in the five regions of the cortex were counted under a microscope (EVOS, AMG) at $\times 400$ magnification. The results are expressed as an average number of positive cells per unit area (mm²).^{8,9} For the

quantitative analysis of inflammatory factors in immunohistochemistry, the investigators who were blinded to the experimental groups randomly collected five high-power field images at $\times 400$ magnification in cerebral cortex areas under a microscope (EVOS, AMG).⁹ The cytoplasmic staining areas that showed light yellow or brownish yellow were selected as positive cells, and the expression of inflammatory factors was quantified by the average count of positive staining cells”

Added in the “Supporting Information” (References 6-9, Page 59):

- “6. Russo, M. V. *et al.* Distinct myeloid cell subsets promote meningeal remodeling and vascular repair after mild traumatic brain injury. *Nat. Immunol.* **19**, 442-452 (2018).
7. Nutma, E. *et al.* A quantitative neuropathological assessment of translocator protein expression in multiple sclerosis. *Brain* **142**, 3440-3455 (2019).
8. Roth, T. L. *et al.* Transcranial amelioration of inflammation and cell death after brain injury. *Nature* **505**, 223-228 (2014).
9. Marschallinger, J. *et al.* Lipid-droplet-accumulating microglia represent a dysfunctional and proinflammatory state in the aging brain. *Nat. Neurosci.* **23**, 194-208 (2020).”

Changed in the “Revised Manuscript” (Figure 6, Page 29):

Figure 6. Inflammation levels in brain tissues after clusterzyme treatment. **a-c** Western blotting for IL-1 β , IL-6, and TNF α in the brain tissues 1, 3, and 7 days post TBI after treatment (n=3 per group), respectively. **d** Western blotting quantitative analysis of inflammatory factors at different time points. It can be seen that Au₂₄Cd₁ can rapidly and significantly reduce the upregulated inflammatory cytokines of IL-1 β and IL-6 after brain injury, while Au₂₄Cu₁ has a better ability to reduce the expression of TNF α . **e-g** ELISA quantitative analysis of IL-1 β , IL-6, and TNF α levels in brain tissues on day 1, 3, and 7 with or without clusterzyme treatment (n=5 per group), respectively. **h** Immunofluorescence co-staining of IL-1 β and microglia (Iba-1), astrocytes (GFAP) or neurons (NeuN) in injured

cortex 3 days post injury with or without clusterzyme treatment. Quantitative analysis of **i** the number of IL-1 β ⁺ expression in different positive cells and **j** the pixels density of Iba-1/NeuN/GFAP cells in the injured cortex with or without clusterzymes treatment (n=3 per group). Data are mean \pm SEM; **P* < 0.05, ***P* < 0.01, and ****P* < 0.001 compared with the Sham group, as analyzed by ANOVA.

Changed in the “Supporting Information” (Figures S26-S27, Pages 42-43):

Figure S26. **a** Immunofluorescence co-staining of IL-6 and microglia (Iba-1), astrocytes (GFAP) or neurons (NeuN) in injured cortex 3 days post injury with or without clusterzyme treatment. Quantitative analysis of **b** the number of IL-6⁺ expression in different positive cells and **c** the pixels density of Iba-1/NeuN/GFAP cells in injured cortex with or without clusterzymes treatment (n=3 per group). Data are mean \pm SEM; **P* < 0.05, ***P* < 0.01, and ****P* < 0.001 versus the Sham group, as analyzed by ANOVA.

Figure S27. **a** Immunofluorescence co-staining of TNF α and and microglia (Iba-1), astrocytes (GFAP) or neurons (NeuN) in injured cortex at 3 days post-injury with or without clusterzyme treatment. Quantitative analysis of **b** the number of TNF α ⁺ expression in different positive cells and **c** the pixels density of Iba-1/NeuN/GFAP cells in injured cortex with or without clusterzymes treatment (n=3 per group). Data are mean \pm SEM; **P* < 0.05, ***P* < 0.01, and ****P* < 0.001 versus the Sham group, as analyzed by ANOVA.

Changed in the “Supporting Information” (Figure S32, Page 48):

Figure S32. Quantitative analysis of immuno-histochemical results including IL-6, IL-1 β and TNF α on day 1, 3, and 7 post injury (n=3 per group).

2. Careful copyediting for typos/grammar should be performed prior to publication.

Reply: Thank you very much for the careful review. We have carefully rechecked and corrected the typos/grammar in the whole revised manuscript.